# Causal inference in medical records and complementary systems pharmacology for metformin drug repurposing towards dementia

Marie-Laure Charpignon [1,16], Bella Vakulenko-Lagun[2,16], Bang Zheng [3,16], Colin Magdamo[4], Bowen Su[5], Kyle Evans[4,6], Steve Rodriguez[4,6], Artem Sokolov [6], Sarah Boswell [6], Yi-Han Sheu[7], Melek Somai[8], Lefkos Middleton[3,9], Bradley T. Hyman [4], Rebecca A. Betensky[10], Stan N. Finkelstein[1,11], Roy E. Welsch[1,12], Ioanna Tzoulaki [5,13,14,17] ✉, Deborah Blacker[7,15,17] ✉, Sudeshna Das [4,17] ✉ & Mark W. Albers [4,6,17] ✉

Metformin, a diabetes drug with anti-aging cellular responses, has complex actions that may alter dementia onset. Mixed results are emerging from prior observational studies. To address this complexity, we deploy a causal inference approach accounting for the competing risk of death in emulated clinical trials using two distinct electronic health record systems. In intention-to-treat analyses, metformin use associates with lower hazard of all-cause mortality and lower cause-specific hazard of dementia onset, after accounting for prolonged survival, relative to sulfonylureas. In parallel systems pharmacology studies, the expression of two AD-related proteins, APOE and SPP1, was suppressed by pharmacologic concentrations of metformin in differentiated human neural cells, relative to a sulfonylurea. Together, our findings suggest that metformin might reduce the risk of dementia in diabetes patients through mechanisms beyond glycemic control, and that SPP1 is a candidate biomarker for metformin's action in the brain.

Repurposing drugs affords a route to therapeutic development that is shorter, less expensive, and more likely to succeed[1]. However, with fewer economic incentives for drug repurposing than for bringing new drugs to market, combined evidence from real-world data and mechanistic studies that supports the therapeutic hypothesis might justify a Randomized Clinical Trial (RCT). Alzheimer's disease (AD), with one to two decades of accumulating pathology prior to symptom onset, brings another challenge: a preclinical period so long that it is often not economically feasible for RCTs and presents ethical problems. Observational studies in Electronic Health Records (EHR) allow longer follow-up times than RCTs and offer the possibility of

evaluating drugs already approved by the Food and Drug Administration (FDA) and/or the European Medicines Agency (EMA) within the preclinical period of dementia. Using the target trial method[2,3] of conducting observational studies that aim to mimic RCTs, we emulated the same target trial in two distinct EHR systems. We reasoned that replicating results across two samples from vastly different settings—one a healthcare system anchored in two large tertiary care hospitals and another a nation-wide primary care network—would provide a more robust estimate of the generalizability of the drug's effect[4,5]. Moreover, differences in medical practice, data collection, timing and length of follow-up, patterns of missingness, and known

A full list of affiliations appears at the end of the paper. ✉e-mail: i.tzoulaki@imperial.ac.uk; dblacker@mgh.harvard.edu; sdas5@mgh.harvard.edu; albers.mark@mgh.harvard.edu

and unknown sources of bias between two EHR databases bolster any signal observed in both samples.

In this study, we emulated a target trial to estimate the effects of metformin compared to sulfonylureas on the risk of death and dementia. Metformin is a first-line antidiabetic drug with additional properties that may slow biological aging[6,7], including some evidence of increased survival[8]. Since the risk of dementia rises very steeply with age[9], it has been hypothesized that metformin would reduce such risk. Clinical studies of metformin, however, have had mixed results in association with dementia risk in older adults[10,11]. To address the potentially opposing influences of metformin on dementia—that it might reduce the hazard of death and therefore put more people at risk of developing dementia while reducing the hazard of dementia by slowing biological aging, we used a competing risks analysis framework[12]. We used the target trial method[2] to emulate a trial in the Research Patient Data Registry (US RPDR[13]) at Mass General Brigham (formerly Partners) Health Care system in the US and the UK Clinical Practice Research Datalink (UK CPRD[14]) database among initiators of metformin vs. the other first-line therapy for diabetes, the sulfonylureas (reference group).

In parallel, we conducted an in vitro systems pharmacology evaluation of both drugs on differentiated human neural cells in culture to identify genes whose expression is differentially altered in neural cells with metformin treatment relative to the vehicle and to glyburide, one of the sulfonylureas. The secreted products from these differentially expressed genes are candidate pharmacodynamic markers of metformin's actions in the brain, which can be quantified in the cerebrospinal fluid (CSF). Our EHR-based results may serve as an example of the use of real world data (RWD) to inform the design of clinical trial eligibility criteria[15] for a trial of metformin with the primary outcome of dementia onset. Further, our systems pharmacology studies may suggest a pharmacodynamic CSF biomarker for metformin's anti-aging actions in the human brain beyond its hypoglycemic actions.

## Results

### Target trial emulation in the EHR from the US RPDR and UK CPRD

We emulated the target trial in cohorts from the US RPDR and UK CPRD EHR databases (Table 1) with a 1-year run-in period. Our target trial outcomes were time to first diagnosis of dementia or death in type 2 diabetics over age 50, starting on metformin- or sulfonylurea-monotherapy, and followed for at least 1 year. Of note, the 1-year run-in period was selected to ensure sufficient drug exposure before measuring outcomes. While the duration of a clinical trial is usually fixed, the duration of follow-up in the emulated trial is often much longer (US RPDR median: 5.0 years (max 12 years); UK CPRD median: 6.0 years (max 16 years)).

The US RPDR cohort, which was drawn from patients receiving primary care at an academic health care system, included 13,191 patients who started on metformin- (11,229; 85%) or sulfonylurea- monotherapy (1962; 15%) (Fig. 1a). Patients who had a diagnosis of dementia, or died within the first year of follow-up, were excluded from the study population to emulate the standard exclusion criterion in clinical trials of patients with baseline cognitive impairment or a high morbidity index. In addition, patients with chronic kidney disease (CKD; see Extended Data Table 1 for definitions) at treatment initiation—a contraindication for metformin, but not for sulfonylureas—were excluded from the cohort. Metformin initiators were younger than their sulfonylurea counterparts (Table 2). Among the metformin initiators, there were more hypertensives and fewer missing values for baseline body mass index (BMI) than among the sulfonylurea initiators (Table 2). The baseline glycosylated hemoglobin (HbA1C) levels and other baseline characteristics, however, were comparable between the two groups (Table 2).

The UK CPRD cohort, which was drawn from primary care practices across 13 regions in the UK, included 108,025 patients in total with

94,208 (87%) metformin initiators and 13,817 (13%) sulfonylurea initiators (Fig. 1b). Patients who were diagnosed with dementia or died within the first year of follow-up were excluded from the study population (Fig. 1b). Those with CKD at treatment initiation (Extended Data Table 2) were also excluded. As in the US RPDR cohort, we found that patients treated with metformin were younger than patients treated with sulfonylureas in the UK CPRD cohort (Table 2). They were also more likely to have entered the cohort more recently and to have lower HbA1C and higher BMI at baseline. Further, they included more cardiovascular disease (CVD) and hypertension cases, but fewer cancer cases at baseline than the sulfonylurea group (Table 2).

No piece of information that identifies individual patients is presented in this paper.

### Metformin improved survival relative to the sulfonylureas in the US and UK cohorts

First, we compared the effect of metformin vs. sulfonylureas on all-cause mortality in both cohorts, since metformin use has previously been reported to improve *survival* relative to the sulfonylureas in distinct US[16] and UK[17] cohorts of type 2 diabetics.

In the US RPDR cohort, 3.7% ($n = 415$) of metformin initiators and 7.8% ($n = 154$) of sulfonylurea initiators died during follow-up (median: 5.0 years; total: 74,107 person-years; range of age at death: 57–104 years). Using a Cox proportional hazards (PH) regression model with inverse probability of treatment weighting (IPTW) to emulate randomization, the estimated hazard ratio for all-cause mortality was 0.57 (95% CI: [0.48;0.67]) for metformin initiators relative to sulfonylurea initiators (Fig. 2a). Next, we examined metformin's effects by age (≤70 vs. >70), sex, and BMI strata. Overall, there was no evidence for heterogeneous treatment effects across baseline age, sex, or BMI levels in the US RPDR cohort (Fig. 2b). Similar results were obtained using age strata defined as ≤65 vs. >65 and ≤75 vs. >75 (Extended Data Table 3).

In the larger UK CPRD cohort, 13.7% ($n = 12,941$) of metformin initiators and 37.4% ($n = 5173$) of sulfonylurea initiators died during follow-up (median: 6.0 years; total: 696,725 person-years; range of age at death: 51–107 years). The UK CPRD had similar results for the effect of metformin vs. sulfonylureas on all-cause mortality in the full study population (Fig. 2c), with an overall hazard ratio of 0.66 (95% CI: 0.61;0.71). Results of subgroup analyses revealed evidence for a stronger effect of metformin among patients with a younger age at treatment initiation (≤70 years), and patients with higher baseline BMI, but there was no difference by sex (Fig. 2d). The age-stratified analysis described above yielded similar results (Extended Data Table 4).

Our harmonized Drug Repurposing in Alzheimer's Disease (DRIAD)-EHR approach—with analyses conducted in two very different patient populations and carefully adjusted for baseline differences in age and other risk factors—demonstrates a robust reduction in the hazard of death in patients treated with metformin compared to those treated with sulfonylureas, consistent with previous reports[18,19]. In both cohorts, we note that the survival curves between the two treatment groups separate ~3 years after treatment initiation, and that this separation persists for a long time (12 years observed in US RPDR and 16 in UK CPRD).

### Metformin reduced the hazard of dementia onset in the US and UK cohorts compared to sulfonylureas, but the risk differences between the drugs over time were clinically negligible

Death is a competing event that precludes the development of dementia, but the use of a competing risks analysis in previous studies[8,17] has been limited: death has been considered as a competing event for dementia only in a proportional hazards model where the hazard ratio is the measure of treatment effect. Here, we emulated a target trial of metformin vs. sulfonylureas in cognitively asymptomatic type 2 diabetics, estimating *both* the time-invariant hazard ratio and the time-dependent cumulative incidence function (CIF) for dementia, using a causal competing risks framework. We defined the average

**Table 1 | Specification and emulation of a target trial of antidiabetic drug metformin vs. sulfonylureas on the risk of death and dementia, using observational data from Electronic Health Records of the US RPDR and the UK CPRD**

| Target trial specification | Emulation (US RPDR) | Emulation (UK CPRD) |
|---|---|---|
| **Eligibility criteria** | | |
| Age ≥ 50 | Same | |
| No hypoglycemics | No recorded prior exposure to any hypoglycemic agents | |
| No MCI*, dementia, or prescription of dementia drugs; normal cognitive testing | No recorded diagnosis of dementia or MCI*, or use of dementia-specific drugs (see Extended Data Tables 10–11) | No recorded diagnosis of dementia (MCI* diagnoses not available in CPRD) or use of dementia-specific drugs (see Extended Data Tables 12–13) |
| No chronic kidney disease (metformin contraindication) | No ICD*-9/10 code for chronic kidney disease or eGFR* <45 (Extended Data Table 1) | No diagnosis of chronic kidney disease at or prior to baseline (Extended Data Table 2) |
| Trial with 1-year run in period conducted for a specified duration with history obtained at baseline and ongoing monitoring of outcomes | • PCP* within Mass General Brigham Health Care system EHR* system<br>• At least one visit during the 18 months preceding baseline<br>• At least 1 year of follow-up<br>• No dementia or death in first year (1 year washout period) | • At least 1-year registration in CPRD practices before the first prescription<br>• At least 1 year of follow-up<br>• No dementia or death in first year (1-year washout period) |
| **Treatment strategies** | | |
| Treatment arm: metformin monotherapy Control arm: sulfonylurea monotherapy | Initiation of metformin or sulfonylurea from 1/2007-9/2017 (see Extended Data Fig. 8 for the number of new prescriptions per year) | Initiation of metformin or sulfonylurea from 1/2001-5/2017, with ≥2 monotherapy prescriptions for first 12 months (see Extended Data Fig. 9 for the number of new prescriptions per year) |
| **Treatment assignment** | | |
| Double-blind, randomized treatment assignment | Emulated randomization by balancing baseline confounders using IPTW* for treatment choice | |
| **Outcomes** | | |
| Diagnosis of MCI* or dementia | Diagnosis of MCI/Dementia by: ICD*-9/10 codes (Extended Data Table 10) OR at least one dementia-specific drug prescription (Extended Data Table 11) | Diagnosis of dementia by: Medcodes in CPRD or ICD*-9/10 codes in linked HES* or ONS* database (Extended Data Table 12) OR at least one dementia-specific drug prescription (Extended Data Table 13) |
| Time to death | Time to death recorded in EHR* | |
| **Follow-up** | | |
| From baseline and ends at dementia onset, death, lost to follow-up, or end of study | From the date of initial prescription of drug until the date of dementia incidence, death, last encounter date, 9/2018 (US RPDR) or 5/2018 (UK CPRD), whichever occurred first | |
| **Causal contrast** | | |
| Intention-to-treat effect | Observational analog of intention-to-treat effect | |
| **Statistical analysis** | | |
| Intention-to-treat analysis of primary outcomes (dementia and death) using Cox PH | Intention-to-treat analysis using Cox Proportional Hazards (PH) regression model and a competing risks framework accounting for death prior to dementia<br>Subgroup analyses by age, sex, and BMI* level at baseline | |

*BMI body mass index, eGFR estimated glomerular filtration rate, EHR Electronic Health Records, HES Hospital Episode Statistics, ICD International Classification of Diseases, IPTW inverse propensity score of treatment weighting, MCI mild cognitive impairment, ONS Office for National Statistics, PCP primary care physician.

treatment effect (ATE) as the difference between risk functions corresponding to two potential outcomes (for definitions, see "Methods").

In the US RPDR cohort, 7.7% (n = 869) of metformin initiators and 12.3% (n = 241) of sulfonylurea initiators were diagnosed with dementia during follow-up (median: 5.0 years; total: 71,191 person-years; range of dementia onset age: 57–113 years). In a cause-specific Cox PH regression model with IPTW for emulation of baseline randomization, the estimated cause-specific hazard ratio for dementia was 0.81 (95% CI: [0.69;0.94]) for metformin initiators relative to sulfonylurea initiators (Fig. 3a). In the UK CPRD cohort, 5.9% (n = 5561) of metformin initiators and 12.3% (n = 1699) of sulfonylurea initiators were diagnosed with dementia during follow-up (median: 6.0 years; total: 695,281 person-years; range of dementia onset age: 51–114 years). The estimated cause-specific hazard ratio for dementia was 0.86 (95% CI: [0.77;0.96]) for metformin initiators, relative to sulfonylurea initiators (Fig. 3b), very similar to the US RPDR cohort.

In the time-dependent CIF analysis, the 5-year risk (for definition, see "Methods") of developing dementia in the US RPDR cohort was 7.2% (95% CI: [6.7;7.8]%) among metformin initiators and 8.8% (95% CI: [7.5;10]%) among sulfonylurea initiators, yielding a risk difference (for definition, see "Methods") of −1.6% (95% CI: [−3.1;−0.17]%) (Fig. 4a, c). In the UK CPRD cohort, the 5-year risk difference was smaller at −0.35%

(95% CI: [−0.68;−0.031]%) (Fig. 4b, d). Although the hazard ratios for dementia in the UK CPRD and US RPDR were similar, the risk differences over time for both death and dementia were strikingly dissimilar in the two cohorts (Fig. 4).

First, while the dementia risk difference between metformin and sulfonylureas was minimal in the US RPDR, it always showed a slight benefit for metformin over sulfonylureas in this cohort. However, the risk difference observed in the UK CPRD changed over time, and the point of no risk difference between the two drugs was reached at about 7.5 years (Fig. 4). The seemingly discordant hazards ratio and CIF results in the UK CPRD sample are likely because metformin has a protective effect on *both* the hazard of dementia (HR = 0.86, 95% CI: [0.77;0.96]) and the hazard of competing death (HR = 0.64, 95% CI: [0.59;0.69]), yielding more "survivors" over time in the metformin group, and thus more individuals at risk of developing dementia.

Second, the risk differences for both death and dementia were much closer to each other in the US RPDR than in the UK CPRD cohort. These differences between the two cohorts could potentially be explained by different population structures, particularly their baseline age distribution (Extended Data Fig. 1). The UK CPRD cohort had a higher death rate than the US RPDR one, affecting the total number of patients at risk over time (Extended Data Fig. 2). The comparison of the

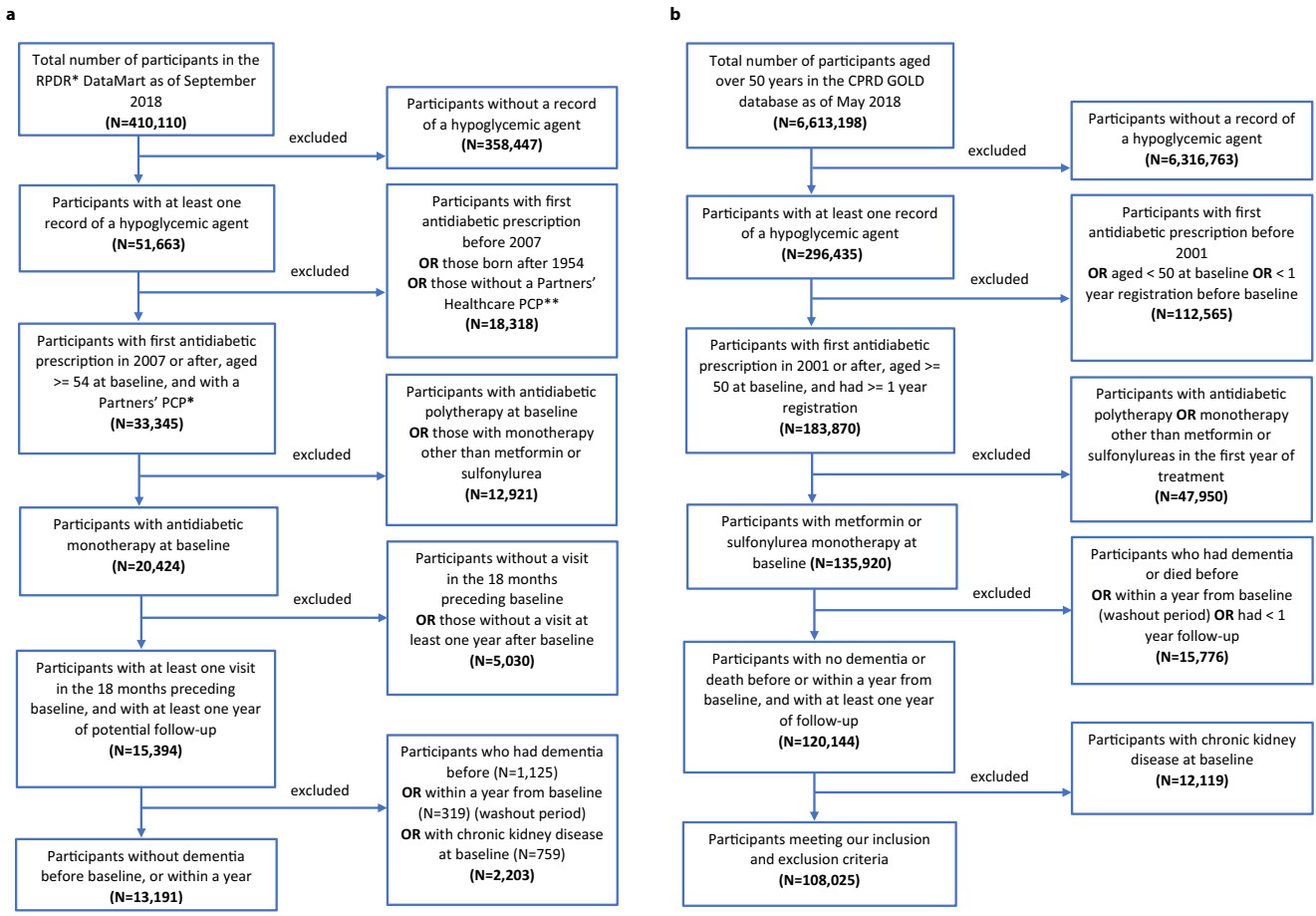

**Fig. 1 | Consort Diagrams for US RPDR and UK CPRD. a** Flowchart of inclusion and exclusion criteria of study population in RPDR. **b** Flowchart of inclusion and exclusion criteria of study population in CPRD.

absolute cumulative hazards of dementia and death in a competing risks approach revealed additional differences between the UK CPRD and US RPRD cohorts, in terms of both the magnitude and trajectory of diagnosed dementia (Extended Data Fig. 3). In the US RPDR cohort, the rates of diagnosed dementia in both treatment arms were higher than the death rates, whereas the opposite pattern was observed in the UK CPRD. These underlying differences likely explain the differing risk curves for death and dementia between the two cohorts.

Overall, the benefits of metformin observed here can be interpreted in terms of both delaying dementia onset (Fig. 4, blue curves) and prolonging life without dementia (Fig. 4, orange curves). Notably, the risk difference for death was of larger magnitude in the UK CPRD than in the US RPDR cohort (respectively, a 10% vs. 5% reduction in risk after an average of 12 years of follow-up, Fig. 4c, d). This can be interpreted as the result of a higher overall death rate in the UK CPRD cohort, relative to the rate of dementia onset.

To assess the robustness of our results to modeling choices, we conducted a sensitivity analysis by using a nonparametric approach, thereby relaxing the proportional hazards assumption. Of note, the PH assumption held for both cause-specific hazards in the US RPDR cohort, while there was evidence of a deviation from this assumption in the UK CPRD cohort in terms of the hazards of death. However, this deviation was not large enough to affect our conclusions (Extended Data Figs. 4, 5).

### Average treatment effect on dementia onset was greater among patients aged ≤70 in the US RPDR

Since age is the principal risk factor for dementia, we further investigated the effect modification of metformin as compared to sulfonylureas by the age at treatment initiation. Stratifying the US RPDR

cohort into two groups (age ≤ 70 and age >70), we found that the ATE of metformin vs. sulfonylureas on dementia onset observed in the full sample was mainly driven by the younger stratum (Fig. 3a), i.e., treatment initiation at age ≤70 (HR = 0.69, 95% CI: [0.54;0.88]). Conversely, the effect of metformin on dementia onset was reduced for patients who started antidiabetic treatment at age >70 (HR = 0.94, 95% CI: [0.79;1.13]). However, there were fewer patients who started antidiabetic treatment at age >70 than earlier (38% vs. 62%) and the older stratum had a shorter length of follow-up (median: 4.1 vs. 5.6 years; total: 22,960 vs. 48,231 person-years). Nevertheless, the age-specific finding in the US RPDR cohort suggests that metformin may be especially beneficial−relative to sulfonylureas−for those who initiate treatment at a younger age.

The difference in treatment effect between age groups was less clear in the larger UK CPRD cohort, with a HR of 0.82 (95% CI: [0.67;0.99]) in patients aged ≤70, and of 0.88 (95% CI: [0.77;0.99]) in those aged >70 (Fig. 3b). Similar results were obtained using the risk difference: a stronger effect of metformin on dementia onset was observed in the US RPDR cohort, as compared to the UK CPRD, in patients who initiated treatment before age 70 (Extended Data Fig. 6).

### Difference in post-treatment HbA1C levels was not clinically significant for metformin vs. sulfonylurea initiators

Since baseline HbA1C levels did not modify the effect of metformin, we also explored whether the drug acted primarily by a better control of blood sugar. For this, we applied a repeated measures mixed effects model on all HbA1C values recorded three months after treatment initiation and beyond. In the US RPDR cohort, 10,180 (77%) patients had HbA1C data available: 8794 (78%) and 1386 (71%) among

**Table 2 | Characteristics of eligible individuals when emulating a target trial of metformin vs. sulfonylurea initiators on the risk of death and dementia in the US RPDR (2007–2017) and the UK CPRD (2001–2017)**

| Patient characteristics | US RPDR cohort (N = 13,191) | | UK CPRD cohort (N = 108,025) | |
|---|---|---|---|---|
| | Metformin initiators (n = 11,229) | Sulfonylurea initiators (n = 1962) | Metformin initiators (n = 94,208) | Sulfonylurea initiators (n = 13,817) |
| Age at baseline (mean, years) | 68.6 | 72.2 | 64.9 | 70.2 |
| Sex (% male) | 49.2 | 52.9 | 57.4 | 58.3 |
| Race (percent) | | | | |
| White | 78.2 | 83.4 | 86.5 | 87.0 |
| Black or African American | 7.6 | 5.4 | 3.6 | 3.0 |
| Asian | 4.4 | 3.7 | Data combined with Other | |
| Other | 9.8 | 7.5 | 9.9 | 10.0 |
| Ethnicity (% Hispanic) | 4.4 | 3.3 | Data not available | |
| Year of first prescription (median) | 2012 | 2012 | 2008 | 2003 |
| SES/IMD[a] (% low income/most deprived) | 3.1 | 1.8 | 17.3 | 17.5 |
| Cancer (%) | 30.2 | 28.7 | 9.9 | 12.4 |
| CVD[b] (%) | 42.8 | 44.4 | 53.8 | 50.8 |
| Hypertension (%) | 74.2 | 67.0 | 94.9 | 90.7 |
| COPD (%) | Data not available | | 4.7 | 4.6 |
| Smoker[a] (%) | | | 17.0 | 18.7 |
| Stroke[b] (%) | 11.9 | 12.1 | Data combined with CVD | |
| Baseline HbA1C level[a] (percent) | | | | |
| <7 (%) | 51.8 | 46.2 | 16.4 | 11.8 |
| 7–10 (%) | 42.0 | 47.9 | 69.1 | 65.3 |
| >10 (%) | 6.2 | 5.9 | 14.5 | 22.9 |
| Baseline BMI[a] (kg/m²) | | | | |
| <25 (%) | 11.1 | 17.5 | 8.1 | 34.8 |
| 25–30 (%) | 29.5 | 36.0 | 32.9 | 43.5 |
| ≥30 (%) | 59.4 | 46.5 | 59.0 | 21.7 |
| All-cause mortality analysis outcomes | | | | |
| Median follow-up time (years) | 5.3 | 5.3 | 6.0 | 7.0 |
| Total person-years | 63,060 | 11,047 | 592,948 | 103,777 |
| Deaths (n, %) | 527 (4.7) | 222 (11.3) | 12,941 (13.7) | 5173 (37.4) |
| Competing risk analysis outcomes | | | | |
| Median follow-up time (years) | 5.0 | 5.0 | 6.0 | 7.0 |
| Total person-years | 60,683 | 10,508 | 592,072 | 103,209 |
| Deaths prior to dementia (n, %) | 415 (3.7) | 154 (7.8) | 11,560 (12.3) | 4,570 (33.1) |
| Incident dementia cases (n, %) | 869 (7.7) | 241 (12.3) | 5561 (5.9) | 1699 (12.3) |

*BMI* body mass index, *COPD* chronic obstructive pulmonary disease, *CVD* cardio-vascular disease, *HbA1C* glycosylated hemoglobin. *IMD* Index of Multiple Deprivation (official measure of relative deprivation by small geographic region in the UK).

[a]SES (US RPDR) and IMD (UK CPRD) are distinct, country-specific socioeconomic indicators, intended for comparison within—rather than across—the two cohorts. IMD, smoking status, HbA1C, and BMI had 7%, 2%, 21%, and 3% of missing values in the UK CPRD, respectively. The corresponding statistics presented in the table are valid percentages out of patients without missing information.

[b]Stroke and CVD indicators were collapsed.

metformin and sulfonylurea initiators, respectively. Interestingly, we found that although the average level of HbA1C was lower ($p < 0.00001$) in metformin vs. sulfonylurea initiators, the effect size was not clinically significant (−0.2056; 95% CI: [−0.2601;−0.1511], see Extended Data Table 5). This suggests that the putative effect of metformin on dementia risk is likely through mechanisms other than the control of blood sugar.

## Metformin reduced the expression of innate immune modulators and APOE levels in cultured human neural cells

The actions of metformin in human neurons have not been characterized well, despite pharmacokinetic evidence that metformin achieves biologically active concentrations in the CSF[20]. For 24 and 72 h, we treated cultured differentiated human neural cells comprised of neurons, glia, and oligodendrocytes[21] with metformin and glyburide (one of the sulfonylureas). We used two biologically relevant concentrations, 10 and 40 μM, which approximate CSF and plasma concentrations[20], respectively (Fig. 5a). Following deep RNA-sequencing, we identified differentially expressed genes that were significantly altered in a dose-dependent manner (Fig. 5b, Extended Data Fig. 7). After treatment exposure, genes with the largest change were different between the two drugs, with greater effect sizes seen for glyburide, relative to metformin, at 72 h. Pathway analysis revealed significant differences in metformin-altered genes were enriched in pathways related to the extracellular matrix, whereas glyburide-altered genes were enriched in pathways related to cholesterol metabolism (Extended Data Table 8).

Next, we limited the analysis to elements of the human secretome, since they are measurable in the CSF[22] (Fig. 5c, Extended Data Table 9). To reflect subacute drug-induced profiles, we considered the 72-h timeframe. Osteopontin (SPP1) emerged as the secreted protein with the greatest reduction in RNA levels at 72 h and the greatest change overall. Previous work reported elevated levels of osteopontin in the serum of aging individuals[23] and in the CSF of AD patients, correlating with cognitive decline[23]. Moreover, elevated levels of SPP1 in microglia were detected in AD mouse models and human brains[24]. By looking at gene expression profiles of postmortem brain specimens in the ROSMAP[25] and Mt. Sinai brain bank[26] cohorts, we observed that SPP1 levels were elevated in tissue from the frontal and temporal lobes of patients with AD, relative to age-matched controls (Fig. 5d). Finally, we observed that RNA levels of APOE in cultured human neural cells were significantly reduced by metformin treatment (Extended Data Table 8). Together, the reduced gene expression of an innate immune modulator (SPP1) and a genetically implicated protein (ApoE) by metformin in human neural cells suggest candidate CSF biomarkers, which if validated, may associate with the delayed onset of clinical symptoms of dementia in type 2-diabetic patients.

## Discussion

Previous observational studies of metformin's action to reduce dementia onset have had mixed results, but these studies did not account for death as a competing event. In this study, we emulated target trials of metformin vs. sulfonylureas in type 2-diabetic patients in two distinct EHRs. The target trial emulation methodology is a recently developed approach to quantify the actions of metformin on dementia onset in incident type 2-diabetic patients. This work implements a rigorous causal framework harmonized across two EHR databases, in incident type 2-diabetic patients. We found that treatment initiation with metformin, as opposed to sulfonylureas, robustly reduces the risk of dementia onset and death among type 2-diabetic patients in two different EHR databases. The beneficial effect of metformin over sulfonylureas is unlikely due to better control of hyperglycemia, prompting us to investigate alternative modes of action for metformin's beneficial effect in an in vitro human differentiated neural

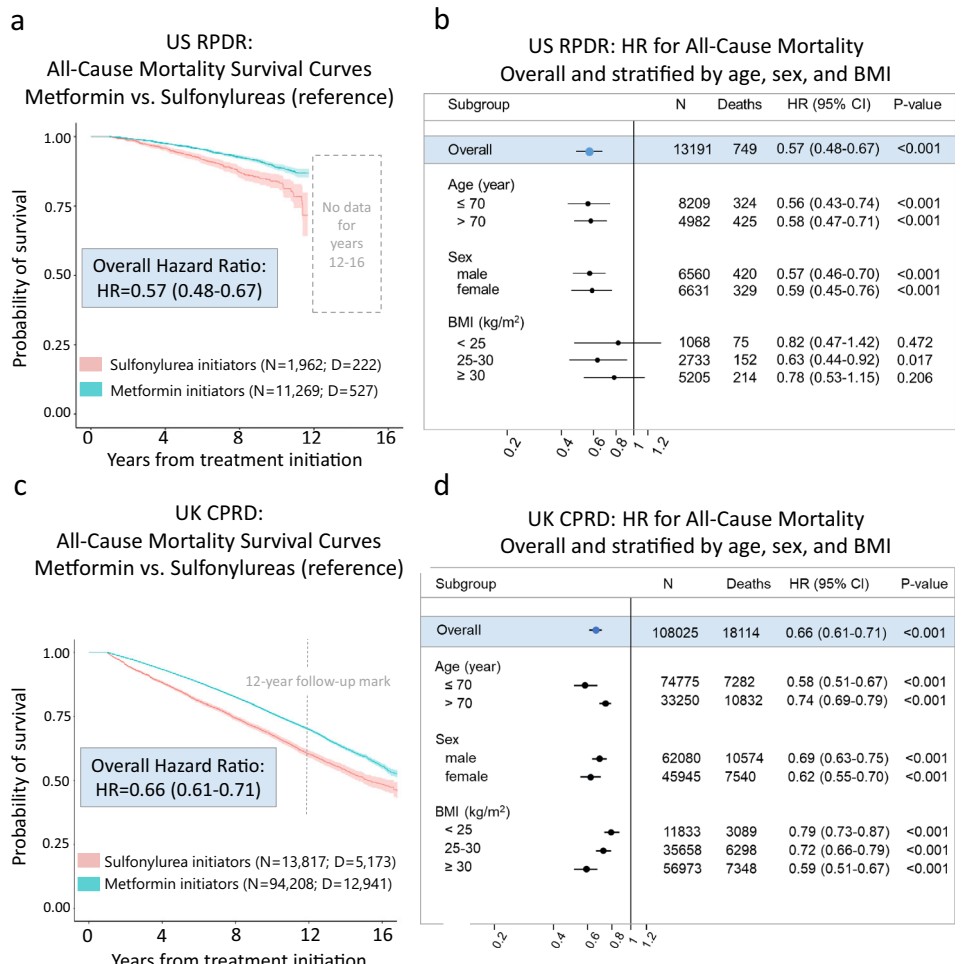

**Fig. 2 | Metformin reduces all-cause mortality relative to sulfonylureas among type 2-diabetic patients aged >50 years at treatment initiation.**
**a**, **c** Kaplan–Meier survival curves for metformin and sulfonylurea initiators with shaded areas representing 95% Confidence Intervals (CI), based on pointwise 0.025- and 0.975-quantiles of sample bootstrap distributions. N = # of patients at baseline, D = # of deaths during follow-up. Hazard ratios (HR) were estimated using the Cox Proportional Hazards (PH) model, with only treatment as a covariate, and baseline covariate distributions between treatment arms balanced by Inverse Propensity score of Treatment Weighting (IPTW). **b**, **d** Forest plots presenting all-cause mortality HRs overall and stratified by age, sex, and BMI level at baseline, with sulfonylurea initiators as the reference group. Covariate balancing using IPTW was conducted in each stratum independently. Error bars represent 95% CIs for hazard ratios. A two-sided Wald test of whether the hazard ratio associated with metformin treatment initiation is 1, with robust variance estimator, was used. No further correction for multiple hypothesis testing was applied.

cell system—in contrast to glyburide, a sulfonylurea. The gene expression of SPP1 and *APOE*, two gene products associated with AD pathology, were both uniquely and significantly reduced by exposure to metformin, a drug that penetrates the blood-brain barrier, at pharmacologically relevant drug concentrations.

One advantage of target trial observational studies is much longer follow-up periods after drug initiation than is feasible in RCTs. This increased length of follow-up is of special importance, since many dementia risk factors likely operate over a long period of time[27,28] and since dementia onset progression is an infrequent outcome. After balancing key baseline demographic variables in the metformin and sulfonylurea cohorts by IPTW, we conducted an intention-to-treat analysis with a 1-year run-in period in two patient populations drawn from vastly diverse settings. One cohort was from a healthcare system anchored in two large tertiary care hospitals in the US and another from a nation-wide primary care network in the UK. Despite differences in medical practice, data collection, timing and length of follow-up, patterns of missingness, and known and unknown sources of bias, we found consistent evidence of metformin's benefit for overall survival and for dementia onset, relative to sulfonylureas. While prior studies only estimated a cause-specific hazard ratio for the impact of

treatment on risk of dementia, in the CIFs from our competing risks framework we were able to account for the treatment effect on both outcomes jointly. Relative to sulfonylurea initiators, metformin initiators had a reduced hazard of dementia onset in both cohorts. Our results corroborate the benefits of metformin on dementia risk in type 2 diabetics reported in previous observational studies[10,11,18,19,29,30]. Moreover, our competing risks analysis demonstrates how the risk of dementia depends on the baseline mortality rate of the population, a potential explanation of the neutral[10,29] or deleterious[11] effect of metformin on dementia onset seen in other observational studies (Extended Data Table 14). To our knowledge, this work offers a unique approach to comprehensively address competing death in a study of metformin and dementia, with a rigorous causal framework harmonized across two EHR databases.

The risk difference offers a nuanced view over time that is not captured by the time-invariant HR metric, which was similar across the two cohorts. An additional value of this observational study for planning future clinical trials is that it explores the source of the signal in subpopulations defined by criteria that could readily be implemented as inclusion and exclusion criteria. Our age-stratified analysis indeed demonstrated that type 2-diabetic patients aged 70 or younger at

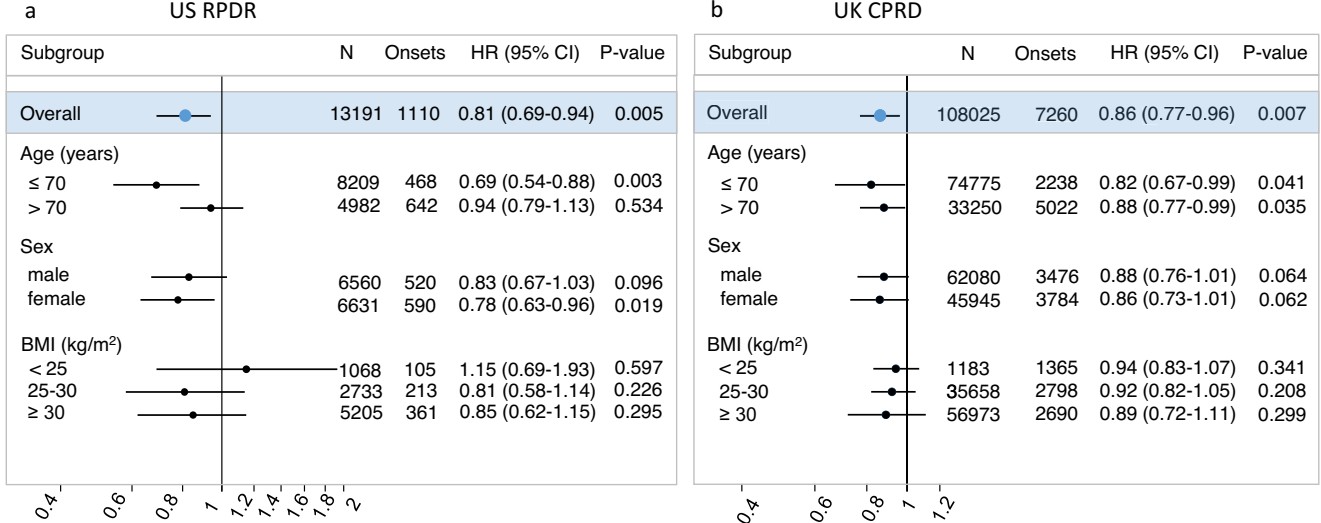

**Fig. 3 | Metformin reduces incident dementia relative to sulfonylureas in type 2-diabetic patients aged >50 years at treatment initiation.** HRs were estimated using the Cox PH model for the cause-specific hazards of dementia, with only treatment as a covariate and baseline covariate distributions between treatment arms balanced by IPTW. **a, b** Forest plots present HRs overall and stratified by age, sex, and BMI level at baseline, with sulfonylureas as the reference group. N = # of patients at baseline, Onsets = # of patients with dementia onset during follow-up and prior to death. Covariate balancing using IPTW was conducted in each stratum independently. Error bars represent 95% CIs for hazard ratios. A two-sided Wald test of whether the hazard ratio associated with metformin treatment initiation is 1, with robust variance estimator, was used. No further correction for multiple hypothesis testing was applied.

treatment initiation benefited most from metformin's effect on cognitive health.

Further, our analysis of HbA1C over time in the US RPDR showed a clinically negligible benefit among metformin vs. sulfonylurea initiators, indicating that the mechanism for metformin's effects on dementia onset and survival is unlikely to be simply a byproduct of better diabetic control. Moreover, our in vitro systems pharmacology analysis in cultured human neural cells treated with metformin and glyburide identified over 100 differentially expressed genes, particularly affecting signaling networks implicated in aging. The neural cells were not derived from a diabetic patient and were not grown under hyperglycemic conditions, supporting the notion that the observed changes may occur in the CNS of non-diabetic patients. The secreted protein SPP1 emerges as a candidate CSF biomarker for metformin's action in the central nervous system (CNS) to be further investigated as an exploratory aim before and after metformin exposure. SPP1 was elevated in the CSF of MCI and mild AD patients[22,23], in the autopsied brains of AD patients[31,32], and in the plasma of AD patients[33]. Further, greater levels of SPP1 correlated with cognitive decline in these patients[22,23]. SPP1 is also elevated in response to TREM2 activation in microglia[34] in brains with AD pathology[24,35]. Lowering SPP1 levels in the CNS may thus be a unique mechanism of neuroprotection by metformin. Clinical investigators may consider adding elevated levels of SPP1 in the CSF as inclusion criteria for a clinical trial of metformin in subjects with preclinical AD biomarkers.

There are many strengths in our DRIAD-EHR approach. First, in this observational study, our two samples were followed for up to 16 years. Given that the preclinical stages of dementia can last 20 years[36], this study is examining a therapeutically relevant timeframe which is not feasible in randomized clinical trials. Second, we harmonized our analyses in two distinct EHR databases. The concordance of the hazard ratio estimates for both the survival and dementia outcomes across these two distinct patient populations indicates robust signals[4,5]. Third, we developed and implemented a causal competing risks

framework, to account for death prior to developing dementia. By analyzing the cumulative incidence of death and dementia in parallel in both cohorts, we found that the mortality rate within a given population could have a significant impact on the cumulative risk of dementia, suggesting that a 3–5 years mortality index should be included as a criterion in clinical trials evaluating the efficacy of metformin to prevent the onset of dementia. Fourth, in complementary mechanistic studies, we analyzed gene expression changes in relevant human neural cell types at drug concentrations commensurate with observed levels in the plasma and CSF.

Nevertheless, this study has several limitations. First, while we addressed many sources of confounding, there were likely others that were unavailable or inadequately measured. In particular, the level of education was systematically unavailable in either dataset, of concern since it is known to affect both the exposure and outcomes of interest in this study. In addition, relevant lifestyle factors, like diet and physical activity[37] and a genetic risk factor, ApoE genotypes, were unavailable. Furthermore, the strong effect of age, the changes in prescribing patterns of sulfonylureas and metformin over the observation period[38,39], gene-environment interactions, and the complex differences observed in age at baseline, length of follow-up, and calendar time across the two treatments raises the possibility of residual confounding. Beyond this, in EHR, data missingness is very often informative and can lead to biases in study results[40–42]. Second, the absence of linkage to claims data in the US RPDR cohort prevented us from verifying that patients were truly treatment *initiators*, or from verifying the length of exposure by confirming that prescriptions were filled and refilled at the expected rate. Third, since this study was an intention-to-treat analysis, it did not include potential add-on drugs incorporated later in the patient's clinical course or consider anti-diabetic treatment switches. Thus, transition from monotherapy to a dual (or more) hypoglycemic regimen could be a possible source of confounding for both dementia and death outcomes. Fourth, this study might suffer from measurement errors in the primary outcome

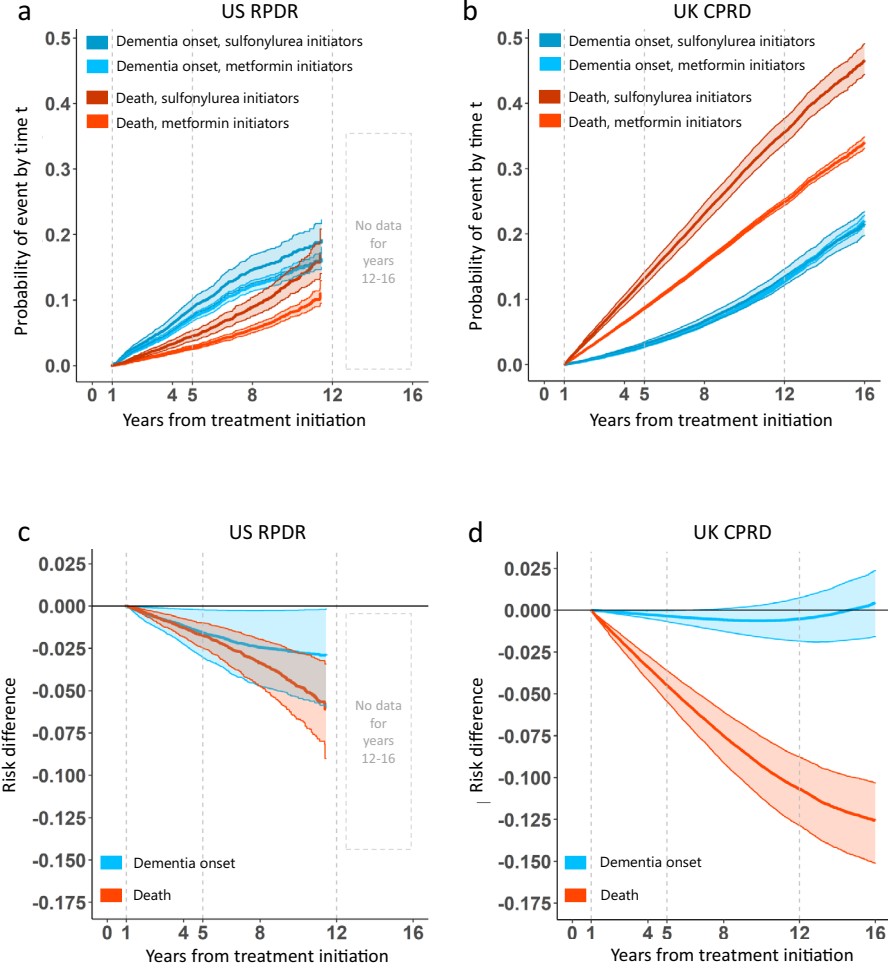

**Fig. 4 | Risk differences in dementia onset over time are negligible for metformin initiators, as compared to sulfonylureas.** Cumulative incidence functions (CIF) or risk curves, along with their 95% CIs (represented by shaded areas), based on pointwise 0.025- and 0.975-quantiles of sample bootstrap distributions, were estimated using the Cox model for the cause-specific hazards, with only treatment as a covariate and baseline covariate distributions between treatment arms balanced by IPTW. **a**, **b** CIF curves for dementia onset (in blue hues) and competing death (in orange hues) for metformin vs. sulfonylurea initiators. Follow-up times are up to 12 and 16 years in the US RPDR (**a**) and the UK CPRD (**b**) cohorts, respectively. **c**, **d** Risk difference curves for dementia onset (in blue) and competing death (in orange), in the US RPDR (**c**) and the UK CPRD (**d**) cohorts, respectively. A negative risk difference value during certain time periods indicates that initiation of metformin is beneficial, as compared to sulfonylureas.

of interest[43]. Dementia is under-diagnosed and under-recorded, both in the US[44] and UK[45,46]. Patients and their families may fail to mention symptoms to their primary care physicians and physicians might not routinely screen for cognitive health. Even when such symptoms are recognized and described directly or indirectly in physicians' notes, the relevant diagnostic code or prescription used here as a proxy for the disease might not be present ever—or may not appear until late in the course. In other cases, dementia might be overcoded[47,48]. Hence, in the future, we aim to deploy text mining and natural language processing techniques on clinical notes, radiologic image analysis, and other clinical data to better identify subjects with dementia and more precisely determine the timing of disease onset. Fifth, while our study was conducted in two different populations, both are primarily white and have access to health care. While the UK CPRD population is fairly representative of the UK population, the US RPDR population is limited to a single region, less diverse, and more advantaged than the US population as a whole. Sixth, in our mechanistic studies, we approximated chronic exposure to metformin and glyburide through relatively short durations, in cultured human neural cells that did not include all the cell types in the brain, including microglia. The candidate pharmacodynamic biomarkers for metformin's actions in the

brain (SPP1 and APOE) will need to be validated in the CSF of patients taking metformin or a sulfonylurea.

Applying a causal competing risks framework to estimate the effects of metformin compared to sulfonylureas among type 2-diabetic patients showed consistent findings across two disparate datasets. These robust EHR findings were buttressed by in vitro analyses of human neural cells at pharmacologically relevant concentrations that revealed neural cell-specific actions of metformin, relative to glyburide, and to metformin's actions in other cell types. Together, this multi-dimensional DRIAD-EHR approach uncovered four pragmatic insights that could be incorporated into clinical trial design: first, an estimate of efficacy among asymptomatic individuals that is validated in two distinct databases; second, the importance of treating mortality as a competing risk for clinical trials that last 3 to 5 years; third, enrolling people aged less than 70 (or perhaps less than 75 in non-diabetics, among whom the onset of dementia is, on average, later); and fourth, our in vitro systems pharmacological studies of metformin and glyburide in cultured human neural cells identified a candidate gene, SPP1/osteopontin, whose expression is reduced significantly more by metformin than by sulfonylureas. Further, the gene product is elevated in the CSF and plasma of patients diagnosed with AD or pre-

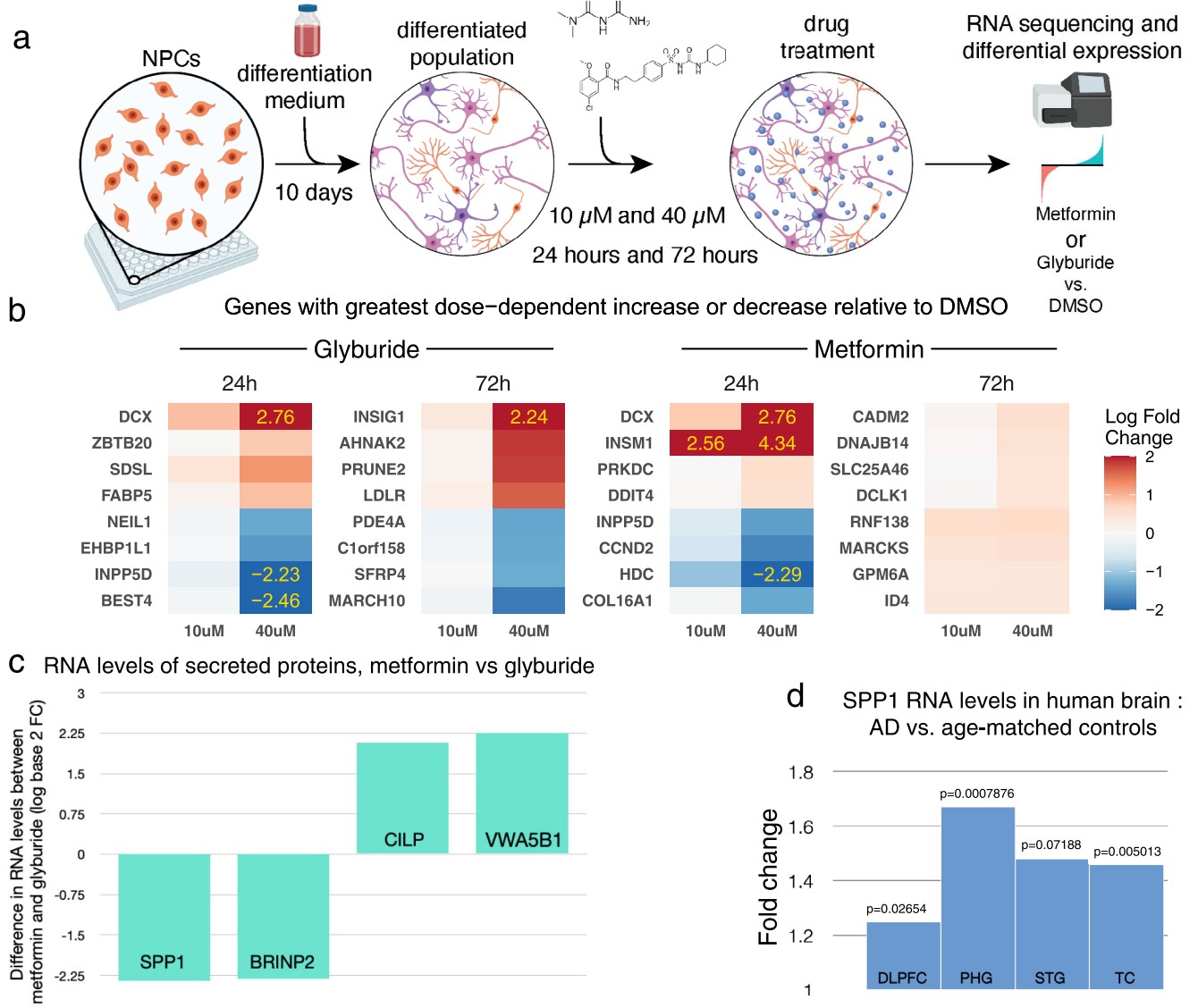

**Fig. 5 | Differential gene expression in human neural cells triggered by metformin and glyburide: markedly reduced levels of the AD biomarker SPP1 (osteopontin). a** Differentiated human ReN VM cells into neural cells were treated with metformin or glyburide at two different concentrations, for either 24 or 72 h. **b** Genes with largest dose-dependent change in expression over 24 and 72 h, for either metformin or glyburide—top 4 increased and top 4 decreased. **c** Genes expressing secreted proteins with greatest differential between metformin and glyburide. **d** SPP1 RNA levels in 4 different brain regions: AD vs. Controls. DLPFC dorsolateral prefrontal cortex, PHG parahippocampal gyrus, STS superior temporal sulcus, TC temporal cortex.

MCI. Future clinical trials of metformin in cognitively intact non-diabetics will determine whether anti-aging actions of metformin beyond hyperglycemic control could be an important component of strategies to prevent dementia.

## Methods

### Data sources

The data used in the study are from two large-scale EHR databases: the Research Patient Data Registry in the United States at Mass General Brigham Healthcare (US RPDR) and the Clinical Practice Research Datalink in the United Kingdom (UK CPRD).

**US RPDR.** The study cohort was selected from the RPDR registry[13]. RPDR is a longitudinal centralized clinical data registry with ~6.5 million patients mainly from the Boston area as of June 2020. The data are collected from EHR systems within Mass General Brigham (MGB)

Healthcare (formerly Partners Healthcare), comprising two major academic hospitals, as well as several community hospitals and community health centers in the Boston area. Death records are updated periodically from the US Social Security Death Index (2007–2017). Use of RPDR data for this study was approved by the institutional review board (IRB) of MGB.

**UK CPRD.** The study cohort was selected from the CPRD database[14]. CPRD is a longitudinal national primary care database, with ~17 million patients from 13 regions across the UK. The data are collected from EHR systems in General Practitioner (GP) practices. GP practices can enroll the CPRD data system on an ongoing basis and can leave it at any time. Over 700 GP practices (8% of total GP practices) have contributed data to CPRD and the mean follow-up time for patients included in CPRD is around 8 years. Death records are updated periodically from the UK Office for National Statistics (ONS). In addition,

data linkages were established with secondary care data from Hospital Episode Statistics (HES) as well as with small-area measures of social deprivation (2001–2017). Use of CPRD data for this study was approved by the Independent Scientific Advisory Committee (ISAC) for Medicines and Healthcare products Regulatory Agency (MHRA) database research (protocol number: 19_065R).

Both the US RPDR and the UK CPRD data includes patient demographics; encounter details such as dates, providers, diagnoses, and procedures; medical notes, drug prescriptions, and laboratory test results. We reported our findings according to the RECORD reporting guidelines[49].

## Study population
The eligibility criteria satisfied by the US RPDR and the UK CPRD populations are:

| Criterion | US RPDR | UK CPRD |
|---|---|---|
| (a) Time period spanned by the observational study | For the emulated trial in the RPDR database, we included patients with initial prescriptions of metformin or sulfonylureas between January 2007 and September 2017. | For the emulated trial in the CPRD database, we included patients with initial prescriptions of metformin or sulfonylureas between January 2001 and May 2017. |
| (b) Minimum age at baseline | We included patients over 50 years old at the first prescription date (time 0, or baseline). | |
| (c) Prior history of primary care within the system, in order to: - allow sufficient time for baseline patient information to be recorded and reduce the likelihood of data missingness - ensure the patient is a new antidiabetic drug user at baseline and maximize the duration of their follow-up | Have a primary care physician (PCP) within MGB before the first prescription of antidiabetic drugs. To identify patients with a MGB PCP, the presence of at least one of the following was required: (i) a CPT code for preventative medicine services, (ii) an annual exams/wellness visit in the EHR, or (iii) an encounter from selected departments (family medicine, general practice, general internal medicine, or preventative care). | Have at least 1-year registration in a CPRD practice before the first prescription of antidiabetic drugs. |
| (d) Metformin- and sulfonylurea-monotherapy assignment | The assignment of patients to the metformin- and the sulfonylurea-monotherapy group was based on the first record of prescription of either drug. Only one prescription was required. Prescriptions of antidiabetic medications was obtained from RPDR (see Methods, "Exposure assessment"). The metformin monotherapy group included patients who were prescribed at baseline only metformin; the sulfonylurea monotherapy group included patients who at baseline were prescribed only sulfonylureas. | The assignment of patients to the metformin- and the sulfonylurea-monotherapy group was based on the first record of prescription of either drug. At least two consistent prescriptions during the initial 12-month treatment period were required. Prescriptions of antidiabetic medications were obtained from CPRD using British National Formulary codes (see Methods, 'Exposure assessment'). The metformin monotherapy group included patients who were exposed only to metformin during their initial 12-month treatment period; the sulfonylurea monotherapy group included |
| | | patients who were exposed only to sulfonylureas during their initial 12-month treatment period. |
| (e) Absence of dementia diagnosis at baseline | Have no dementia diagnosis or dementia-specific drug prescription before the baseline date (see "Dementia outcome ascertainment"). | |
| (f) Over 1 year of follow-up | Have 1-year of follow-up after treatment initiation. Have no dementia or death record during the first year of follow-up. | |
| (g) Absence of chronic kidney disease (CKD) diagnosis at baseline | Have no CKD diagnosis at the time of metformin- or sulfonylurea-monotherapy treatment initiation, since CKD is a contraindication for metformin[77]. | |
| Resulting sample size | A total of 13,191 patients in RPDR met these eligibility criteria and were included in the analyses (see Fig. 1a for the US RPDR consort diagram). | A total of 108,025 eligible patients in CPRD met these eligibility criteria and were included in the analyses (see Fig. 1b for the UK CPRD consort diagram). |

## Exposure assessment
In both cohorts, any patients in combination therapy at baseline, including with insulin, were excluded. We first identified individuals who met the eligibility criteria and assigned them to the treatment indicated in their medical record at baseline. The sulfonylurea monotherapy group included patients who were exposed only to sulfonylureas, including first generation (tolbutamide, chlorpropamide, tolazamide, or acetohexamide) and second generation (gliclazide, glibenclamide, glipizide, glimepiride, gliquidone, glibornuride, or glymidine sodium). Similarly, the metformin monotherapy group included those who were exposed only to metformin (Extended Data Figs. 8–9).

## Dementia outcome ascertainment
In both the US RPDR and the UK CPRD cohorts, the date of dementia onset was defined as the first dementia diagnosis date or the first prescription date of dementia-specific drugs, whichever occurred earlier.

**US RPDR.** In RPDR, dementia incidence was defined by the presence of either one or several dementia diagnosis codes (expertly curated list of International Classification of Diseases (ICD) codes including: (a) ICD10 codes: 290.X, 294.X, and 331.X; (b) ICD9 codes: 780.93, G30.X, and G31.X (Extended Data Table 10), and/or by the initiation of drugs primarily used for dementia (Donepezil, Galantamine, Rivastigmine, and their respective brand names Aricept, Razadyne, Exelon (Extended Data Table 11)).

**UK CPRD.** In CPRD, dementia incidence was defined by the presence of either one or several dementia diagnosis codes (expertly curated list including: (a) a selected set of CPRD Medcodes: see Extended Data Table 12 for the detailed code list; (b) ICD 9/10 codes in linked HES or ONS databases: see Extended Data Table 13 for the detailed code list), and/or by the initiation of drugs primarily used for dementia (Donepezil, Galantamine, Rivastigmine, or Memantine (Extended Data Table 13)).

We performed an intention-to-treat analysis. We were interested in assessing the comparative effectiveness of metformin- vs. sulfonylurea-monotherapy on time-to-dementia-onset in the presence of competing death, in the population of people who survived a year post baseline without dementia.

## Covariates
**Confounder selection.** We considered all available covariates that potentially influence both the treatment assignment and one or both outcomes, or strongly related to at least one of the outcomes: dementia or death[50,51]. To harmonize the confounders included in the

dementia onset and death model, we considered covariates that were influencing either outcome. Solely data-driven approaches to covariate selection can negatively affect the precision of the estimates and even amplify the residual bias[52,53], and are a particular concern in EHR research, where informative missingness is the rule[41,42]. When adjusting for sources of confounding, both VanderWeele[50] and Brookhart et al.[52] recommend including covariates that are weakly related to the treatment assignment but are strongly related to the outcome of interest.

Specifically, we included the following covariates. Age is the largest risk factor for dementia[54] and death, and hence was included as a covariate. Since the UK CPRD observational study was spanning a longer time period (2001–2017) than the US RPDR (2007–2017), the calendar year of the first prescription was added to control for temporal changes in prescribing practices, mortality trends, and age-specific incidences of dementia[11]. Given that the US RPDR cohort captures patients who initiated after the new antidiabetic treatment recommendations formulated in 2006, there were no specific concerns about a potential shift in prescription patterns. It is unknown whether sex affects antidiabetic treatment assignment, but it strongly relates to the death outcome and may affect the dementia outcome beyond its effect on surviva[43,55,56]. In addition, there might be sex-based differences in disease detection and reporting in the medical records, resulting in women having a higher incidence of dementia than men, as documented in the EHR. Therefore, we included both age at baseline (as a continuous covariate) and sex in the model. Hypertension, cardiovascular diseases (CVD), and stroke are also associated with dementia, and hence were also considered[57]. We included body mass index (BMI) at treatment initiation, and baseline levels of HbA1C, which measures the average blood sugar levels over a period of about three months and the severity of diabetes. Socioeconomic status (SES) is associated with both dementia and death outcomes[58]. Finally, cancer is associated with death. For a patient to be categorized as having a history of cancer at baseline, we required at least two instances of cancer ICD codes in the EHR. The choice was based on previous literature, which suggested that accuracy is highest with two instances of cancer ICD codes[59].

In both cohorts, information on the following covariates before the baseline date was extracted: age at the first prescription, sex, SES (index of multiple deprivation (IMD) in the UK CPRD and median annual household income by zip code in the US RPDR), BMI (<25, 25–30, ≥30 kg/m$^2$, or missing), HbA1C (<7%, 7–10%, >10%, or missing), and comorbidities (hypertension, CVD, stroke, and cancer). Additional covariates which were not available in the US RPDR dataset were extracted in the UK CPRD cohort, including smoking status (non-smoker, current smoker, ex-smoker, or missing), and presence of chronic obstructive pulmonary disease (COPD) before baseline. Whereas the US RPDR cohort mainly includes patients living in the Boston area, the UK CPRD cohort is representative of patients nationally. To adjust for the geographical heterogeneity, the region of residence was additionally incorporated in the UK CPRD (as a categorical covariate with 12 levels and the reference).

**Emulation of baseline randomization.** The covariates defined above were used to emulate baseline randomization. We adjusted for confounding by rebalancing the metformin- and sulfonylurea-treatment groups, using Inverse Propensity score Treatment Weighting (IPTW). For both our analyses, all-cause mortality and competing risks, we used the same IPTW approach and the same set of confounders[60]. The contribution of each participant was reweighted to achieve balanced treatment arms with respect to a set of measured confounders.

In the US RPDR, we chose not to include two covariates, SES and history of stroke. In the UK CPRD, the IMD covariate was included, and the history of stroke was relatively rare (7%) and was combined with CVD in a single covariate for simplicity. In the US RPDR study population, there was essentially no variability in the SES variable, as 96% of

the cohort had a family income greater than the US poverty threshold, and there was almost no difference between the treatment groups (96 and 97% among metformin and sulfonylurea initiators). For stroke, there was somewhat more variability—12% of the US RPDR cohort had an indication of prior stroke at treatment initiation, but there was almost no difference between the treatment groups (11.9% of metformin and 12.1% of sulfonylurea initiators).

Let $A$ be the treatment assignment random variable with $A = 1$ for metformin and $A = 0$ for sulfonylureas. Let $C$ be a set of confounders. We estimated propensity scores defined as follows: $ps_i = P(C = c_i)$ for individual $i$ with treatment $A = a_i$ and covariates $C = c_i$, by fitting a logistic regression model. We denote the estimates of $ps_i$ by $\widehat{ps_i}$. Subject-specific weights were obtained by $\widehat{w_i^*} = 1/\widehat{ps_i}$, i.e., by inverse-probability of being assigned to the actual treatment $A = a_i$. To reduce the influence of potentially extreme weights, we used stabilized weights[3], defined as $\widehat{w_i} = \widehat{P}(A = a_i)/\widehat{P}(C = c_i) = \widehat{P}(A = a_i)\widehat{w_i^*}$.

Our choice of weights as described above corresponds to the ATE for the overall cohort, either the US RPDR or the UK CPRD, with a covariate composition as detailed in Table 2 of the main text.

**Assessment of covariate balance between treatment groups.** In both cohorts, the balance between measured confounders in the two groups was achieved by inverse probability of treatment weighting. The achieved balance for age is presented in Extended Data Fig. 10, while overall covariate balance is summarized in Extended Data Fig. 11. In the estimation of treatment effects in strata of age, of sex and of BMI, we conducted separate analyses in each subgroup of patients[61] and estimated the IPTW weights for each stratum of the covariate.

**Covariate missingness**
In both cohorts, we quantified missingness rates for each covariate (see notes to Table 2), and missing values in these variables were treated as a separate category. For each categorical variable affected by missingness in the US RPDR (i.e., BMI and HbA1C), a binary indicator was added in the propensity score model. Similarly, for each categorical variable affected by missingness in the UK CPDR (i.e., BMI, HbA1C, smoking status), a binary indicator was added in the propensity score model.

**US RPDR.** In the RPDR cohort, there were missing values in both the BMI (32%) and HbA1C (38%) variables at antidiabetic treatment initiation. Missingness affected sulfonylurea- more than metformin-initiators, as 42% of them were missing BMI information at baseline and 52% did not have an HbA1C measure (Table 2). In the propensity score model, we treated missing data as a separate category, for both the baseline BMI and HbA1C variables. Further, we combined the two categories of missing BMI and BMI <25 into one, assuming that the baseline BMI would be more likely to be captured in the medical records if it were >25. In addition, we noted that the effects of missing BMI and BMI <25 indicators on treatment assignment were similar, and thus we collapsed the two into one reference category.

**UK CPRD.** Similarly, sulfonylurea initiators had more missing values than metformin initiators in the CPRD cohort, as 21%, 35 and 5% of them were missing BMI, HbA1C and smoking status information at baseline, respectively.

**Statistical methods**
In this study, our target estimands of interest are (a) the population hazard ratios (time-invariant) and (b) the population risk differences (time-varying).

**Estimation of treatment effect on all-cause mortality.** For a single time-to-death outcome, we estimated the Cox proportional hazards model and the nonparametric Kaplan–Meier survival curves for both

inverse-probability-of-treatment-weighted treatment arms. The latter model allows to estimate robustly the time-varying causal survival curves[62], while the former model provides a one-number summary of the treatment effect through a fixed hazard ratio. We used the Cox proportional hazards model

$$h^1(t) = h^0(t) \exp(\beta)$$

assuming that the hazards of death are proportional in two counterfactual worlds, a world where everyone receives metformin, $h^1(t)$, and a world where everyone receives sulfonylureas, $h^0(t)$, with a proportionality factor $HR = \exp(\beta)$.

Technically, to estimate the effect of metformin on all-cause mortality, we considered the same causal framework as for the competing risks (detailed below) but only for a single outcome, time-to-death. This means that both our analyses rely on the same causal assumptions (*A1*), (*A2*), and (*A3*), and the assumption of independent censoring. Practically, for both cases we used our R package (version 1.0.3), *causalCmprsk*, to estimate the causal survival curves[63].

To allow some time for the antidiabetic drug to have an effect, we introduced a 1-year lag: in our trial emulations, the follow-up started a year after treatment initiation. Therefore, the effect measures we estimated are to be interpreted for the cohort of patients who survive at least a year post baseline.

**Estimation of treatment effect on dementia in the presence of competing death.** In this section, we provide details on statistical methods used for the estimation of the intention-to-treat effect of metformin versus sulfonylureas on the risk of dementia in the presence of competing death.

To allow some time for the antidiabetic drug to have an effect on dementia, a 1-year lag was introduced: in our trial emulations, the follow-up started a year after treatment initiation. Therefore, effect measures are to be interpreted for the cohort of patients who are at risk for both events, dementia and death, a year after baseline. Patients who developed dementia within twelve months post antidiabetic treatment initiation may well have had cognitive problems at baseline and would likely have not met the eligibility criteria of any randomized clinical trial. Similarly, patients who died in the first year likely had a high mortality risk and would not be included in an actual clinical trial.

**Notation**

Let $T$ denote the time from treatment initiation to dementia onset or death (without prior dementia), whichever comes first. Let $E$ denote the indicator of the type of event, with $E = 1$ if $T$ corresponds to dementia onset, and $E = 2$ if $T$ corresponds to death. If neither dementia nor death is observed during the follow-up period, then $T$ is censored by the time to the last visit, and $E = 0$. It is important to note that we only consider here death without having prior dementia, i.e., the direct transition to death that does not go through the dementia state (see Extended Data Fig. 12). Dementia and death (without prior dementia) are two mutually exclusive outcomes, and it is assumed that treatment can potentially affect both. The observed data are assumed to be $n$ independent observations of the quadruplet $(T, E, A, C)$, i.e., $(t_i, e_i, a_i, c_i)$, for $i = 1, \ldots, n$.

**Assumptions**

Let $(T^a, E^a)$, for $a = 0, 1$ denote the potential outcomes that would be observed if a patient were to receive treatment a. Our causal assumptions are the following:

**(*A1*) No unmeasured confounding:** treatment assignment $A$ is independent of potential outcomes given $C$, i.e.:

$$A \perp (T^a, E^a) | C, \text{ for } a = 0, 1$$

**(*A2*) Positivity:**

$$0 < P(A = a | C) < 1, \text{ for } a = 0, 1$$

**(*A3*) SUTVA (Stable Unit Treatment Value Assumption):** the outcome of every patient does not depend on the treatment of others (non-interference), and the outcome does not depend on the way a treatment was assigned (consistency).

In addition, we assume that given $A$, the time to the last visit (censoring time) is independent of the outcome $(T, E)$.

**Measures of treatment effect**

Let $h_k^a(t)$ ($a = 0, 1$; $k = 1, 2$) be the single-world cause-specific hazards of transitioning to states 1 or 2 in a world corresponding to treatment $a = 0, 1$ (see Extended Data Fig. 12). This quantity is defined as follows:

$$h_k^a(t) = \lim_{\Delta t \to 0} \frac{1}{\Delta t} P(t \le T^a < t + \Delta t, E^a = k | T^a \ge t), \text{ for } k = 1, 2.$$

The single-world cumulative incidence functions (CIF) are defined by:

$$CIF_a(t, k) = E\left[ I_{(T^a \le t, E^a = k)} \right] = P(T^a \le t, E^a = k) = \int_0^t S_a(s) h_k^a(s) ds, \text{ for } k = 1, 2$$

(1)

where $S_a(t)$ is an overall survival function in the counterfactual world corresponding to treatment $a$, i.e., $S_a(t)$ is the probability of not having any event, neither dementia nor death, by time $t$: $S_a(t) = \exp\left\{ -\int_0^t h_1^a(s) ds - \int_0^t h_2^a(s) ds \right\} = S_1^a(t) S_2^a(t)$, where $S_1^a(t) = \exp\left\{ -\int_0^t h_1^a(s) ds \right\}$ and $S_2^a(t) = \exp\left\{ -\int_0^t h_2^a(s) ds \right\}$.

From (1), it is clear that the risk of dementia, denoted by $CIF_a(t, 1)$ ($a = 0, 1$), depends on the cause-specific hazard of death, denoted by $h_2^a(t)$ ($a = 0, 1$), through the overall survival function $S_a(t)$ ($a = 0, 1$). The function $CIF_a(t, k)$ ($a = 0, 1$; $k = 1, 2$), which is often called *risk*[61], represents the absolute probability of failing from cause $k = 1, 2$ by time $t$, in the counterfactual world corresponding to treatment $a = 0, 1$.

We emphasize that a risk function, $CIF_a(t, k)$ ($a = 0, 1$; $k = 1, 2$), is much more intuitive for interpretation and communication of findings than a *hazard* (or *rate*) parameter, $h_k^a(t)$ ($a = 0, 1$; $k = 1, 2$). The latter represents an instantaneous probability of failure from cause $k$ at time $t$, conditional on still being at risk at time $t$, in the counterfactual world corresponding to treatment $a = 0, 1$[64,65].

In our emulations of a target trial, we used the two following measures of treatment effect.

(a) The hazard ratios for both events are defined by:

$$HR_k(t) = \frac{h_k^1(t)}{h_k^0(t)}, \text{ for } k = 1, 2$$

Notice that here the ratios $HR_k(t)$ ($k = 1, 2$) can depend on time $t$, since they are defined in complete generality regardless of the statistical model used for estimation of hazard functions $h_k^a(t)$ ($a = 0, 1$; $k = 1, 2$). However, it is often assumed that $HR_k(t)$ ($k = 1, 2$) are time-invariant and equal to a constant value $HR_k$ ($k = 1, 2$) for all time points $t$, which follows from assuming the Cox proportional hazards model (PH) for $h_k^a(t)$ ($a = 0, 1$; $k = 1, 2$), as defined by [2] below. The PH assumption cannot be tested in general, since only one potential outcome is observed for every person. However, it can be tested or checked graphically under causal assumptions (*A1*), (*A2*), and (*A3*) listed above.

Although the time-invariant hazard ratios $HR_k$ ($k = 1, 2$) are problematic parameters for causal inference due to their non-collapsibility[66–68], they are traditionally used as effect measures in

the medical literature. To conform with previous research on antidiabetic drugs and their effects on dementia, we thus considered hazard ratios in our target trial emulations as well.

(b) The risk difference functions are defined by:

$$RD(t,k) = E\left[I_{(T^1 \leq t, E^1 = k)}\right] - E\left[I_{(T^0 \leq t, E^0 = k)}\right] = CIF_1(t,k) - CIF_0(t,k), \text{ for } k = 1, 2.$$

$RD(t,k)$ ($k = 1, 2$) is the average treatment effect (ATE) on getting outcome $k$ by time $t$. We chose the risk difference as a summary of a treatment effect, but other options, e.g., risk ratios, could be considered as well.

### Estimation

**Assuming the proportional hazards model.** Under assumptions (A1), (A2), and (A3), we checked graphically that both $HR_k(t)$ ($k = 1, 2$) do not depend on $t$. This allowed us to use the Cox PH models for both transitions:

$$h_k^0(t) = h_{0k}(t); h_k^1(t) = h_{0k}(t) \exp(\beta_k), \text{ for } k = 1, 2. \qquad (2)$$

Model 2 assumes that cause-specific hazards for dementia are proportional in two counterfactual worlds, a world where everyone receives metformin and a world where everyone receives sulfonylureas, with a proportionality factor $HR_1 = \exp(\beta_1)$. Similarly, according to model 2, the hazards $h_2^1(t)$ and $h_2^0(t)$ corresponding to the direct transition to death (without prior dementia) are assumed to be proportional with a proportionality factor $HR_2 = \exp(\beta_2)$.

**Details on the estimation procedure for treatment effect measures.** The estimators of $HR_k$, $CIF_a(t,k)$ ($a = 0, 1$; $k = 1, 2$), and $RD(t,k)$ ($k = 1, 2$) are obtained by plugging in the estimators of $\beta_k$ and $h_{0k}(t)$ ($k = 1, 2$). The estimator of $\beta_k$ ($k = 1, 2$) is the solution of a weighted version of the Cox score equation (Cox, 1972) and the estimator of the cumulative baseline hazard function $H_{0k}(t) = \int_0^t h_{0k}(s)ds$ ($k = 1, 2$) is a weighted version of the Breslow-type estimator with a plugged-in $\widehat{\beta_k}$[12]. The estimator of $HR_k$ is $\widehat{HR_k} = \exp\left(\widehat{\beta_k}\right)$ ($k = 1, 2$) and $RD(t,k)$ is given by: $\widehat{RD}(t,k) = \widehat{CIF}_1(t,k) - \widehat{CIF}_0(t,k)$, for $k = 1, 2$ (for more details, see ref. 58).

$$\sum_{i=1}^n \widehat{w_i} I(e_i = k) \left[ a_i - \frac{\sum_{j=1}^n \widehat{w_j} a_j \exp(\beta_k a_j) I(t_j \geq t_i)}{\sum_{j=1}^n \widehat{w_j} \exp(\beta_k a_j) I(t_j \geq t_i)} \right] = 0$$

where $\hat{w}_i$ are the weights emulating baseline randomization defined above.

(a) The estimator of the cumulative baseline hazard function $H_{0k}(t) = \int_0^t h_{0k}(s)ds$ ($k = 1, 2$) is a weighted version of the Breslow-type estimator with a plugged-in $\widehat{\beta_k}$[12]:

$$\widehat{H_{0k}}(t) \sum_{i: e_i = k} \frac{\widehat{w_i} I(t_i \leq t)}{\sum_j \widehat{w_j} \exp(\widehat{\beta_k} a_j) I(t_i \leq t_j)}$$

(b) The estimator of $HR_k$ is $\widehat{HR_k} = \exp\left(\widehat{\beta_k}\right)$ ($k = 1, 2$).

(c) The estimator of $CIF_a(t,k)$ is given by:

$$\widehat{CIF_a}(t,k) = \sum_{i: e_i = k, t_i \leq t} \frac{\widehat{w_i} \exp(-\widehat{H_{01}}(t_i) e^{\widehat{\beta_1} a} - \widehat{H_{02}}(t_i) e^{\widehat{\beta_2} a})}{\sum_j \widehat{w_j} \exp(\widehat{\beta_k} a_j) I(t_i \leq t_j)}, \text{ for } a = 0, 1; k = 1, 2$$

(d) The estimator of $RD(t,k)$ is given by: $\widehat{RD}(t,k) = \widehat{CIF}_1(t,k) - \widehat{CIF}_0(t,k)$, for $k = 1, 2$.

We estimated the 95% confidence intervals for all the parameters using the Bayesian bootstrap (https://bcbio-nextgen.readthedocs.io/), where a bootstrap sample comprises an original cohort, but every subject's contribution is reweighted with a random bootstrap weight

$w_i^{bs} = V_i/\bar{V}$ for $i = 1, \ldots, n$, where $V_1, \ldots, V_n \sim Exp(1)$ are independent, and $\bar{V} = \frac{1}{n} \sum_{i=1}^n V_i$.

For each bootstrap replication, we repeated the steps of fitting the logistic regression to obtain the balancing weights, and the steps (a)–(e), in order to obtain estimates for all the parameters, i.e., $\beta_k$, $H_{0k}(t)$, $CIF_a(t,k)$ ($a = 0, 1$; $k = 1, 2$), and $RD(t,k)$ ($k = 1, 2$) for each of the bootstrap samples.

The Bayesian bootstrap is a better and more stable alternative to the standard bootstrap in survival data, since it does not have a problem of ties, and since the risk sets in all Bayesian bootstrap replications change at the same time points as in the original sample. The 95% confidence interval was obtained as 2.5th and 97.5th percentiles from the distributions of bootstrap estimates. For the time-dependent parameters such as $CIF_a(t,k)$ ($a = 0, 1$; $k = 1, 2$) or $RD(t,k)$ ($k = 1, 2$), the confidence intervals were obtained pointwise for every $t$. In our target trial emulations, we used 500 bootstrap replications.

**Checking a proportional hazards assumption using a nonparametric framework.** Under assumptions (A1), (A2), and (A3), we checked graphically that both $HR_k(t)$ ($k = 1, 2$) do not depend on $t$. We did this by using the nonparametric framework which does not assume any structure for the hazard functions $h_k^a(t)$ ($a = 0, 1$; $k = 1, 2$).

In the all-cause mortality analyses, we tested the PH assumption using other approaches as well. These include the global test based on Schoenfeld residuals and a log-rank test. The tests indicated that the proportional hazards assumption was not violated in the US RPDR cohort. However, given that the graphical check revealed violation of the PH assumption in the UK CPRD cohort, we used nonparametric estimates of causal survival curves based on IPTW Kaplan–Meier estimates.

**Sensitivity analysis to the choice of weighting strategy.** In addition to the main analysis based on ATE weights, we conducted a sensitivity analysis based on ATT weights. When using ATE weights, one seeks to address the following question: "At the population level, what is the effect of initiating on metformin rather than sulfonylureas?". In contrast, when using ATT weights, one seeks to address the following question: "Among treated patients, what is the effect of initiating on metformin rather than sulfonylureas?". Across the three outcomes of interest, i.e., all-cause mortality, dementia onset, and death without dementia, we found consistent results. With respect to all-cause mortality, treatment decreased the hazard times 0.57 and 0.60 on average when using ATE and ATT weights, respectively. For dementia onset, treatment decreased the cause-specific hazard times 0.81 and 0.80 on average when using ATE and ATT weights, respectively. Similarly, results obtained for death without dementia aligned: treatment decreased the cause-specific hazard times 0.60 on average, irrespective of the weighting strategy.

**Sensitivity analysis to covariate adjustment.** In addition to the main analysis relying on covariate balancing via IPTW and no further covariate adjustment in the outcome models, we conducted a sensitivity analysis with further adjustment for age (considered as a continuous variable) and sex in the Cox PH models. Across the three outcomes of interest, i.e., all-cause mortality, dementia onset, and death without dementia, we found consistent results. With respect to all-cause mortality, treatment decreased the hazard times 0.58 and 0.57 on average with and without adjustment, respectively, when using ATE weights (0.62 and 0.60 with ATT weights). For dementia onset, treatment decreased the cause-specific hazard times 0.83 and 0.81 on average with and without adjustment, respectively, when using ATE weights (0.81 and 0.80 with ATT weights). Similarly, results obtained for death without dementia aligned: treatment decreased the cause-specific hazard times 0.62 and 0.60 on average with and without adjustment, respectively, irrespective of the weighting strategy.

### Systems pharmacological analysis of metformin and sulfonylureas in human neural cells

Both 10 µM and 40 µM of metformin or glyburide were added to differentiated human ReNcell VM neural cultures for 72 h (drugs were refreshed at 48 h). At 72 h, RNA was isolated using RNease mini kit (catalog #74104, Qiagen, Germantown, MD). RNA quality was verified using Bioanalyzer (Agilent, 2100 Bioanalyzer Systems); all samples scored RINs of >9.0. RNA-sequencing library preparation was performed with the TruSeq Stranded mRNA Library Prep Kit (Illumina) following the manufacturer's protocol at half reaction volume. Input for each sample consisted of 500 ng of RNA and 5 µl of 1:500 diluted ERCC spike-in mix (Ambion). Libraries were amplified for 12 cycles during the final amplification step. Libraries were sequenced on a NextSeq RNA sequencer (Illumina). Raw sequencing reads were aligned against the hg38 (build 94) reference and quantified using the bcbio-nextgen RNA-seq analysis pipeline[69]. Differential gene expression between compound-treated samples and DMSO controls was performed by the R package *edgeR* version 3.26.5[70]. Genes were subsequently sorted by the resulting log-fold change values and queried against canonical pathways in the Molecular Signatures Database[71] using Gene Set Enrichment Analysis[72]. Secreted genes were identified by detection in human CSF proteome[73]. The differential expression analysis of the human SPP*1* gene in gene expression profiles of AD brains from AMP-AD datasets was conducted as follows. The aligned RNAseq data was provided as input to differential gene expression analysis contrasting advanced Braak stages (Braak V, VI) versus controls (Braak 0, I, II) in the ROSMAP, Mount Sinai Brain Bank and Mayo Brain Bank cohorts as described elsewhere[73,74]. Differentiated human ReNcell VM neural cells were grown for various times in the presence of 10 µM or 40 µM of metformin or glyburide. At prespecified time points, the medium was withdrawn, and human SPP1 protein levels were analyzed by ELISA (ThermoFisher). The results were analyzed using ANOVA with drug and concentrations as covariates with post hoc Tukey tests used to test for significant findings. The R/Bioconductor package *limma* was used for analysis. Gene expression profiles of postmortem brain specimens along with the corresponding clinical annotations were downloaded from Synapse (www.synapse.org/AMPAD) and conducted using www.alzdatalens.org[74]. The methodology being used to measure and harmonize RNA-seq values across the brain samples has been previously described[75,76] and can be found at https://github.com/Sage-Bionetworks/amp-rnaseq.

#### Reporting summary

Further information on research design is available in the Nature Portfolio Reporting Summary linked to this article.

## Data availability

US RPDR: Researchers can obtain an anonymized version of the study dataset from the authors upon request and completion of the MGB Health data use agreement for the use of RPDR data. This agreement ensures the privacy of MGB patients and compliance with US regulatory standards and has been approved by the MGB IRB. In addition, we provide a summary about patient data security and privacy. Patients who visit Mass General Brigham receive a HIPAA notice that states that their identifiable data may be used for research with proper Institutional Review Board approval. The patient has an opportunity to object to this usage of their data by seeking care outside of Mass General Brigham. The IRB classifies aggregate queries of online patient registries that are populated with appropriately obfuscated, de-identified/encrypted data, performed by authorized staff as a category of research that is exempt from IRB review. In addition, HIPAA privacy rules do not apply to de-identified information. Therefore, the Mass General Brigham Research Council determined that faculty, and those overseen directly by faculty, are approved for access to the Query Tool. However, with respect to the Detailed Data Wizard, the IRB must review and approve the release of identified and de-identified medical record data to researchers. In an effort to secure patient privacy and prevent a security breach, all patient identifiers are encrypted throughout the database. The detailed data requests which are stored on shared drives for researchers to access are also encrypted. UK CPRD: According to the UK Data Protection Act, information governance restrictions (to protect patient confidentiality) prevent data sharing via public deposition. Therefore, CPRD data that support the findings of this study are not publicly available. Data extracts can be requested by applying to the Clinical Practice Research Datalink for data spanning the years 2000 to 2018 (https://www.cprd.com). The code to process the data is available from the authors upon request. All requests will be answered in 30 days or less. Metformin/Glyburide RNA-seq: https://www.synapse.org/#!Synapse:syn22213067.

## Code availability

The R package (version 1.0.3), *causalCmprsk*, developed for the competing risks analysis is available at CRAN[63]. Extended data items 1, 2, 10, 11, 12, and 13 are available on GitHub: https://github.com/labsyspharm.

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

## Acknowledgements

We thank Alexander Soukas, Victor Castro, and Peter K. Sorger for helpful discussions. The authors thank Shawn Murphy, Henry Chueh, and the Partners Health Care Research Patient Data Registry group for facilitating use of their database. This study is based in part on data from the Clinical Practice Research Datalink database obtained under license from the UK Medicines and Healthcare products Regulatory Agency. However, the interpretation and conclusions contained in this article are those of the authors alone and not necessarily those of the NHS, the NIHR, or the Department of Health. The results published here are in part based on data obtained from the AMP-AD Knowledge Portal (doi:10.7303/syn2580853). These data were generated from postmortem brain tissue collected through the Mount Sinai VA Medical Center Brain Bank, led by Dr. Eric Schadt from Mount Sinai School of Medicine, by the Rush Alzheimer's Disease Center, Rush University Medical Center, Chicago, and by the following sources: The Mayo Clinic Alzheimer's Disease Genetic Studies, led by Dr. Nilufer Taner and Dr. Steven G. Younkin, Mayo Clinic, Jacksonville, FL using samples from the Mayo Clinic Study of Aging, the Mayo Clinic Alzheimer's Disease Research Center, and the Mayo Clinic Brain Bank. We thank the NIH R01 AG058063 (awarded to M.W.A.), P30 AG062421 (awarded to B.T.H.), P30 AG066512 (awarded to R.A.B.), a CART grant (awarded to M.W.A.), an administrative supplement to U54 CA22508 (awarded to M.W.A.), IBM Research (awarded to R.E.W. and S.N.F.), the Abdul Latif Jameel Clinic for Machine Learning in Health grant (awarded to R.E.W. and S.N.F.), the UK Dementia Research Institute which receives its funding from UK DRI Ltd, funded by the UK Medical Research Council (awarded to I.T.), the Alzheimer's Society and Alzheimer's Research UK, support by the British Heart Foundation Centre for Research Excellence at Imperial College (awarded to I.T.), the Hellenic Foundation for Research and Innovation (HFRI) (awarded to I.T.), and the General Secretariat for Research and Technology (GSRT) (awarded to I.T.) for support.

## Author contributions

I.T., D.B., S.D., and M.W.A. conceived of the study; M.L.C., B.V.L., C.M., and S.D. curated data and performed analyses on the RPDR EHR dataset; B.Z. and B.S. curated data and performed analyses on the CPRD EHR dataset; I.T. and L.M. supervised the CPRD EHR dataset analysis; M.L.C., B.V.L., B.Z., C.M., Y.-H.S., R.A.B., and S.D. conceived the analysis and wrote code for the competing risks analysis; K.E., S.R., A.S., and S.B. generated and analyzed data from the in vitro systems pharmacology studies; M.L.C., B.Z., B.V.L., C.M., S.F., R.E.W., D.B., S.D., and M.W.A. wrote the first draft; and all authors revised the manuscript. S.N.F., R.E.W., and M.W.A. secured funding for the study.

## Competing interests

The authors declare the following competing interests: B.T.H. has a family member who works at Novartis, and owns stock in Novartis; he serves on the SAB of Dewpoint and owns stock. He serves on a scientific advisory board or is a consultant for AbbVie, Aprinoia Therapeutics, Arvinas, Avrobio, Axial, Biogen, BMS, Cure Alz Fund, Cell Signaling, Eisai, Genentech, Ionis, Novartis, Sangamo, Sanofi, Takeda, the US Dept of Justice, Vigil, Voyager. His laboratory is supported by research grants from the National Institutes of Health, Cure Alzheimer's Fund, Tau Consortium, and the JPB Foundation – and sponsored research agreements from Abbvie, BMS, and Biogen. R.A.B. serves on an advisory board for Biogen. M.W.A. is a member of the SAB and owns shares in Aromha. He also is a consultant for Transposon Rx, TLL Pharma, and receives research support from IFF and TLL Pharma. A.S. is an employee of Flagship Labs 84, Inc, a subsidiary of Flagship Pioneering. The remaining authors declare no competing interests.

## Additional information

[1]Institute for Data, Systems, and Society, Massachusetts Institute of Technology, Cambridge, MA, USA. [2]Department of Statistics, University of Haifa, Mt Carmel Haifa, Israel. [3]Ageing Epidemiology Research Unit, School of Public Health, Imperial College London, London, UK. [4]Department of Neurology, Massachusetts General Hospital/Harvard Medical School, Boston, MA, USA. [5]Department of Epidemiology and Biostatistics, School of Public Health, Imperial College London, London, UK. [6]Laboratory of Systems Pharmacology, Harvard Program in Therapeutic Science, Harvard Medical School, Boston, MA, USA. [7]Department of Psychiatry, Massachusetts General Hospital/Harvard Medical School, Boston, MA, USA. [8]Inception Labs, Collaborative for Health Delivery Sciences, Medical College of Wisconsin, Wauwatosa, WI, USA. [9]Public Health Directorate, Imperial College London NHS Healthcare Trust, London, UK. [10]Department of Biostatistics, School of Global Public Health, New York University, New York, NY, USA. [11]Division of Clinical Informatics, Beth Israel Deaconess Medical Center, Boston, MA, USA. [12]Sloan School of Management, Massachusetts Institute of Technology, Cambridge, MA, USA. [13]Dementia Research Institute, Imperial College London, London, UK. [14]Department of Hygiene and Epidemiology, University of Ioannina, Ioannina, Greece. [15]Department of Epidemiology, Harvard T.H. Chan School of Public Health, Boston, MA, USA. [16]These authors contributed equally: Marie-Laure Charpignon, Bella Vakulenko-Lagun, Bang Zheng. [17]These authors jointly supervised this work: Ioanna Tzoulaki, Deborah Blacker, Sudeshna Das, Mark W. Albers.
✉e-mail: i.tzoulaki@imperial.ac.uk; dblacker@mgh.harvard.edu; sdas5@mgh.harvard.edu; albers.mark@mgh.harvard.edu

