## [Peer Review File · Nature Communications]

Reviewers' Comments:

Reviewer #1:

Remarks to the Author:

Charpignon and colleagues provide compelling evidence that metformin may be useful in mitigating cognitive decline in non-diabetics and that improved glycemic control is not the main basis for the benefit. The comparison of metformin and sulfonylureas implies that both modulate similar molecular events despite their differential effects on cognitive decline (metformin being the one with the more important effect on cognition). APOE and SPP1 are implicated, but again both drugs appear to modulate these factors (despite their differential effects on clinical cognitive function), and the isotype of APOE (the usual correlate of the cognitive effects of this molecule) is not considered. Certainly better understanding of whether and if so how metformin modulates clinical cognitive function would be most welcome, but I leave this paper with a headline that diabetes and glycemic control are not important for the procognitive effects, and even there, the failure of the metformin vs sulfonylurea to have a selective impact on cognition that is tightly linked to an omic change leaves me without a major takeaway. If there is a clear story here that underscores a clear pathway toward elucidation of metformin's effect on brain, I am not getting it from the way the data are currently presented.

Reviewer #2:

Remarks to the Author:

The study is trying to answer an ambitious question: if Metformin could reduce dementia risk. But it is a tough one to be answered. The study combined two parts, the first part used medical health records data, the second part used in vitro pharmacological system. Although it is an important question, I feel not very convinced about the conclusion derived from the study.

1. I understand that the study wants to clarify if metformin has some effects on dementia hazard based on two large EHR database. because previous two studies showed different effects of metformin (also based on health records). I personally do not think EHR dataset is suitable for addressing such kind of questions (if merformin could reduce dementia risk), because there are many factors that will affect AD onset, which are not able to be well controlled by EHR dataset, such as Genetics, other diseases, etc. Those factors are often hard to be controlled in the medical health records dataset. Probably a better design will be a randomized clinical trials using AD patients with clear genetic information, disease history and medical informations.

2. for major results, the study showed some effects of metformin (vs. Sul) on dementia onset; see figure 2-4. But there is no other risk factors controlled, eg. genetics (APP, PS1, PS2, APOE, SORL1, TREM2); or mortality (there are other factors affecting mortality).

3. In abstract "Together, our findings suggest that metformin might prevent dementia in patients without type II diabetes."

I'm unclear about why the study is focusing on non-diabetic patients, as we know that diabetes patients have higher risk of developing AD; which might be more suitable for the study. And why those participants were given metformin if they are not diabetic?

4. The study design is to compare the effects of metformin vs sulfonylureas on the risk of death and dementia. It is unclear why selecting sulfonylureas, and would it be better to include a placebo group (as a control) or a group treated with anti-AD drugs?

5. Previous beneficial effect of metformin was only observed in younger African Americans (see reference 10 and 11). But how is the effect in the current study of AA? and most participants are whites, instead of AA in the current study.

6. some sentences are confusing, eg. page 4 paragraph 3: "As in the US cohort, metformin initiators were younger than their sulfonylurea counterparts (Table 2)." does it mean those treated with metformin?

7. figure 5d

It is needed to have a detail methods to describe characteristics of samples, sample numbers and how the proteins were measured? also how about APOE level in the brain samples?

8. table 2

how many of them are diabetes? or hyperglycemia?

9. figure 2

survival can be affected by many other factors: eg. other drugs, and treatments; eg. if there are some surgeries for cancers/CVD. and there is no placebo group

10. figure 3.

many other factors will affect dementia onset, eg. genetics (APOE status), other diseases (eg. diabetes and CVDs). the effect observed here could be compounded by other factors.

11. It is nice to have a in vitro system to test the effect of metformin in human neural cells. but it is not well established about the link between metformin and dementia

Reviewer #3:

Remarks to the Author:

This is a solid analysis using inverse propensity weighting but I dislike the suggestion that the analysis is replicating a randomized trial. That must be removed. I would consider adding the ATET estimator. That is, compare the metformin users to a balanced population not receiving metformin (weights = $(1-p)/p$). I would also consider estimating the population hazard ratio controlling for sex or age (sex and age in the structural model).

Specific Points

"Emulation of baseline randomization. The covariates defined above were used to emulate baseline randomization." – I would remove the idea that use of weighting emulates a randomized trial. Like methods that model the outcome in terms of exposure and covariates, weighting does require the assumption of no omitted covariates. However it is worth stating that the weighting does achieve or approximates balance with respect to the measured covariates.

ATE – please add definition/description of the average treatment effect. In addition, the estimand of interest is a hazard ratio. In this case reference to "average" treatment effect is not quite appropriate. The quantity being interested is the population hazard ratio; the ratio of hazards to compare the counterfactual population if everyone received metformin to the counterfactual population if no one did. The same goes for the sub-distribution hazard ratio.

"In the estimation of treatment effects in strata of age, of sex and of BMI, we conducted separate analyses in each subgroup of patients and estimated the IPTW weights for each stratum of the covariate." – Do you mean that for each subgroup, you conducted a separate IPTW analysis, i.e. find a different set of weights for each strata ?

It is not clear why there is statistical analysis described before the section titled "Statistical Analysis"

"time-varying causal survival curves" – survival curves are by definition functions of time

"In our emulations of a target trial, we used the two following measures of treatment effect" – I find this language disingenuous

The statistical analysis section is too long as is. Most of it should be in a supplement.

Reviewer #1

COMMENT 1. Charpignon and colleagues provide compelling evidence that metformin may be useful in mitigating cognitive decline in non-diabetics and that improved glycemic control is not the main basis for the benefit.

Thank you for this comment. We are very grateful for the Reviewer's encouraging words.

Importantly, we wish to point out that the two target trials were carried out in type 2 diabetic patients in two distinct EHR databases, using harmonized methods. Furthermore, we agree that our novel approach to electronic health record (EHR) analysis for drug repurposing offers compelling evidence that metformin may mitigate cognitive decline in the preclinical time period before the clinical diagnosis of mild cognitive impairment or dementia. Moreover, we agree with the summary statement that the beneficial effects of metformin on cognition are not likely caused by improved glycemic control.

COMMENT 2. The comparison of metformin and sulfonylureas implies that both modulate similar molecular events despite their differential effects on cognitive decline (metformin being the one with the more important effect on cognition). APOE and SPP1 are implicated, but again both drugs appear to modulate these factors (despite their differential effects on clinical cognitive function).

We would like to emphasize that our data and analyses reveal that metformin modulates different molecular pathways relative to glyburide (a commonly used sulfonylurea), in parallel to their differential effects on cognitive decline. That is, among type 2 diabetic patients, initiation with metformin monotherapy was more beneficial than initiation with sulfonylurea monotherapy with respect to (their) subsequent cognitive decline.

In the present study, we make a step towards identifying the actions of metformin in the brain by first investigating differences in gene expression profiles between differentiated human neural cells treated with either metformin or glyburide, at pharmacologically-relevant concentrations. As reported in the Results section, pathway analysis revealed that metformin-altered genes were enriched in pathways related to an innate immune pathway (defensins), whereas glyburide-altered genes had the opposite effect on this innate immune pathway (**Extended Data Table 8; see p.62 and attached Excel file**). In addition, we found that metformin- and glyburide-treated differentiated human neural cells differentially expressed genes encoding secreted proteins that are measurable in the cerebrospinal fluid (CSF). Specifically, both APOE and SPP1 were significantly altered in a dose-dependent manner (**Fig. 5c, Extended Data Fig. 7; see p.28 and p.50**). Importantly, our findings are clinically actionable, i.e., the corresponding protein levels are measurable in the CSF of living patients. Furthermore, these proteins have been implicated in AD risk (APOE) or pathogenesis (SPP1), as

noted. They are novel in that they have not been previously reported in the vast literature of metformin's gene expression profiles. Further validation of the results of our study in other patient cohorts are needed to determine whether these proposed pharmacodynamic CSF biomarkers are robust. Future research beyond the scope of this paper will be needed to elucidate a deeper understanding of metformin's mechanism of action in the human brain. A convergent description of metformin's mechanism of action has been elusive as findings in numerous cancer cell types and insights from model organisms have not translated readily to other cell types. In response to this comment, we have added the following sentence about our study's limitations in the Discussion section: **"The candidate pharmacodynamic biomarkers for metformin's actions in the brain (SPP1 and APOE) will need to be validated in the CSF of patients taking metformin or a sulfonylurea"** (see p.10).

COMMENT 3. The isotype of APOE (the usual correlate of the cognitive effects of this molecule) is not considered.

Thank you for this comment. We agree that it would be informative to stratify the effects of metformin based on APOE genotyping. However, our study cohorts were clinical samples, i.e., the data for each individual patient was drawn from their EHR. Currently, APOE genotyping is not standard of care for type 2 diabetes treatment or for dementia treatment. For this reason, APOE genotyping information was not readily available in either EHR database. Further, knowledge of APOE genotypes will not influence clinical decision-making of patients who already have type 2 diabetes or dementia.

In an attempt to address this point, we tried to identify patients enrolled in our EHR target trials (observational studies) who had also participated in research to have their genotypes determined under a research protocol. Notably, the Mass General Brigham (MGB) BioBank is committed to increase the number of enrollees. We are currently working closely with their team and monitoring the number of type 2 diabetic patients in our cohort who consented to share their genetic information with MGB BioBank. However, the number of patients with available APOE genotypes in the U.S. RPDR cohort was unfortunately too small (1,094 or 8.3%) to have adequate power to assess the possibly heterogeneous effect of metformin APOE2-, APOE3-, and APOE4-defined strata. Indeed, we would anticipate 25-30% of these patients to carry the APOE4 allele, putting them at high risk of dementia onset. This would imply comparing ~300 high-risk individuals against ~800 low-risk controls, before further stratification by age and sex—the latter being an important APOE4 effect modifier [1].

In the U.K. CPRD, we were restricted to information which is routinely gathered in primary care and thus available in patient EHR. As explained in detail by Ford et al. in a recent publication about the detection of dementia patients in the CPRD database [2], known risk factors such as APOE4 genotype are not well-captured in these records.

In response to this comment, we have thus added APOE genotypes to the list of limitations of our study: “**Additionally, relevant lifestyle factors, like diet and physical exercise, and a genetic risk factor, APOE genotypes, were unavailable**” (see p.11).

COMMENT 4. Certainly better understanding of whether and if so how metformin modulates clinical cognitive function would be most welcome, but I leave this paper with a headline that diabetes and glycemic control are not important for the procognitive effects, and even there, the failure of the metformin vs sulfonylurea to have a selective impact on cognition that is tightly linked to an omic change leaves me without a major takeaway. If there is a clear story here that underscores a clear pathway toward elucidation of metformin's effect on the brain, I am not getting it from the way the data are currently presented.

Thank you for this feedback. We regret that we were not clear about the main takeaways from this manuscript. We have summarized them below:

Main takeaway: Among incident type 2 diabetic patients, treatment initiation with metformin, as opposed to sulfonylureas, reduces the risk of dementia onset in two sizable EHR databases, after accounting for death as a competing risk.

Subclaim 1: Previous observational studies of metformin's action to reduce dementia onset have had mixed results. However, these studies did not account for death as a competing event. The target trial emulation methodology described in this manuscript is a novel approach to quantify the actions of metformin on dementia onset in incident type 2 diabetic patients (see **Reviewer #3, Comment 2** for more details). To our knowledge, our work is the first application of this methodology to dementia onset as an outcome, and we hope it will become the new standard to carefully evaluate drug repurposing candidates for dementia onset.

Subclaim 2: The beneficial effect of metformin on dementia onset cannot be solely due to better diabetes management through the control of hyperglycemia, prompting us to investigate alternative modes of action for metformin's beneficial effect.

Subclaim 3: The gene expression of SPP1 and APOE were both uniquely and significantly reduced by exposure to metformin, a drug that penetrates the blood-brain barrier, at pharmacologically relevant drug concentrations. Quantifying SPP1 and APOE in the CSF would validate these candidate biomarkers both in natural history, longitudinal studies of disease progression, and in response to metformin drug therapy in the context of a prospective clinical trial or retrospective clinical study.

We have modified the opening of the Discussion section to emphasize the main takeaway and the subclaims of our work for the reader.

Reviewer #2

COMMENT 1. I understand that the study wants to clarify if metformin has some effects on dementia hazard based on two large EHR databases because two previous studies showed different effects of metformin (also based on health records). I personally do not think EHR dataset is suitable for addressing such kind of questions (if metformin could reduce dementia risk), because there are many factors that will affect AD onset, which are not able to be well controlled by EHR dataset, such as Genetics, other diseases, etc. Those factors are often hard to be controlled in the medical health records dataset. Probably a better design will be a randomized clinical trial using AD patients with clear genetic information, disease history, and medical information.

Thank you for sharing your thoughts. We agree that a randomized, double blind, placebo-controlled clinical trial of metformin in cognitively normal seniors (not AD patients, as it was suggested), with dementia onset as the primary endpoint, would provide the strongest evidence to address the hypothesis that metformin can act in the preclinical phase of the disease and prevent dementia onset. Unfortunately, RCTs in asymptomatic seniors with dementia onset are not feasible: they are underpowered due to a slow rate of disease progression and remain too expensive to conduct at this date.

Observational data from electronic health records has become increasingly available due to digitization of medical information. EHR-based retrospective data analysis relying on valid causal inference frameworks is valuable and can help explore counterfactual questions such as: “What would have happened to a patient’s (cognitive) health if they were given a repurposed drug?”. Observational studies have two added benefits: first, they can be conducted inexpensively, in contrast with RCTs; second, they are highly relevant for the investigation of long-term outcomes such as dementia onset and death and provide a longer time horizon for analysis than is feasible in a clinical trial.

Importantly, our work builds upon a growing literature on the *in-silico* pre-evaluation of potential drug repurposing candidates. Although the existing literature predominantly investigates cancer-related outcomes [3,4,5,6,7], more EHR-based drug repositioning studies and reviews focusing on dementia as the outcome of interest are being published [8]. In their reviews, Bauzon et al. [9], Zhang et al. [10], and Park [11] describe in depth the current stage of the field of EHR-based drug repurposing evaluation.

We embarked on applying the target trial methodology to the two aforementioned EHR databases, with death as a competing outcome precluding the onset of dementia. The goal was to measure the potential benefits offered by metformin towards years of life without dementia and towards overall survival. In addition to providing real-world evidence that informs the design of a prevention clinical trial of metformin for dementia onset, the target trial methodology is amenable to evaluating pre-specified patient

subpopulations, e.g., based on age at treatment initiation or sex. A positive signal in these pre-specified patient strata may assist in the determination of a set of inclusion/exclusion criteria of future RCTs to increase their power.

We also agree that adjusting for prior disease history and comorbidities is critical. Although adjusting for genetic factors was challenging, we made our best effort to incorporate available clinical information as part of the analyses presented in the manuscript. In this study, we did control for other diseases in our models, namely by adjusting for a recorded medical history of hypertension, cardiovascular disease (CVD), stroke, and cancer, as explained in the **Supplementary Material (Confounder selection paragraph of Part V. titled Covariates; see pp.32-33)** and presented in **Table 2 (see p.22)**.

For a thorough discussion of genetic risk factors, see our detailed comments about APOE in response to **Reviewer #1, Comment 3** and **Reviewer #2, Comment 2**. In addition, we would like to highlight three critical points:

- 1. Study design:** Our study cohorts were clinical samples, i.e., the data for each individual patient was drawn from their EHR. Currently, APOE genotyping is not standard of care for type 2 diabetes treatment or for dementia treatment. Therefore, information related to APOE isotype and other genetic factors was not available from this source.
- 2. Statistical estimation:** Even if such genetic factors were readily accessible, they would not constitute a source of confounding between treatment and outcomes: indeed, the nature of a patient's APOE isotype and/or other gene alleles is not commonly visible to the physician deciding which antidiabetic treatment to prescribe to their patient. In the scenario that such genetic factors were weakly associated with clinical decision-making about antidiabetic prescription, it would most likely not affect the significance of our results, based on the robustness of multiple sensitivity analyses we conducted (**see Reviewer #3, Comment 3**). Moreover, the goal of our study is to robustly test hypotheses regarding the repurposing potential of metformin for cognitive health and longevity, but not to claim conclusive results that only RCTs can achieve.
- 3. Selection bias:** Further, although a subset analysis among patients who enrolled in a biobank and provided consent for use of their genetic data in research studies could be conceivable in the future, we should be aware that this filtering could lead to marked selection bias due to the lack of sociodemographic representation among biobank enrollees to date [**12,13,14,15,16**].

COMMENT 2. For major results, the study showed some effects of metformin (vs. Sul) on dementia onset; see figure 2-4. But there is no other risk factors controlled, eg. genetics (APP, PS1, PS2, APOE, SORL1, TREM2); or mortality (there are other factors affecting mortality).

Thank you for these remarks. Based on your comment, we edited the abstract to further clarify our approach, which **(a)** does account for death as a competing event and **(b)** additionally controls for other risk factors such as medical history of hypertension, cardiovascular disease (CVD), stroke, and cancer (see **Table 2** on **p.22** for the full list of considered confounders, along with their distributions, in our cohorts).

Death is a competing event that precludes the development of dementia, but the use of a competing risk analysis in previous studies **[17,18]** has been limited. To our knowledge, this work is the first attempt to comprehensively address competing death in a study of metformin and dementia, with a rigorous causal framework harmonized across two EHR databases.

(a) In this research, we used a competing risk approach **[19]** to address the conundrum that metformin might reduce the risk of death and put more people at risk of developing dementia, while simultaneously reducing the risk of dementia. Results of the all-cause mortality analysis are presented in the **Results section (paragraph 2 titled *Metformin improved survival relative to the sulfonylureas in the US and UK cohorts*; see pp.4-5)**, while results of the competing risk analysis, in which dementia onset and death without dementia are competing events, are presented in **paragraphs 3 and 4 of that same section (see pp.5-7)**. While prior studies only estimated a cause-specific hazard ratio for the impact of treatment on risk of dementia, our competing risk framework allowed the joint estimation of metformin's treatment effects on two outcomes, dementia and death without dementia.

Our findings show that, in intention-to-treat analyses, metformin was associated with a lower hazard of all-cause mortality than sulfonylureas in both cohorts. In competing risk analyses, there was also a lower cause-specific hazard of dementia onset among metformin initiators. Further, our competing risk analysis has two added benefits. First, it also shows how the risk of dementia depends on the baseline mortality rate of the population, a potential explanation of the neutral **[20,21]** or deleterious **[22]** effect of metformin on dementia onset seen in some other studies. Second, it also enables a fair comparison of the absolute cumulative hazards of dementia and death across clinical samples, thereby facilitating the interpretation of differences in both the magnitude and trajectory of diagnosed dementia between the U.S. RPDR and U.K. CPRD cohorts **(Extended Data Fig. 3; see p.46)**.

(b) Further, we carefully embedded the competing risk approach within a causal inference framework, namely target trial emulation. For this, we adjusted for important confounders of dementia onset and death without dementia. Our approach relies on guidelines previously formulated by VanderWeele, Lesko et al., and Brookhart et al. **[23,24,25]** regarding the use of control factors in time-to-event studies. When adjusting for sources of confounding, this prior work recommends including covariates that are weakly related to the treatment assignment but are strongly related to the outcome of interest. In our study, we considered all available covariates that potentially influence

both the treatment assignment and one or both outcomes. To harmonize the confounders included in the dementia onset and death model, we considered covariates that were influencing either outcome.

As a result, in both patient cohorts, information on the following covariates before the baseline date was extracted: age at the first prescription, sex, socioeconomic status or SES (index of multiple deprivation (IMD) in the U.K. CPRD and median annual household income by zip code in the U.S. RPDR), BMI (<25, 25-30, ≥30 kg/m², or missing), HbA1C (<7%, 7-10%, >10%, or missing), and other risk factors (hypertension, CVD, stroke, and cancer). Additional covariates which were not available in the U.S. RPDR dataset were extracted in the U.K. CPRD cohort, including smoking status (non-smoker, current smoker, ex-smoker, or missing), and presence of chronic obstructive pulmonary disease (COPD) before baseline. Whereas the U.S. RPDR cohort mainly includes patients living in the Boston area, the U.K. CPRD cohort is representative of patients nationally. To adjust for geographical heterogeneity, the region of residence was additionally incorporated in the U.K. CPRD (as a categorical covariate with 12 levels and the reference).

Specific note regarding adjustment for genetic risk factors:

(i) Three of the aforementioned risk factors (APP, PS1, and PS2) were not considered relevant in this study. Reasons for not considering these factors include their low prevalence (~3%) [26] and their clinical expression as early-onset dementia with symptom onset prior to the age of 60, e.g., 35-54 years [27]. In contrast, we set the cohort inclusion/exclusion criteria to focus on patients with late-onset, sporadic dementia, which accounts for 97% of dementia cases. Indeed, we only selected asymptomatic patients, with a mean age at baseline of 69 in the U.S. RPDR and 66 in the U.K. CPRD, as they were the focus of our work. Notably, patients with a record of memory loss at baseline were excluded.

(ii) SORL1 and TREM2 are two genetic markers that are not available in the EHR. That is why it is generally infeasible to control for such factors in retrospective, observational data settings. Although these markers may be considered in clinical trial research studies, related information was not provided in the EHR systems involved in this study.

(iii) Please refer to our detailed comments about APOE in response to **Reviewer #1, Comment 3** and **Reviewer #2, Comment 2**.

COMMENT 3. In abstract "Together, our findings suggest that metformin might prevent dementia in patients without type 2 diabetes." I'm unclear about why the study is focusing on non-diabetic patients, as we know that diabetes patients have higher risk of developing AD; which might be more suitable for the study. And why were those participants given metformin if they are not diabetic?

Thank you for this question. We would like to emphasize that our study focused on patients with type 2 diabetes. Towards the end of the abstract, we were suggesting that our results—robustly obtained in two distinct diabetic populations, in the U.S. and the U.K.—might also generalize to the case of non-diabetics. Based on your comment, we understand that this final portion of the abstract could be misleading. We edited the end of the abstract to read: “**Together, our findings suggest that metformin might prevent dementia in patients with type 2 diabetes**” (see p.2). We hope that our updated ending clarifies the significance of our results.

COMMENT 4. The study design is to compare the effects of metformin vs sulfonylureas on the risk of death and dementia. It is unclear why selecting sulfonylureas, and would it be better to include a placebo group (as a control) or a group treated with anti-AD drugs?

Thank you for this question. To assess the comparative effectiveness of metformin on prolonging cognitive health and longevity among incident type 2 diabetics who are cognitively normal, two options could be considered: metformin initiators would be compared either to a placebo group (i.e., patients who are not taking any antidiabetic treatment) or to a control group (i.e., type 2 diabetic patients initiating another drug, with a different mechanism of action). Given the target population of interest (i.e., incident type 2 diabetics), using a placebo group is not appropriate. Indeed, type 2 diabetes is—beyond age—the main risk factor for dementia. Even if we were to match patients based on age, sex, and other available sociodemographic characteristics, comparing metformin-treated diabetic with untreated diabetic patients would be confounded by the effects of diabetes on the outcome of interest, i.e., dementia onset. Similarly, comparing metformin-treated diabetic with non-diabetic patients would be confounded by the effects of diabetes on the dementia onset outcome.

Although a considerable portion of the U.S. and the U.K. population aged 50 or more and suffering from hyperglycemia remains undiagnosed, our study involves two clinical samples taken from EHR data where we only get to observe patients with a recorded diagnosis or prescription. For all the aforementioned reasons, investigators on our team came to an agreement that a non-placebo control drug was needed to compare metformin against. To determine the best control drug, we explored the history of antidiabetic drug prescription both in the U.S. and the U.K. In particular, we considered the change in the guidelines formulated by the American Diabetes Association in 2005-2007 [28,29]. To mitigate distributional shifts that could arise in the U.S. RPDR, we thus selected only patients who received their first prescription in 2007 or later. In both the U.S. and the U.K., prior literature reporting about the mix of antidiabetic drugs being prescribed established that metformin ranked first and sulfonylureas second among treatment initiation strategies [30,31,32]. A bar chart characterizing the number of new prescriptions of metformin and sulfonylureas every calendar year analyzed in our study is available in the **Supplementary Material** for each of the considered cohorts (**Extended Data Figs. 8 and 9; see pp.49-50**).

We hope this response helps justify our rationale for using type 2 diabetic patients initiating sulfonylureas as the control group.

COMMENT 5. Previous beneficial effect of metformin was only observed in younger African Americans (see reference 10 and 11). But how is the effect in the current study of AA? and most participants are whites, instead of AA in the current study.

Thank you for this question. As noted, most patients who are represented in the U.S. RPDR and U.K. CPRD study cohorts are White and non-Hispanic (U.S. RPDR: 79%; U.K. CPRD: 87% among the 35% of patients with available race/ethnicity information).

U.S. RPDR

Based on this suggestion, we investigated the distribution of the three outcomes of interest (all-cause mortality, dementia onset, death without dementia) in the Black/African American patient subgroup, representing 7.3% of the cohort. Further, we characterized the distribution of outcomes among younger Black/African American patients (i.e., aged less than 70 at treatment initiation), representing 74% of the Black/African American patients. In the U.S. RPDR cohort, Black/African American patients thus represented 8.7% of those aged less than 70 at treatment initiation. However, the low number of death events recorded among patients initiating sulfonylureas (i.e., the control group) precluded us from pursuing full subset analyses, including the estimation of race/ethnicity-specific hazard ratios, due to a lack of power. For details, see sections A and B below.

U.K. CPRD

For the three outcomes of interest, we also considered conducting a subset analysis among Black patients in the U.K. CPRD, representing 3% of those with available race/ethnicity information. However, given the high level of missingness affecting the race/ethnicity variables in that database, we believe it would be inappropriate to proceed with a full subset analysis, including the estimation of race/ethnicity-specific hazard ratios, due to the risk of selection bias leading to erroneous findings. Of note, among Black patients, 77% initiated before age 70. In the U.K. CPRD, Black patients thus represented 3.3% of those aged less than 70 at treatment initiation with available race/ethnicity information. For details, see sections A and B below.

A. Characterization of unadjusted outcomes among Black patients

U.S. RPDR

- In the U.S. RPDR cohort, there were 966 Black/African American patients (7.3%). A total of 31 death events (3.2%) were recorded in that patient subgroup. Further, a total of 21 death events without dementia (2.2%) and 79 dementia onset events (8.2%) were recorded. Among Black/African American patients, 857 (88.7%) initiated metformin, while 109 (11.3%) initiated sulfonylureas.

- Among Black/African American patients initiating metformin, 27 (3.2%) had a death event. Further, 19 died without experiencing dementia (2.2%) and 67 had a recorded dementia onset (7.8%).
- Among Black/African American patients initiating sulfonylureas, 4 (3.7%) had a death event. Further, 2 died without experiencing dementia (1.8%) and 12 had a recorded dementia onset (11.0%).

All-cause mortality and competing risk analyses in the Black/African American patient subgroup

Per the aforementioned recommendation, we attempted to conduct a subset analysis among Black/African American patients for the three outcomes of interest. However, given the low number of death events among patients initiating sulfonylureas (2 deaths without dementia and 4 deaths overall, out of 102 patients), we were underpowered to formulate any definitive conclusions about the potential benefits of metformin initiation on all-cause mortality, dementia onset, or death without dementia among Black patients.

U.K. CPRD

Of note, race/ethnicity information could only be found for 38,216 patients (35%) of the U.K. CPRD cohort, due to a high missingness rate for this sociodemographic variable. Thus, we decided not to proceed with the analysis as the results would suffer from selection bias. We reported the distribution of outcomes in Black patients for whom race/ethnicity information was available in the EHR.

All-cause mortality and competing risk analyses in the Black patient subgroup

Per the aforementioned recommendation, we considered conducting a full subset analysis among Black patients for the three outcomes of interest. However, given the high level of missingness affecting the race/ethnicity variable, we believe it would be inappropriate to proceed due to the risk of selection bias leading to erroneous and uninterpretable findings.

B. Characterization of unadjusted outcomes among Black patients aged less than 70 at treatment initiation

Similarly, we did not pursue all-cause mortality and competing risk modeling in the subgroup of Black patients aged less than 70 at treatment initiation, either due to an insufficient number of events (U.S. RPDR) or to a high missingness rate affecting the race/ethnicity variable (U.K. CPRD). Nevertheless, we provide the characterization of outcome distribution in that patient subgroup for reference, as it supported our decision not to proceed with the full, adjusted subset analysis.

U.S. RPDR cohort

- In the U.S. RPDR cohort, 711 Black/African American patients initiated at age 70 or before (74% of Black/African American patients). A total of 18 death events were recorded in that patient subgroup. Further, a total of 14 death events

without dementia (2.0%) and 45 dementia onset events (6.3%) were recorded. Among Black/African American patients starting treatment at age 70 or before, 636 (89.5%) initiated metformin, while 75 (10.5%) initiated sulfonylureas.

- Among Black/African American patients initiating metformin at age 70 or before, 16 (2.5%) had a death event. Further, 13 died without experiencing dementia (2.0%) and 39 had a recorded dementia onset (6.1%).
- Among Black/African American patients initiating sulfonylureas at age 70 or before, 2 (2.7%) had a death event. Further, 1 died without experiencing dementia (1.3%) and 6 had a recorded dementia onset (8.0%).

U.K. CPRD cohort

Of note, race/ethnicity information could only be found for 26,789 patients (37.6%) aged less than 70 at treatment initiation in the U.K. CPRD cohort, due to a high missingness rate for this sociodemographic variable. Thus, we decided not to proceed with the distribution of outcomes in patients initiating at age 70 or before for whom race/ethnicity information was available in the EHR and who were identified as Black, as the results would likely suffer from selection bias.

COMMENT 6. Some sentences are confusing, eg. page 4 paragraph 3: "As in the US cohort, metformin initiators were younger than their sulfonylurea counterparts (Table 2)." Does it mean those treated with metformin?

Thank you for this remark. Your interpretation is correct. To further clarify this sentence, we rewrote it as follows: "**As in the U.S. RPDR cohort, we also found in the U.K. CPRD cohort that patients treated with metformin were younger than patients treated with sulfonylureas.**" (see p.4).

COMMENT 7. Figure 5d: It is needed to have detailed methods to describe characteristics of samples, sample numbers and how the proteins were measured? Also, how about APOE level in the brain samples?

Thank you for this suggestion. We have now cited a detailed description of the characteristics of samples from the ROSMAP [33] and Mt. Sinai Brain Bank (MSBB) AMP-AD [34] databases in the **Supplementary Material (Part VII. titled *Systems pharmacological analysis of metformin and sulfonylureas in human neural cells*): "Gene expression profiles of postmortem brain specimens, along with the corresponding clinical annotations, were downloaded from the Synapse portal at www.synapse.org/AMPAD and conducted using www.alzdatalens.org [35]. The methodology used to measure and harmonize RNA-seq values across brain samples has been previously described [36] and can be found at <https://github.com/Sage-Bionetworks/amp-rnaseq>"** (see p.42).

Of note, in this work, we are analyzing RNA levels (not protein levels). In bulk RNA-seq data, we found using AlzDataLens [35] that APOE levels were not changed in AD cases vs control samples, neither in the ROSMAP nor in the MSBB dataset. However, in single-nucleus RNA-seq data, we found that APOE expression was significantly lower

in astrocytes and significantly higher in the microglia in AD cases vs control samples [37,38].

COMMENT 8. Table 2: How many of them are diabetes? Or hyperglycemia?

Thank you for these questions. Importantly, all patients considered in the study were taking an antidiabetic drug at baseline (either metformin or one medication in the sulfonylureas drug class) that was prescribed for type 2 diabetes and/or hyperglycemia. Of note, patients who received polytherapy at baseline or monotherapy other than metformin or sulfonylurea were excluded (**Figs. 1a and 1b; see pp.19-20**). In the U.S. RPDR cohort, 11,229 (85%) patients initiated with metformin, while 1,962 (15%) patients are initiated with a sulfonylurea. In the U.K. CPRD cohort, 94,208 (87%) patients initiated with metformin, while 13,817 (13%) patients initiated with a sulfonylurea. We also characterized the distribution of glycosylated hemoglobin (HbA1C) levels at baseline in each patient group. See **Table 2 (p.22)** for more details about the composition of the two patient cohorts.

COMMENT 9. Figure 2: Survival can be affected by many other factors: eg. other drugs, and treatments; eg. if there are some surgeries for cancers/CVD. And there is no placebo group.

Thank you for this remark. For a detailed discussion about prior disease history, including CVD, cancer, and other comorbidities, please refer to our detailed response to **Reviewer #2, Comment 1**. For an explanation about the choice of sulfonylureas as the control group, see our response to **Reviewer #2, Comment 4**.

As for prior treatments and the concurrent prescription of other drugs, we carefully excluded patients with chronic kidney disease (CKD) at treatment Initiation from our study cohorts, since CKD is a contraindication for metformin but not for sulfonylureas (**Extended Data Table 1; see pp.20-21 for definitions**). Additionally, we characterized patients taking hypertensive medications, given that it may confound the relationship between antidiabetic treatment with metformin and outcomes. In a confirmatory study, e.g., a RCT, the inclusion of other drugs may help refine the estimation of metformin's treatment effect; however, through our discussions with subject matter experts in endocrinology (Alexander Soukas), neurology (Mark Albers, Lefkos Middleton, Sudeshna Das), as well as geriatric epidemiology and psychiatry (Deborah Blacker), we did not identify any other treatments as potential confounders that could affect the generation of robust repurposing hypotheses—the main object of our research.

In short:

- Baseline confounding factors – As explained in the **Supplementary Material (see pp.32-33)**, we thoroughly addressed all known baseline confounders.
- Post-baseline confounding factors – Given that our study is an intention-to-treat analysis, there is no need to account for post-baseline confounding factors. Due to the risk of reverse causation, it would actually be inappropriate, in an

intention-to-treat setup, to define and use post-baseline confounding factors as part of the emulated target trial framework.

- Absence of a placebo group – There is no relevant placebo (i.e., untreated) group among individuals with type 2 diabetes.

COMMENT 10. Figure 3: Many other factors will affect dementia onset, eg. genetics (APOE status), other diseases (eg. diabetes and CVDs). The effect observed here could be compounded by other factors.

Thank you for this remark. We share your concern about other known and unknown factors that could impact the timing of dementia onset and death in observational studies. This is the rationale for adopting the target trial methodology to balance at least the known factors between the two treatment groups.

To specifically address the important factors mentioned above:

- Diabetes – All patients included in this study were incident type 2 diabetic.
- CVD – We had identified CVD as a risk factor and considered this variable in our analyses. Specifically, prior history of CVD was included as a binary covariate in the propensity score models. As reported in **Table 2 (see p.21)**, prior history of CVD affected 5,679 patients, i.e., 43.1% of the U.S. RPDR cohort (42.8% and 44.4% of metformin and sulfonylurea initiators, respectively) and 53.5% of U.K. CPRD patients (53.8% and 50.8% of metformin and sulfonylurea initiators, respectively). In the U.K. CPRD, metformin initiators thus included more CVD cases at baseline than sulfonylurea initiators, while the opposite pattern was observed in the U.S. RPDR. Of note, history of stroke at baseline was relatively rare in the U.K. CPRD (7%) and was therefore combined with CVD in a single covariate for simplicity.
- Genetics – Please refer to our responses to **Reviewer #2, Comments 1 and 2** as well as to our follow-up with the comments provided by **Reviewer #1**.

Notably, we considered all available covariates that potentially influenced both the treatment assignment and one or both outcomes, or strongly related to at least one of the outcomes: dementia or death [47,48 – 23.24]. For this, we leveraged the expertise of senior collaborators in endocrinology (Alexander Soukas), neurology (Mark Albers, Lefkos Middleton, Sudeshna Das), as well as geriatric epidemiology and psychiatry (Deborah Blacker). Considered confounders include prior history of cancer, smoking, and stroke, among other comorbidities, as well as sociodemographic factors such as race and SES and diabetes characteristics. Our process for covariate selection is described in detail in the **Supplementary Material (see pp.31-32)**. The full covariate distribution results, stratified by treatment at initiation for each of the two considered EHR databases, are available in **Table 2 (see p.21)**.

COMMENT 11. It is nice to have an in vitro system to test the effect of metformin in human neural cells. But it is not well established about the link between metformin and dementia.

We agree that an *in-vitro* system to test the effect of metformin in differentiated human neural cells is a powerful tool to generate candidate biomarkers of metformin's action in the human brain for further validation in a clinical trial. Further, we share the perspective that a RCT will be the strongest evidence to establish a link between metformin and dementia onset. In this work, we substantially strengthen the evidence that testing metformin in cognitively normal seniors with dementia onset as the primary outcome is a clinical trial worth pursuing. In two target trial emulation studies (in the U.S. RPDR and U.K. CPRD cohorts), we have demonstrated a robust reduction in the risk of dementia onset among patients initiating metformin, thereby confirming prior findings [20,22,39]. Further, our target trial emulation suggested that the protective effect of metformin vs sulfonylurea was not solely explained by better glucose homeostasis.

Prevention RCTs of clinical dementia onset are very expensive due to slow disease progression requiring a longer study horizon with greater than 1500 enrolled patients. Since metformin has adequate blood-brain barrier penetration, biomarkers that could be used for patient selection and as pharmacodynamic responses to metformin's action in the human brain would facilitate the execution and interpretation of a prevention RCT. In that context, our *in-vitro* systems pharmacology work in human neural cells is an exploratory approach to identify actionable CSF biomarkers implicated in the putative action of metformin in the human brain. This analysis revealed two biologically plausible proteins that have been implicated in the pathogenesis of AD (APOE and SPP1). We would thus recommend their use as secondary or exploratory endpoints in a RCT of metformin to prevent dementia onset.

Reviewer #3

COMMENT 1. This is a solid analysis using inverse propensity weighting. We are very grateful for the Reviewer's encouraging words and thoughtful feedback. Thank you!

COMMENT 2. I dislike the suggestion that the analysis is replicating a randomized trial. That must be removed.

Thank you for this comment. The current version of the manuscript does not suggest that our analysis replicates a clinical trial.

In our study, we built upon Hernan and Robins' methods and guidelines for target trial emulation [40] as well as Labrecque and Swanson's recommendations [41]. Specifically, we conducted two target trial emulations of metformin vs sulfonylureas among incident type 2 diabetic patients. Hernan and Robins define a target trial as a hypothetical randomized trial to be conducted to answer a causal question, e.g., determining which existing treatments may be beneficial (i.e., identifying potential drug repurposing candidates) and which may be harmful (i.e., estimating the toxicity of

select compounds). A causal analysis of observational data can thus be viewed as an attempt to emulate some target trial. Prior to Hernan, Robins, Labrecque, and Swanson, the idea of target trial had been suggested more or less explicitly by several authors, including Dorn, Cochran, Rubin, Feinstein, and Dawid. Initially developed for simple settings with a time-fixed treatment and a single eligibility point, it was then generalized by Hernan and Robins in 2016 to time-varying treatments and multiple eligibility points. In a systematic review on rheumatoid arthritis recently published in the BMJ [42], Zhao et al. emphasize that “Target trial emulation is a structured approach for designing observational comparative effectiveness research studies that helps to avoid potential sources of bias”.

A rigorous analysis requires specifying the target trial protocol before emulating it as deviations from the target trial may be the source of bias in observational analyses. Specifically, Hernan and Robins highlight two key components of target trial emulation: the specification of time zero to synchronize the determination of eligibility with the assignment of treatment strategies and adjustment for confounding factors to emulate baseline randomization. This is what we have done and reported in **Table 1** (see pp.20-21) of the manuscript: we used observational data, from the U.S. RPDR and U.K. CPRD EHR databases, to emulate a target trial with eligibility criteria, treatment strategies, outcomes, causal contrast, and analysis plan similar to the randomized trial. Notably, specifying the protocol of the target trial requires detailed knowledge of the considered databases. In our team, Bang Zheng, Bowen Su, Ioanna Tzoulaki, and Lefkos Middleton all have a thorough understanding of the advantages and limitations of the U.K. CPRD database. Similarly, Colin Magdamo, Sudeshna Das, and Mark Albers have several years of experience with the U.S. RPDR database.

Nevertheless, we recognize that residual confounding remains an issue when using observational data: the target trial emulation is typically a compromise between the ideal randomized double-blind, placebo-controlled trial we would really like to conduct and the naive analysis of the observational data. Despite this limitation, we believe the target trial emulation methodology offers a more rigorous approach than commonly-used observational methods that can guide biomedical research through EHR-based data analysis and inference. A number of recent papers successfully leveraged this approach, including a target trial of statin use and dementia risk [8], observational studies related to cardiovascular health [43,44], hypothetical target trials in oncology [45,46] and critical care [47], and comparative effectiveness research on COVID-19 vaccination [48,49,50].

COMMENT 3. I would consider adding the ATET estimator. That is, compare the metformin users to a balanced population not receiving metformin (weights = $(1-p)/p$). I would also consider estimating the population hazard ratio controlling for sex or age (sex and age in the structural model).

Thank you for the suggestion of considering an alternative weighting scheme and of adjusting for sex and age at baseline in the three semiparametric Cox Proportional

Hazards (PH) outcome models presented in this study: death (all-cause mortality analysis) as well as dementia onset and death without dementia (competing risk analysis).

In the paragraphs below, ATE stands for average treatment effect (at the population level), while ATT (or ATET) stands for average treatment effect on the treated.

Based on this recommendation, we conducted the following sensitivity analyses:

1. For the all-cause mortality analysis (**Complementary Tables 1a and 2a**)
 1. Using ATE weights
 1. Without further adjustment for age and sex in the structural model for time-to-death
 2. With further adjustment for age and sex in the structural model for time-to-death
 2. Using ATT weights
 1. Without further adjustment for age and sex in the structural model for time-to-death
 2. With further adjustment for age and sex in the structural model for time-to-death
2. For the competing risk analysis (**Complementary Tables 1b-c and 2b-c**)
 1. Using ATE weights
 1. Without further adjustment for age and sex in the structural model for time-to-death without dementia and time-to-dementia onset
 2. With further adjustment for age and sex in the structural model for time-to-death without dementia and time-to-dementia onset
 2. Using ATT weights
 1. Without further adjustment for age and sex in the structural model for time-to-death without dementia and time-to-dementia onset
 2. With further adjustment for age and sex in the structural model for time-to-death without dementia and time-to-dementia onset

Results of these sensitivity analyses are presented in the two complementary tables below (**Complementary Tables 1a-c**: U.S. RPDR cohort, **Complementary Tables 2a-c**: U.K. CPRD cohort). We also provide a brief interpretation of our results. Given the relevance of the question, we incorporated these additional sensitivity analyses in the **Supplementary Material (see pp.40-42 and Extended Data Tables 5-6)**.

Importantly, similarities between the estimates of the effects of metformin on all three outcomes (i.e., all-cause mortality, dementia onset, death without dementia), at the population level (ATE) and among treated patients (ATT) as well as in both the U.S. RPDR and U.K. CPRD databases, suggest that the prescription of metformin could be beneficial if generalized to a wider population of eligible patients (without contraindications).

Complementary Table 1. U.S. RPDR cohort. Sensitivity analyses with respect to weighting scheme and further covariate adjustment in the structural outcome models.

Complementary Table 1a. All-cause mortality

Weighting scheme / Adjustment scheme	Cox PH without (age, sex)	Cox PH with (age, sex)
ATE weights	HR (ref. = sulf): 0.570 95% CI: (0.480; 0.677)	HR (ref. = sulf): 0.583 95% CI: (0.491; 0.692)
ATT weights	HR (ref. = sulf): 0.569 95% CI: (0.476; 0.680)	HR (ref. = sulf): 0.585 95% CI: (0.490; 0.690)

Complementary Table 1b. Dementia onset

Weighting scheme / Adjustment scheme	Cox PH without (age, sex)	Cox PH with (age, sex)
ATE weights	HR (ref. = sulf): 0.806 95% CI: (0.685; 0.949)	HR (ref. = sulf): 0.826 95% CI: (0.702; 0.971)
ATT weights	HR (ref. = sulf): 0.796 95% CI: (0.672; 0.944)	HR (ref. = sulf): 0.814 95% CI: (0.686; 0.967)

Complementary Table 1c. Death (without dementia)

Weighting scheme / Adjustment scheme	Cox PH without (age, sex)	Cox PH with (age, sex)
ATE weights	HR (ref. = sulf): 0.600 95% CI: (0.490; 0.735)	HR (ref. = sulf): 0.619 95% CI: (0.505; 0.757)
ATT weights	HR (ref. = sulf): 0.597 95% CI: (0.484; 0.736)	HR (ref. = sulf): 0.616 95% CI: (0.499; 0.760)

Interpretation of aforementioned tabulated results in the U.S. RPDR

When using ATE weights

We address the following question: “At the population level, what is the effect of initiating metformin rather than sulfonylureas?”

- All-cause mortality: treatment decreases the hazard times **0.57** (without adjustment), times **0.58** (with adjustment)

- Dementia onset: treatment decreases the cause-specific hazard times **0.81** (without adjustment), times **0.83** (with adjustment for age and sex)
- Death without dementia: treatment decreases the cause-specific hazard times **0.60** (without adjustment), times **0.62** (with adjustment for age and sex)

When using ATT weights

We address the following question: “Among treated patients, what is the effect of initiating metformin rather than sulfonylureas?”

- All-cause mortality: treatment decreases the hazard times **0.60** (without adjustment), times **0.62** (with adjustment)
- Dementia onset: treatment decreases the cause-specific hazard times **0.80** (without adjustment), times **0.81** (with adjustment)
- Death without dementia: treatment decreases the cause-specific hazard times **0.60** (without adjustment), times **0.62** (with adjustment for age and sex)

Complementary Table 2. U.K. CPRD cohort. Sensitivity analyses with respect to weighting scheme and further covariate adjustment in the structural outcome models.

Complementary Table 2a. All-cause mortality

Weighting scheme / Adjustment scheme	Cox PH without (age, sex)	Cox PH with (age, sex)
ATE weights	HR (ref. = sulf): 0.657 95% CI: (0.611; 0.707)	HR (ref. = sulf): 0.664 95% CI: (0.618; 0.711)
ATT weights	HR (ref. = sulf): 0.621 95% CI: (0.570; 0.670)	HR (ref. = sulf): 0.630 95% CI: (0.579; 0.680)

Complementary Table 2b. Dementia onset

Weighting scheme / Adjustment scheme	Cox PH without (age, sex)	Cox PH with (age, sex)
ATE weights	HR (ref. = sulf): 0.863 95% CI: (0.775; 0.951)	HR (ref. = sulf): 0.878 95% CI: (0.789; 0.967)
ATT weights	HR (ref. = sulf): 0.843 95% CI: (0.741; 0.945)	HR (ref. = sulf): 0.877 95% CI: (0.773; 0.981)

Complementary Table 2c. Death (without dementia)

Weighting scheme / Adjustment scheme	Cox PH without (age, sex)	Cox PH with (age, sex)
ATE weights	HR (ref. = sulf): 0.641 95% CI: (0.593; 0.694)	HR (ref. = sulf): 0.648 95% CI: (0.599; 0.700)
ATT weights	HR (ref. = sulf): 0.605 95% CI: (0.552; 0.662)	HR (ref. = sulf): 0.613 95% CI: (0.559; 0.672)

Interpretation of aforementioned tabulated results in the U.K. CPRD

When using ATE weights

We address the following question: “At the population level, what is the effect of initiating metformin rather than sulfonylureas?”

- All-cause mortality: treatment decreases the cause-specific hazard times **0.66** (with or without adjustment for age and sex)
- Dementia onset: treatment decreases the cause-specific hazard times **0.86** (without adjustment), times **0.88** (with adjustment for age and sex)
- Death without dementia: treatment decreases the cause-specific hazard times **0.64** (without adjustment), times **0.65** (with adjustment for age and sex)

When using ATT weights

We address the following question: “Among treated patients, what is the effect of initiating metformin rather than sulfonylureas?”

- All-cause mortality: treatment decreases the hazard times **0.62** (without adjustment), times **0.63** (with adjustment)
- Dementia onset: treatment decreases the cause-specific hazard times **0.84** (without adjustment), times **0.88** (with adjustment)
- Death without dementia: treatment decreases the cause-specific hazard times **0.61** (with or without adjustment for age and sex)

Specific Points

COMMENT 4. “Emulation of baseline randomization. The covariates defined above were used to emulate baseline randomization.” – I would remove the idea that use of weighting emulates a randomized trial. Like methods that model the outcome in terms of exposure and covariates, weighting does require the assumption of no omitted covariates. However it is worth stating that the weighting does achieve or approximate balance with respect to the measured covariates.

Thank you for this comment. As noted, the use of weighting in observational studies requires the assumption of no omitted covariates or no unmeasured confounding. We explicitly stated this assumption (A1) in the **Supplementary Material (Assumptions paragraph of Part VI. titled *Statistical Methods*; see pp.36-37)**. We agree that stating that balance was achieved for measured covariates is valuable. Results of our

covariate balance assessment were presented in **Extended Data Figs. 10 and 11** (see pp.53-54). To complement the original sentence initially provided in the **subsection titled *Assessment of covariate balance between treatment groups*** (i.e., “The achieved balance for age is presented in Extended Data Fig. 10, while overall covariate balance is summarized in Extended Data Fig. 11.”), we added the following comment: **“In both cohorts, the balance between measured confounders in the two groups was achieved by inverse probability of treatment weighting”** (see p.34).

Further, the question of sensitivity to unmeasured confounding could be addressed using E-values. The E-value can be interpreted as the confounding strength capable of shifting the point estimate or the upper bound of the confidence interval towards the null (i.e., HR=1), thereby getting rid of the observed association on average (point estimate) or in the worst case scenario (upper bound of the confidence interval). However, assessing the sensitivity of our findings regarding the effects of metformin on all-cause mortality, dementia onset, and death without dementia would require an extension of the original concept of E-value developed by Mathur and Vanderweele [51] to time-to-event outcomes and in the presence of competing risks. Such methodological improvement is beyond the scope of this manuscript but will be a topic of future research for our group.

COMMENT 5. ATE – please add definition/description of the average treatment effect. In addition, the estimand of interest is a hazard ratio. In this case reference to “average” treatment effect is not quite appropriate. The quantity being interested is the population hazard ratio; the ratio of hazards to compare the counterfactual population if everyone received metformin to the counterfactual population if no one did. The same goes for the sub-distribution hazard ratio.

Thank you for this remark. As it was pointed out, carefully choosing an estimand when weighting in observational studies is “essential for making valid inferences from the analysis of observational data and ensuring results are replicable and useful for practitioners”. This core principle is explained in detail by Greifer and Stuart in a pre-publication that was recently released [52].

In this study, our estimands were **(a)** the population hazard ratios (time-invariant) and **(b)** the population risk differences (time-varying). For each outcome of interest, the ratio of hazards contrasts outcomes in the counterfactual population in which every patient initiates metformin with the counterfactual population in which every patient initiates sulfonylureas. In the Results section, we defined the average treatment effect (ATE) as **“the difference between risk functions corresponding to two potential outcomes”** (see p.5). Further, in the **Supplementary Material (*Measures of treatment effect paragraph of Part VI. titled *Statistical Methods*; see pp.37-38*)**, we provided a mathematical definition of the risk difference function (RD) associated with a given outcome, i.e., the difference between the counterfactual cumulative incidence function under treatment and the counterfactual cumulative incidence function under control,

evaluated at time t . This definition is valid for each competing outcome, i.e., dementia onset ($k=1$) and death without dementia ($k=2$).

$$RD(t, k) = E[I(T_1 \leq t, E_1 = k)] - E[I(T_0 \leq t, E_0 = k)] = CIF_1(t, k) - CIF_0(t, k), \text{ for } k = 1, 2$$

Thus, $RD(t, k)$ is the average treatment effect (ATE) of metformin, relative to sulfonylureas, on the risk of experiencing a given outcome ($k=1,2$) by time t . The time horizon considered in our study ranges from treatment initiation ($t=0$) to the maximum follow-up time ($t=12$ years in the U.S. RPDR and $t=16$ years in the U.K. CPRD). Notably, we chose the risk difference as a summary of treatment effect, but other options, e.g., risk ratios, could be considered too.

In the main analyses, our estimators were in turn obtained through IPTW, using stabilized ATE weights, followed by semiparametric structural outcome models (Cox Proportional Hazards) without further covariate adjustment. In sensitivity analyses, we also considered nonparametric outcome models (one for each treatment-outcome pair) and derived the ratio of hazard functions

$HR_k(t) = h_{k,1}(t)/h_{k,0}(t)$ (for $k=1,2$) as an additional estimand of interest (**see p.38**), to be compared against the time-invariant population hazard ratio and thereby test for the presence of temporal changes in the treatment effect of metformin vs sulfonylureas.

Taking into consideration this comment, we added a sentence in the manuscript to clearly state the nature of the estimands of interest. It reads as follows: “**In this study, our target estimands of interest are (a) the population hazard ratios (time-invariant) and (b) the population risk differences (time-varying)**” (**see p.35**).

COMMENT 6. “In the estimation of treatment effects in strata of age, of sex and of BMI, we conducted separate analyses in each subgroup of patients and estimated the IPTW weights for each stratum of the covariate.” – Do you mean that for each subgroup, you conducted a separate IPTW analysis, i.e. find a different set of weights for each strata ?

Thank you for this question. Indeed, we conducted a separate analysis within each subgroup. For this, we built a distinct propensity score model within each stratum of interest. Consequently, a given patient may be assigned a certain inverse propensity score of treatment weight (IPTW) as part of the overall study, while receiving another IPTW weight as part of a particular subgroup analysis.

With respect to age, two patient subgroups were considered in the main analysis (≤ 70 and > 70). The corresponding results are presented in **Figs. 2b, 2d, and 3 (see pp.25-26)**. In two additional analyses (**Extended Data Tables 3 and 4; see pp.57-58**), we assessed the sensitivity of our results to this age cutoff (≤ 65 and > 65 ; ≤ 75 and > 75). With respect to BMI, three patient subgroups were considered (< 25 , $25-30$, and ≥ 30). We also studied male and female patients separately. The corresponding results

are presented in **Figs. 2b, 2d, and 3** (see pp.25-26). When studying age-specific strata, the age variable was removed from the propensity score model. Similarly, the BMI and sex variables were removed from the propensity score model when studying BMI-specific and sex-specific strata, respectively.

COMMENT 7. It is not clear why there is statistical analysis described before the section titled “Statistical Analysis”

Thank you for pointing this out. To address this remark, we renamed **Part VI** of the Supplementary Material “**Statistical methods**” (instead of “Statistical analyses”). In contrast, **Part V** focuses on content related to “**Covariates**”, including confounder selection, emulation of baseline randomization, assessment of covariate balance between treatment groups, and covariate missingness.

COMMENT 8. “time-varying causal survival curves” – survival curves are by definition functions of time”

Thank you for this note. We agree with this remark and have incorporated the suggestion, i.e., removing the “time-varying” qualifier in front of “causal survival curves”. Of note, our intent was simply to clarify that survival curves, cumulative incidence functions (CIFs), and risk differences (RDs) are all time-varying quantities, while hazard ratios and cause-specific hazard ratios are static aggregate measures.

COMMENT 9. “In our emulations of a target trial, we used the two following measures of treatment effect” – I find this language disingenuous

We regret that the terminologies used in the manuscript may have been perceived as disingenuous. We hope that our detailed response to **Reviewer #3, Comment 2** provided a strong rationale for the appropriate use of a target trial emulation approach in our study, motivated by the foundational paper of Hernan and Robins published in 2016 [40] and a growing literature since that date—yet predominantly in cancer research. In addition, we hope that the clarification of estimands and estimators considered in this study provided in response to **Reviewer #3, Comment 5**, including a precise definition of ATE along with its mathematical formulation, are helpful.

COMMENT 10. The statistical analysis section is too long as is. Most of it should be in a supplement.

Thank you for this remark. Of note, the *Statistical analyses* section was already part of the Supplementary Material. To account for **Reviewer #3, Comment 7**, we renamed Part VI of the Supplementary Material **Statistical methods** (instead of *Statistical analyses*).

REFERENCES.

1. Altmann A, Tian L, Henderson VW, Greicius MD; Alzheimer's Disease Neuroimaging Initiative Investigators. Sex modifies the APOE-related risk of developing Alzheimer disease. *Ann Neurol*. 2014 Apr;75(4):563-73. doi: 10.1002/ana.24135. Epub 2014 Apr 14. PMID: 24623176; PMCID: PMC4117990.

- 2.** Ford E, Starlinger J, Rooney P, Oliver S, Banerjee S, van Marwijk H, Cassell J. Could dementia be detected from UK primary care patients' records by simple automated methods earlier than by the treating physician? A retrospective case-control study. *Wellcome Open Res.* 2020 Jun 8;5:120. doi: 10.12688/wellcomeopenres.15903.1. PMID: 32766457; PMCID: PMC7385545.
- 3.** Liu, R., Rizzo, S., Whipple, S. et al. Evaluating eligibility criteria of oncology trials using real-world data and AI. *Nature* 592, 629–633 (2021). <https://doi.org/10.1038/s41586-021-03430-5>
- 4.** Dickerman BA, García-Albéniz X, Logan RW, Denaxas S, Hernán MA. Emulating a target trial in case-control designs: an application to statins and colorectal cancer. *Int J Epidemiol.* 2020 Oct 1;49(5):1637-1646. doi: 10.1093/ije/dyaa144. PMID: 32989456; PMCID: PMC7746409.
- 5.** Dickerman BA, García-Albéniz X, Logan RW, Denaxas S, Hernán MA. Avoidable flaws in observational analyses: an application to statins and cancer. *Nat Med.* 2019 Oct;25(10):1601-1606. doi: 10.1038/s41591-019-0597-x. Epub 2019 Oct 7. PMID: 31591592; PMCID: PMC7076561.
- 6.** Wu Y, Warner JL, Wang L, Jiang M, Xu J, Chen Q, Nian H, Dai Q, Du X, Yang P, Denny JC, Liu H, Xu H. Discovery of Noncancer Drug Effects on Survival in Electronic Health Records of Patients With Cancer: A New Paradigm for Drug Repurposing. *JCO Clin Cancer Inform.* 2019 May;3:1-9. doi: 10.1200/CCI.19.00001. PMID: 31141421; PMCID: PMC6693869.
- 7.** Xu H, Aldrich MC, Chen Q, Liu H, Peterson NB, Dai Q, Levy M, Shah A, Han X, Ruan X, Jiang M, Li Y, Julien JS, Warner J, Friedman C, Roden DM, Denny JC. Validating drug repurposing signals using electronic health records: a case study of metformin associated with reduced cancer mortality. *J Am Med Inform Assoc.* 2015 Jan;22(1):179-91. doi: 10.1136/amiajnl-2014-002649. Epub 2014 Jul 22. PMID: 25053577; PMCID: PMC4433365.
- 8.** Caniglia EC, Rojas-Saunero LP, Hilal S, Licher S, Logan R, Stricker B, Ikram MA, Swanson SA. Emulating a target trial of statin use and risk of dementia using cohort data. *Neurology.* 2020 Sep 8;95(10):e1322-e1332. doi: 10.1212/WNL.0000000000010433. Epub 2020 Aug 4. PMID: 32753444; PMCID: PMC7538212.
- 9.** Cummings J, Lee G, Zhong K, Fonseca J, Taghva K. Alzheimer's disease drug development pipeline: 2021. *Alzheimers Dement (N Y).* 2021 May 25;7(1):e12179. doi: 10.1002/trc2.12179. PMID: 34095440; PMCID: PMC8145448.
- 10.** Zhang Z, Zhou L, Xie N, Nice EC, Zhang T, Cui Y, Huang C. Overcoming cancer therapeutic bottleneck by drug repurposing. *Signal Transduct Target Ther.* 2020 Jul 2;5(1):113. doi: 10.1038/s41392-020-00213-8. PMID: 32616710; PMCID: PMC7331117.
- 11.** Park K. The use of real-world data in drug repurposing. *Transl Clin Pharmacol.* 2021 Sep;29(3):117-124. doi: 10.12793/tcp.2021.29.e18. Epub 2021 Sep 27. PMID: 34621704; PMCID: PMC8492393.
- 12.** Fry A, Littlejohns TJ, Sudlow C, Doherty N, Adamska L, Sprosen T, Collins R, Allen NE. Comparison of Sociodemographic and Health-Related Characteristics of UK Biobank Participants With Those of the General Population. *Am J Epidemiol.* 2017 Nov 1;186(9):1026-1034. doi: 10.1093/aje/kwx246. PMID: 28641372; PMCID: PMC5860371.

- 13.** Hathcock MA, Kirt C, Ryu E, Bublitz J, Gupta R, Wang L, Thibodeau SN, Larson NL, Cicek MS, Cerhan JR, Olson JE. Characteristics Associated With Recruitment and Re-contact in Mayo Clinic Biobank. *Front Public Health*. 2020 Feb 4;8:9. doi: 10.3389/fpubh.2020.00009. PMID: 32117849; PMCID: PMC7010638.
- 14.** Ridgeway JL, Han LC, Olson JE, Lackore KA, Koenig BA, Beebe TJ, Ziegenfuss JY. Potential bias in the bank: what distinguishes refusers, nonresponders and participants in a clinic-based biobank? *Public Health Genomics*. 2013;16(3):118-26. doi: 10.1159/000349924. Epub 2013 Apr 12. PMID: 23595106; PMCID: PMC3821039.
- 15.** Cohn EG, Husamudeen M, Larson EL, Williams JK. Increasing participation in genomic research and biobanking through community-based capacity building. *J Genet Couns*. 2015 Jun;24(3):491-502. doi: 10.1007/s10897-014-9768-6. Epub 2014 Sep 18. PMID: 25228357; PMCID: PMC4815899.
- 16.** Broekstra R, Aris-Meijer J, Maeckelberghe E, Stolk R, Otten S. Demographic and prosocial intrapersonal characteristics of biobank participants and refusers: the findings of a survey in the Netherlands. *Eur J Hum Genet*. 2021 Jan;29(1):11-19. doi: 10.1038/s41431-020-0701-1. Epub 2020 Jul 31. PMID: 32737438; PMCID: PMC7852517.
- 17.** Campbell JM, Bellman SM, Stephenson MD, Lisy K. Metformin reduces all-cause mortality and diseases of ageing independent of its effect on diabetes control: A systematic review and meta-analysis. *Ageing Res Rev*. 2017 Nov;40:31-44. doi: 10.1016/j.arr.2017.08.003. Epub 2017 Aug 10. PMID: 28802803.
- 18.** Bannister CA, Holden SE, Jenkins-Jones S, Morgan CL, Halcox JP, Schernthaner G, Mukherjee J, Currie CJ. Can people with type 2 diabetes live longer than those without? A comparison of mortality in people initiated with metformin or sulphonylurea monotherapy and matched, non-diabetic controls. *Diabetes Obes Metab*. 2014 Nov;16(11):1165-73. doi: 10.1111/dom.12354. Epub 2014 Jul 31. PMID: 25041462.
- 19.** Andersen PK, Borgan O, Gill RD & Keiding N. *Statistical Models Based on Counting Processes*. 1993.
- 20.** Scherrer JF, Morley JE, Salas J, Floyd JS, Farr SA, Dublin S. Association Between Metformin Initiation and Incident Dementia Among African American and White Veterans Health Administration Patients. *Ann Fam Med*. 2019 Jul;17(4):352-362. doi: 10.1370/afm.2415. PMID: 31285213; PMCID: PMC6827650.
- 21.** Scherrer JF, Salas J, Floyd JS, Farr SA, Morley JE, Dublin S. Metformin and Sulfonylurea Use and Risk of Incident Dementia. *Mayo Clin Proc*. 2019 Aug;94(8):1444-1456. doi: 10.1016/j.mayocp.2019.01.004. PMID: 31378227; PMCID: PMC7029783.
- 22.** Imfeld P, Bodmer M, Jick SS, Meier CR. Metformin, other antidiabetic drugs, and risk of Alzheimer's disease: a population-based case-control study. *J Am Geriatr Soc*. 2012 May;60(5):916-21. doi: 10.1111/j.1532-5415.2012.03916.x. Epub 2012 Mar 28. PMID: 22458300.
- 23.** VanderWeele TJ. Principles of confounder selection. *Eur J Epidemiol*. 2019 Mar;34(3):211-219. doi: 10.1007/s10654-019-00494-6. Epub 2019 Mar 6. PMID: 30840181; PMCID: PMC6447501.

- 24.** Lesko CR, Lau B. Bias Due to Confounders for the Exposure-Competing Risk Relationship. *Epidemiology*. 2017 Jan;28(1):20-27. doi: 10.1097/EDE.0000000000000565. PMID: 27748680; PMCID: PMC5489237.
- 25.** Brookhart MA, Schneeweiss S, Rothman KJ, Glynn RJ, Avorn J, Stürmer T. Variable selection for propensity score models. *Am J Epidemiol*. 2006 Jun 15;163(12):1149-56. doi: 10.1093/aje/kwj149. Epub 2006 Apr 19. PMID: 16624967; PMCID: PMC1513192.
- 26.** 2020 Alzheimer's disease facts and figures. *Alzheimers Dement*. 2020 Mar 10. doi: 10.1002/alz.12068. Epub ahead of print. PMID: 32157811.
- 27.** Tedde A, Nacmias B, Ciantelli M, Forleo P, Cellini E, Bagnoli S, Piccini C, Caffarra P, Ghidoni E, Paganini M, Bracco L, Sorbi S. Identification of new presenilin gene mutations in early-onset familial Alzheimer disease. *Arch Neurol*. 2003 Nov;60(11):1541-4. doi: 10.1001/archneur.60.11.1541. PMID: 14623725.
- 28.** Kahn SE, Haffner SM, Heise MA, Herman WH, Holman RR, Jones NP, Kravitz BG, Lachin JM, O'Neill MC, Zinman B, Viberti G; ADOPT Study Group. Glycemic durability of rosiglitazone, metformin, or glyburide monotherapy. *N Engl J Med*. 2006 Dec 7;355(23):2427-43. doi: 10.1056/NEJMoa066224. Epub 2006 Dec 4. Erratum in: *N Engl J Med*. 2007 Mar 29;356(13):1387-8. PMID: 17145742.
- 29.** American Diabetes Association. Standards of medical care in diabetes--2006. *Diabetes Care*. 2006 Jan;29 Suppl 1:S4-42. Erratum in: *Diabetes Care*. 2006 May;29(5):1192. PMID: 16373931
- 30.** Wilkinson S, Douglas I, Stirnadel-Farrant H, Fogarty D, Pokrajac A, Smeeth L, Tomlinson L. Changing use of antidiabetic drugs in the UK: trends in prescribing 2000-2017. *BMJ Open*. 2018 Jul 28;8(7):e022768. doi: 10.1136/bmjopen-2018-022768. PMID: 30056393; PMCID: PMC6067400.
- 31.** Curtis HJ, Dennis JM, Shields BM, Walker AJ, Bacon S, Hattersley AT, Jones AG, Goldacre B. Time trends and geographical variation in prescribing of drugs for diabetes in England from 1998 to 2017. *Diabetes Obes Metab*. 2018 Sep;20(9):2159-2168. doi: 10.1111/dom.13346. Epub 2018 Jun 5. PMID: 29732725; PMCID: PMC6099452.
- 32.** Farmer RE, Beard I, Raza SI, Gollop ND, Patel N, Tebboth A, McGovern AP, Kanumilli N, Ternouth A. Prescribing in Type 2 Diabetes Patients With and Without Cardiovascular Disease History: A Descriptive Analysis in the UK CPRD. *Clin Ther*. 2021 Feb;43(2):320-335. doi: 10.1016/j.clinthera.2020.12.015. Epub 2021 Feb 10. PMID: 33581878.
- 33.** Bennett DA, Schneider JA, Buchman AS, Mendes de Leon C, Bienias JL, Wilson RS. The Rush Memory and Aging Project: study design and baseline characteristics of the study cohort. *Neuroepidemiology*. 2005;25(4):163-75. doi: 10.1159/000087446. Epub 2005 Aug 15. PMID: 16103727.
- 34.** Hodes RJ, Buckholtz N. Accelerating Medicines Partnership: Alzheimer's Disease (AMP-AD) Knowledge Portal Aids Alzheimer's Drug Discovery through Open Data Sharing. *Expert Opin Ther Targets*. 2016;20(4):389-91. doi: 10.1517/14728222.2016.1135132. Epub 2016 Feb 7. PMID: 26853544.
- 35.** Bihlmeyer NA, Merrill E, Lambert Y, Srivastava GP, Clark TW, Hyman BT, Das S. Novel methods for integration and visualization of genomics and genetics data in Alzheimer's disease. *Alzheimers Dement*. 2019 Jun;15(6):788-798. doi: 10.1016/j.jalz.2019.01.011. Epub 2019 Mar 29. PMID: 30935898; PMCID: PMC6664293.

- 36.** Rodriguez S, Hug C, Todorov P, Moret N, Boswell SA, Evans K, Zhou G, Johnson NT, Hyman BT, Sorger PK, Albers MW, Sokolov A. Machine learning identifies candidates for drug repurposing in Alzheimer's disease. *Nat Commun.* 2021 Feb 15;12(1):1033. doi: 10.1038/s41467-021-21330-0. PMID: 33589615; PMCID: PMC7884393.
- 37.** Mathys H, Davila-Velderrain J, Peng Z, Gao F, Mohammadi S, Young JZ, Menon M, He L, Abdurrob F, Jiang X, Martorell AJ, Ransohoff RM, Hafler BP, Bennett DA, Kellis M, Tsai LH. Single-cell transcriptomic analysis of Alzheimer's disease. *Nature.* 2019 Jun;570(7761):332-337. doi: 10.1038/s41586-019-1195-2. Epub 2019 May 1. Erratum in: *Nature.* 2019 Jun 17;; PMID: 31042697; PMCID: PMC6865822.
- 38.** Grubman A, Chew G, Ouyang JF, Sun G, Choo XY, McLean C, Simmons RK, Buckberry S, Vargas-Landin DB, Poppe D, Pflueger J, Lister R, Rackham OJL, Petretto E, Polo JM. A single-cell atlas of entorhinal cortex from individuals with Alzheimer's disease reveals cell-type-specific gene expression regulation. *Nat Neurosci.* 2019 Dec;22(12):2087-2097. doi: 10.1038/s41593-019-0539-4. PMID: 31768052.
- 39.** Orkaby AR, Cho K, Cormack J, Gagnon DR, Driver JA. Metformin vs sulfonylurea use and risk of dementia in US veterans aged ≥ 65 years with diabetes. *Neurology.* 2017 Oct 31;89(18):1877-1885. doi: 10.1212/WNL.0000000000004586. Epub 2017 Sep 27. PMID: 28954880; PMCID: PMC5664297.
- 40.** Hernán MA, Robins JM. Using Big Data to Emulate a Target Trial When a Randomized Trial Is Not Available. *Am J Epidemiol.* 2016 Apr 15;183(8):758-64. doi: 10.1093/aje/kwv254. Epub 2016 Mar 18. PMID: 26994063; PMCID: PMC4832051.
- 41.** Labrecque JA, Swanson SA. Target trial emulation: teaching epidemiology and beyond. *Eur J Epidemiol.* 2017 Jun;32(6):473-475. doi: 10.1007/s10654-017-0293-4. Epub 2017 Aug 2. PMID: 28770358; PMCID: PMC5550532.
- 42.** Zhao SS, Lyu H, Solomon DH, Yoshida K. Improving rheumatoid arthritis comparative effectiveness research through causal inference principles: systematic review using a target trial emulation framework. *Ann Rheum Dis.* 2020 Jul;79(7):883-890. doi: 10.1136/annrheumdis-2020-217200. Epub 2020 May 7. PMID: 32381560; PMCID: PMC8693471.
- 43.** Mei H, Wang J, Ma S. An emulated target trial analysis based on Medicare data suggested non-inferiority of Dabigatran versus Rivaroxaban. *J Clin Epidemiol.* 2021 Nov;139:28-37. doi: 10.1016/j.jclinepi.2021.07.001. Epub 2021 Jul 14. PMID: 34271110.
- 44.** Matthews AA, Szummer K, Dahabreh IJ, Lindahl B, Erlinge D, Feychting M, Jernberg T, Berglund A, Hernán MA. Comparing Effect Estimates in Randomized Trials and Observational Studies From the Same Population: An Application to Percutaneous Coronary Intervention. *J Am Heart Assoc.* 2021 Jun;10(11):e020357. doi: 10.1161/JAHA.120.020357. Epub 2021 May 15. PMID: 33998290; PMCID: PMC8483524.
- 45.** Petit LC, García-Albéniz X, Logan RW, Howlader N, Mariotto AB, Dahabreh IJ, Hernán MA. Estimates of Overall Survival in Patients With Cancer Receiving Different Treatment Regimens: Emulating Hypothetical Target Trials in the Surveillance, Epidemiology, and End Results (SEER)-Medicare Linked Database. *JAMA Netw Open.* 2020 Mar 2;3(3):e200452. doi: 10.1001/jamanetworkopen.2020.0452. Erratum in: *JAMA Netw Open.* 2020 Apr 1;3(4):e204966. PMID: 32134464; PMCID: PMC7059023.

- 46.** Groenwold RHH. Trial Emulation and Real-World Evidence. *JAMA Netw Open*. 2021 Mar 1;4(3):e213845. doi: 10.1001/jamanetworkopen.2021.3845. PMID: 33783521.
- 47.** Admon AJ, Donnelly JP, Casey JD, Janz DR, Russell DW, Joffe AM, Vonderhaar DJ, Dischert KM, Stempek SB, Dargin JM, Rice TW, Iwashyna TJ, Semler MW. Emulating a Novel Clinical Trial Using Existing Observational Data. Predicting Results of the PreVent Study. *Ann Am Thorac Soc*. 2019 Aug;16(8):998-1007. doi: 10.1513/AnnalsATS.201903-241OC. PMID: 31038996; PMCID: PMC6774748.
- 48.** Dagan N, Barda N, Kepten E, Miron O, Perchik S, Katz MA, Hernán MA, Lipsitch M, Reis B, Balicer RD. BNT162b2 mRNA Covid-19 Vaccine in a Nationwide Mass Vaccination Setting. *N Engl J Med*. 2021 Apr 15;384(15):1412-1423. doi: 10.1056/NEJMoa2101765. Epub 2021 Feb 24. PMID: 33626250; PMCID: PMC7944975.
- 49.** Nsanzimana S, Gupta A, Uwizihwe JP, Haggstrom J, Dron L, Arora P, Park JJH. The Need for a Practical Approach to Evaluate the Effectiveness of COVID-19 Vaccines for Low- and Middle-Income Countries. *Am J Trop Med Hyg*. 2021 Jul 16;105(3):561-563. doi: 10.4269/ajtmh.21-0482. PMID: 34270458; PMCID: PMC8592367.
- 50.** Dickerman BA, Gerlovin H, Madenci AL, Kurgansky KE, Ferolito BR, Figueroa Muñiz MJ, Gagnon DR, Gaziano JM, Cho K, Casas JP, Hernán MA. Comparative Effectiveness of BNT162b2 and mRNA-1273 Vaccines in U.S. Veterans. *N Engl J Med*. 2021 Dec 1:NEJMoa2115463. doi: 10.1056/NEJMoa2115463. Epub ahead of print. PMID: 34942066; PMCID: PMC8693691.
- 51.** Mathur MB, Ding P, Riddell CA, VanderWeele TJ. Web Site and R Package for Computing E-values. *Epidemiology*. 2018 Sep;29(5):e45-e47. doi: 10.1097/EDE.0000000000000864. PMID: 29912013; PMCID: PMC6066405.
- 52.** Greifer N, Stuart E. Choosing the Estimand When Matching or Weighting in Observational Studies. 2021. arXiv. Eprint: 2106.10577.

Reviewers' Comments:

Reviewer #1:

Remarks to the Author:

The authors have provided thoughtful responses to the criticisms from me and from other referees.

Metformin is one of the highest profile drugs that has been proposed for repurposing in order to improve various aging-related conditions.

I support advancing the revised paper toward publication in its current form.

Reviewer #2:

Remarks to the Author:

The paper is a re-submission after a rejection of first submission.

The study tested the hypothesis of whether metformin improves survival and reduced dementia risk, relative to sulfonylureas. But the conclusion is that their findings suggest that metformin may prevent dementia onset in type 2 diabetes patients. I think it is inappropriate, as it should mention the beneficial effect is only when compared to sulfonylureas treatment. It's a long manuscript and I feel that many parts of the story is confusing.

In the last sentence of the introduction, "The goal of these emulated trials and in-vitro studies is to inform the design of clinical trials in cognitively healthy non-diabetics by enriching trials with participants most likely to benefit from metformin." I'm confused that since the major findings of the study is about metformin's effect in T2D patients developing dementia; how could it inform the design in non-diabetics?

I somehow agree that metformin might have beneficial effect on dementia risk in T2D, compared to sulfonylureas. But isn't it reported previously? see reference (PMID: 28954880; and PMID: 21297276) that they were using a retrospective cohort study in US and Taiwan with better controls. .it reminds me of what's the novelty of the current study? Also, for SPP1, which was proposed as a pharmacodynamic biomarker, but I think it is probably mainly based on the in vitro RNA-expression data, which is kind of weak.

The results section is too long and not very strait forward for me to understand. Some parts should be moved to methods, and some should be moved to discussion.

In results-"Difference in post-treatment HbA1C levels was not clinically significant for metformin vs. sulfonylurea initiators". The authors suggested that the putative effect of metformin on dementia risk is likely through mechanisms other than the control of blood sugar. This is very confusing, first of all, it should be comparison vs sulfonylureas treatment, and there is not comparison vs. no treatment. In addition, the 3-month effects on HbA1C levels might not be sufficient, if looking at Fig.4 the effect on dementia onset was obvious only from years follow-up, eg 5-12 years in US cohort. Finally, the measurement of HbA1C levels is not the only marker representing glycemic control, how about fasting glycemic level and Glucose tolerance test results; and other medicines may also affect glycemic control, how about the usage of insulin in patients when doing the comparison, does these factors be controlled? if insulin use at baseline was excluded, T2D patients might still use insulin after several years-follow up; which will affect glycemic control, which may still affect dementia onset.

Therefore, In rebuttal letter, the response to reviewer1- 1st question. The authors replied that "the beneficial effects of metformin on cognition are not likely caused by improved glycemic control." I disagree with this conclusion.

In results, and Fig.5c. where is control group, need to see the gene expression difference among

control, metformin and glyburide treatments.

In Discussion last paragraph. the authors mentioned "fourth, a novel, mechanistically relevant CSF biomarker, SPP1, for recruitment into the clinical trial that can also serve as a pharmacodynamic mechanism of action.". Based on the current results (mainly from cell level), I cannot agree that SPP1 is a novel CSF pharmacodynamic biomarker. 1. it is unclear if metformin changes SPP1 in human samples (blood, CSF). 2. it is unclear how long-term treatment affect SPP1 level (currently is just up to 72hours in cells; in real patients, the usage of metformin could be months to years); 3. unclear with how metformin affects SPP1 expression. miss the molecular mechanisms. 4. it is unclear how metformin affects SPP1 in specific cell types eg. microglial (currently is a mixture of cell types), and in the human brain samples. eg. hippocampus. 5. it is unclear of how different dose effect of metformin on SPP1 in human samples.

Figure 1. not clear, too many words, and some numbers are blocked and not clearly visible

Table 1 can be moved to supplemental

Fig.4 Two blue color lines are unclear to me to see the difference, which is the major results of the study. It looks like the effect of Metformin is not very strong in UK cohort, if looking at Fig4d, so after 12 years, Metformin increased the risk dementia onset ? what will be the reasons of such heterogeneity.

In reply to comment 1 of reviewer 2: "more EHR-based drug repositioning studies and reviews focusing on dementia as the outcome of interest are being published [8]." I looked at reference 8, which is actually a prospective study . Rotterdam study. Not EHR

In reply to comment 1 of reviewer 2:"In the scenario that such genetic factors were weakly associated with clinical decision-making about antidiabetic prescription, it would most likely not affect the significance of our results, based on the robustness of multiple sensitivity analyses we conducted" . I disagree with this. I understand that genetic components are not available in EHR data, but they are important for understanding the risk of dementia onset.

In reply to comment 3 of reviewer 2:"Towards the end of the abstract, we were suggesting that our results—robustly obtained in two distinct diabetic populations, in the U.S. and the U.K.—might also generalize to the case of non-diabetics". I disagree that the results can be extended to non-diabetics. pls see my previous comments regarding glycemic control

In reply to comment 4 of reviewer 2: "Even if we were to match patients based on age, sex, and other available sociodemographic characteristics, comparing metformin-treated diabetic with untreated diabetic patients would be confounded by the effects of diabetes on the outcome of interest, i.e., dementia onset". I do not understand this. If the authors want to analyze the effect of Metformin, comparing to untreated/not well treated cases will be a naturally logical design.

Reviewer #3:

Remarks to the Author:

I found the reviewer responses very satisfactory, and thank them for clear explanations of their methods.

Reviewer #4:

Remarks to the Author:

The study seeks to inform trial design in health non-diabetic persons through identify the demographic characteristics and CNS biomarkers for AD of persons most likely to benefit from metformin. They do this by first, measuring the effect of metformin use compared to sulfonylureas on dementia risk among persons with diabetes and second, identifying "genes whose expression is

differentially altered in neural cells with metformin treatment relative to the vehicle and to glyburide, one of the sulfonylureas.”

The combined approach of analysis of the two drugs in real world data and in-vitro systems pharmacology evaluation of both drugs is a strength. However, to inform the goals as stated (inform clinical trials of cognitively healthy, non-diabetic) the assumptions that are made should be explicitly stated as well as the evidence supporting them. For example, is the assumption that the genes whose expression is differentially altered in neural cells with metformin will be true for persons without diabetes? For example, do we need to assume that gene X environmental exposures that accompanied the onset of diabetes in one population and not another does not matter?

The assumptions in the empirical analyses should also be more explicitly described and better explained. For example, in what ways do results from primary v. hospital setting do and do not provide more robust estimates? That is, who are you likely additionally capturing and not by utilizing primary care setting at an academic health care system (e.g. not capturing minoritized populations)?

The assumption that across settings, at medical practice differences, missing data etc. bolsters signal observed in both is again not well-justified or explained. That is, under what circumstances is this true? What is not the case that both have biases that could lead to the same conclusions?

Addressing mortality bias is always important in any observational study of this kind and it has been addressed in different ways in prior study. The potential bias resulting from sample selection into metformin v. sulf. is not solved by this study design. There are multiple differences in observables between metformin and sulf. (e.g. other health conditions) and while some of these may be controlled for in the analysis, the issue are the unobserved and unmeasured factors as in all observational studies that do not use some other quasi experimental design. The use of IPTW in CPH model estimation does not solve the problem of unobservable variable bias.

The assumptions of the PH model should not only be tested results should be reported. What other nonparametric approach was used and what were results?

The use of the word causal is too strong given these limitations.

Is the assumption that if has a dementia diagnosis in the first year, and excluded from sample, this also excludes persons with dementia diagnosis before the study start date? Typically, study design would include a wash out window of no diagnosis of dementia over several years prior to time drug initiation is measured as well as time of drug initiation.

Best practice for identifying dementia diagnosis using ICD9/10 codes include a second diagnosis to exclude rule out diagnoses.

Limitations of EHR records for real world data analysis should be described (e.g. if leave system no longer followed) as well as other limitations (who is recording the information and heterogeneity in EHR). Strengths should also be described.

The large differences in mortality of sulfonylurea users in the US (7.8%) v UK (37.4% die) study and the finding of age differences in the UK and none in the US suggest that the selection may be different into drug use across the US and UK and rather than provide for evidence of robustness, calls into question who is taking sulfonylureas compared to the more common first line treatment of metformin.

Some conclusions may be consistent with the data but are not tested.

Whether the discordant hazard ratio and CIF results in the UK samples were due to the protective effect on both dementia and death is not tested - it may be consistent with the findings, but other things may also have changed over time (e.g. switching across diabetes treatments, onset of other health conditions and medications). Similarly, the conclusion that metformin is especially beneficial for those who initiate at a younger age again provokes the question, who does not initiate metformin and why?

In the discussion, it is not sufficient to only state that previous observational studies were mixed given the heterogeneity of study design. How do the results here compare to the most rigorous of

prior studies? There were others that used a match design.
The conclusions are a bit strong given that causality here too cannot be established.

Response to Reviewer Comments

Manuscript ID: 21-20798

Title: Drug repurposing of metformin for Alzheimer's disease: Causal competing risks framework in medical records and complementary systems pharmacology for biomarker identification

Summary: we highlight three key points raised by reviewers, which we have addressed in this point-by-point response letter.

1. Novelty of our work

In contrast with prior research, our study is the first to deploy a causal inference framework to evaluate metformin's effect on dementia onset, relative to sulfonylureas. We account for death as a competing risk and handle selection into treatment by inverse propensity score of treatment weighting. Death is a competing event that cannot be avoided—even in the setting of a randomized controlled trial. Our approach accounts for its presence and estimates treatment effects, e.g., through risk differences, on both competing outcomes. Additionally, we couple computational and empirical approaches: two EHR-based retrospective analyses were combined with a set of *in-vitro* systems pharmacology experiments that complement the EHR findings by nominating candidate biomarkers of metformin's action in the central nervous system. Our dual approach produced insights that could inform the design of future clinical trials.

2. Role of metformin, beyond glycemic control

The hypothesis motivating our study is that metformin improves survival and reduces dementia risk by a mechanism beyond glycemic control, such as through its purported anti-aging mechanisms. Our time series analyses of patient-specific HbA1C levels in each antidiabetic treatment group, following drug initiation, confirms that the role of metformin in delaying dementia onset cannot be solely explained by a better control of glycemia.

3. Genetic factors and confounding

Given that a patient's genetic information has generally not been available to their doctor in the past, genetic factors would not play a role in antidiabetic treatment decision-making during the time period covered by our study (2001-2017 in the UK and 2007-2017 in the US). Therefore, genetic components were not likely to influence the prescription of metformin or sulfonylureas at the time, and they did not constitute a confounder that needed to be adjusted for in the propensity score model. However, general practitioners will increasingly have access to the genetic information of their patients going forward and would in part use it in their decision-making. Thus, we agree with the reviewers that genetic factors may be used to predict baseline risk of disease or response to a specific treatment, thereby justifying the need to include them as confounders in the future.

In addition, we describe our experimental settings with precision and comment on potential gene-environment interactions, which cannot be tested in the laboratory. We also elaborate on the following: selection bias, the lack of documentation of race-ethnicity in the EHR, the impact of differential mortality patterns between cohorts, and the strengths and limitations of our study. When needed, we have softened the language to clearly indicate which results were causal and which were only observational.

Reviewers' comments (our responses in blue)

Reviewer #1 (Remarks to the Author):

The authors have provided thoughtful responses to the criticisms from me and from other referees.

Metformin is one of the highest profile drugs that has been proposed for repurposing in order to improve various aging-related conditions.

I support advancing the revised paper toward publication in its current form.

We would like to thank the reviewer for sharing their appreciation of our first response letter.

Reviewer #2 (Remarks to the Author):

Comment #1

The paper is a resubmission after a rejection of first submission. The study tested the hypothesis of whether metformin improves survival and reduced dementia risk, relative to sulfonylureas. But the conclusion is that their findings suggest that metformin may prevent dementia onset in type 2 diabetes patients. I think it is inappropriate, as it should mention the beneficial effect is only when compared to sulfonylureas treatment. It's a long manuscript and I feel that many parts of the story are confusing.

Author Reply

Per your recommendation, we re-emphasized throughout the manuscript that the beneficial effect of metformin monotherapy (treatment) estimated in the EHR-based observational studies presented here was relative to treatment initiation with sulfonylureas (control).

For example, on **Pages 7-8**, we added the following: *“Nevertheless, the age-specific finding in the US RPDR cohort suggests that metformin may be especially beneficial – **relative to sulfonylureas** – for those who initiate treatment at a younger age.”*

Similarly, on **Page 9**, we incorporated mentions of the comparator: *“The beneficial effect of metformin **over sulfonylureas** cannot be solely due to better diabetes management through the control of hyperglycemia, prompting us to investigate alternative modes of action for metformin's beneficial effect in an in vitro human differentiated neural cell system – **in contrast to another antidiabetic.**”*

Comment #2

In the last sentence of the introduction, "The goal of these emulated trials and in-vitro studies is to inform the design of clinical trials in cognitively healthy non-diabetics by enriching trials with participants most likely to benefit from metformin." I'm confused that since the major findings of the study is about metformin's effect in T2D patients developing dementia; how could it inform the design in non-diabetics?

Author Reply

Thank you for this question, which prompted us to clarify our rationale.

The hypothesis motivating our study is that metformin improves survival and reduces dementia risk by a mechanism beyond glycemic control, such as through its purported anti-aging

mechanisms described by many other laboratories. This hypothesis is particularly relevant for dementia since age is the principal risk factor. We demonstrated that the change in HbA1C induced by antidiabetic treatment initiation was similar in both arms (**Extended Data Table 5**), supporting a mechanism beyond glycemic control.

To increase clarity, we rephrased the last sentences of our introduction as follows (**Page 3**):

“Our EHR-based results may serve as an example of the use of real world data (RWD) to inform the design of clinical trial eligibility criteria¹⁵ for a trial of metformin with the primary outcome of dementia onset. Further, our systems pharmacology studies may suggest a pharmacodynamic CSF biomarker for metformin’s anti-aging actions in the human brain beyond its hypoglycemic actions.”

Per your comment, we edited the first paragraph of the discussion to reflect this clarification (**Page 9**).

Comment #3

I somehow agree that metformin might have a beneficial effect on dementia risk in T2D, compared to sulfonylureas. But isn't it reported previously? see reference (PMID: 28954880; and PMID: 21297276) that they were using a retrospective cohort study in US and Taiwan with better controls. It reminds me of what's the novelty of the current study?

Author Reply

Thank you for this question, which motivates us to highlight the novel aspects of our research. We believe our work used novel methodologies to account for the limitations of observational research, which contributes to the increased robustness of our findings. Additionally, we provide a replication in external databases and linked systems pharmacology experiments to support the drug’s mechanisms of action underlying the effects observed at the cohort level. Of note, previous studies cited by the reviewer did not use death as a competing risk. Yet incorporating death as a competing risk is critical in research involving elderly individuals, as documented elsewhere [**1,2,3,4**].

Traditional approaches to describe time-to-event, such as dementia onset, include Cox proportional hazards (PH) regression. Employed in the two studies cited by the reviewer, this model can overestimate the risk of disease when failing to account for the competing risk of death. This artifact is especially problematic in studies of older patients with a substantial number of participants dying during follow-up [**1**]. Previous work demonstrated that studies of absolute risk of disease should instead use methods that account for competing risks (e.g., death). Such a strategy is imperative to avoid over-stating risk for the disease of interest and derive accurate estimates of the effect of differential treatment initiation; only then can clinical decision-making be improved [**5**].

Further, we would like to emphasize that Hsu et al. [**6**] only accounted for a limited number of covariates in their adjusted Cox PH regression—age, gender, CCI score, types of stroke, and antidiabetic treatment type—raising concerns about the potential for residual confounding.

Although Orkaby et al. [**7**] (**ref. 30**) considered a larger set of confounders—similar to ours—and used re-balancing via inverse propensity score of treatment weighting, an important limitation is that the competing risk of death was ignored, potentially biasing treatment effect estimates.

In contrast with prior work, our study is the first to deploy a causal inference approach to evaluate metformin’s effect on dementia onset, relative to sulfonylureas, that both accounts for

death as a competing risk and handles selection into treatment by inverse propensity score of treatment weighting. Death is a competing event that cannot be avoided—even in the setting of a randomized controlled trial. Our approach accounts for its presence and estimates treatment effects, e.g., through risk differences, on both competing outcomes. Additionally, we coupled computational and empirical approaches: two EHR-based retrospective analyses were combined with a set of *in-vitro* systems pharmacology experiments that complement the EHR findings by nominating candidate biomarkers of metformin's action in the central nervous system (CNS).

Per your comment, we now put greater emphasis on the originality of our work in the abstract and discussion. Although metformin has previously been proposed as an anti-aging medication, our manuscript is the first to implement a statistical framework handling confounding and immortal bias using EHR data while deploying *in-vitro* systems pharmacology experiments that validate and contextualize the effect of metformin on brain cells, at pharmacologically relevant drug concentrations.

Because of the dual approach we adopted, we were also able provide to derive insights that inform the design of future clinical trials:

1. Additional criteria for clinical trial recruitment may include lowering the threshold for age at enrollment: our EHR-based target trials helped identify candidate target populations who would be more responsive to the drug (e.g., patients aged less than 70 at antidiabetic monotherapy treatment initiation).

2. New exploratory biomarkers may be investigated in clinical trials to better understand the presymptomatic phase of Alzheimer's disease and the patterns of disease onset (e.g., the levels of SPP1 protein in the cerebrospinal fluid).

We highlight the novelty of our work on **Page 9**: *“To our knowledge, this work is the first attempt to comprehensively address competing death in a study of metformin and dementia, with a rigorous causal framework harmonized across two EHR databases, in incident type 2-diabetic patients.”*

We discuss the strengths of our study in more detail on **Pages 10-11** (see paragraph starting with *“There are many strengths in our DRIAD-EHR approach...”*) and on **Page 12** (*“These robust EHR findings were buttressed by in-vitro analyses of human neural cells at pharmacologically relevant concentrations that revealed novel actions of metformin, relative to glyburide, and to metformin's actions in other cell types.”*).

Comment #4

Also, for SPP1, which was proposed as a pharmacodynamic biomarker, but I think it is probably mainly based on the in vitro RNA-expression data, which is kind of weak.

Author Reply

Thank you, this remark has inspired us to better explain our findings. The mechanism of action of metformin is pleiotropic and likely cell-type dependent. Although metformin's mechanism of action has not been fully delineated and cell-type dependency has yet to be proven, the breadth of metformin's actions in different cell types indicates that a common mechanism of action is not likely. We propose that metformin acts directly on human neural cells, beyond its powerful effects on glycemic control, to modulate secreted proteins in the CSF, which may mediate its *“anti-aging”* action in the context of the brain.

We supplemented our findings, presented in **Figure 5** and **Extended Data Tables 8 and 9** by providing three additional references that SPP1 protein levels are increased in the brains and plasma of patients with Alzheimer's disease [8,9,10] (refs. 31-33).

We agree that the role of SPP1 as a potential pharmacodynamic biomarker, both during the presymptomatic phase of the disease and following disease onset, should be further investigated. That is why we recommend that future clinical trials quantify SPP1 as an exploratory aim in the CSE, before and after metformin exposure, with enrollment of younger patients.

On **Page 10**, we therefore nuanced our recommendation and updated the manuscript as follows: *"The secreted protein SPP1 emerges as a candidate biomarker for metformin's action in the central nervous system (CNS) to be further investigated as an exploratory aim before and after drug exposure."*

Comment #5

The results section is too long and not very straightforward for me to understand. Some parts should be moved to methods, and some should be moved to discussion.

In results-"Difference in post-treatment HbA1C levels was not clinically significant for metformin vs. sulfonylurea initiators". The authors suggested that the putative effect of metformin on dementia risk is likely through mechanisms other than the control of blood sugar. This is very confusing, first of all, it should be comparison vs sulfonylureas treatment, and there is not comparison vs. no treatment. In addition, the 3-month effects on HbA1C levels might not be sufficient, if looking at Fig.4 the effect on dementia onset was obvious only from years follow-up, eg 5-12 years in US cohort. Finally, the measurement of HbA1C levels is not the only marker representing glycemic control, how about fasting glycemic level and Glucose tolerance test results; and other medicines may also affect glycemic control, how about the usage of insulin in patients when doing the comparison, does these factors be controlled? if insulin use at baseline was excluded, T2D patients might still use insulin after several years-follow up; which will affect glycemic control, which may still affect dementia onset.

Author Reply

Metformin was compared to sulfonylureas through a linear mixed effect model. First, we would like to clarify that we did compare the effect of metformin (treatment) vs. sulfonylureas (control). Specifically, we used a linear mixed effect model with a binary variable encoding treatment initiation (metformin: 1, sulfonylureas: 0). Model specifics are provided in **Extended Data Table 7 (Page 61)**.

Repeated measurements, beyond the three-month mark, were considered. In addition, we would like to emphasize that we used a modeling approach handling repeated measurements: for each patient, we considered all HbA1C measurements following treatment initiation—rather than solely the HbA1C level measured three months after initiation. So, we did effectively explore the long-term effects of metformin, beyond the three-month mark. We agree with the reviewer that considering glycemic levels over the full follow-up period is critical, given that the effect of metformin on dementia onset may only appear five years or more after treatment initiation.

HbA1C tests were selected over other available options to assess blood glucose management because of their higher sampling frequency. We appreciate the reviewer's recommendation to explore other markers of glycemic control than HbA1C levels. However, glucose tolerance tests are not routinely conducted and differences in sampling frequency between patients may further contribute to selection bias. Indeed, specific events may trigger the ordering of such tests, including onset of infection or an acute event leading to hospitalization. In contrast, one measurement of HbA1C represents a more stable three-month average glucose level. Further, this test is more commonly conducted in practice. Our intent was to capture variation among type 2 diabetic patients using a test that is routinely conducted at regular intervals and represents a smoothed version of blood sugar levels, rather than a point-in-time snapshot as it is with a fasting glucose level.

Add-on insulin prescriptions during subsequent phases of the disease are possible. We agree that some patients might be prescribed insulin at a later point in time, which might affect the control of their blood glucose levels. However, our analysis is an intention-to-treat analysis, aiming at comparing metformin monotherapy with sulfonylureas monotherapy, as prescribed at the first diagnosis of type 2 diabetes. Evaluating the effect of treatments that change over time goes beyond the scope of this paper. Future work, using a time-varying analysis framework, could explore the role of insulin and other concurrent medications.

Comment #6

Therefore, In rebuttal letter, the response to reviewer1- 1st question. The authors replied that "the beneficial effects of metformin on cognition are not likely caused by improved glycemic control." I disagree with this conclusion.

Author Reply

Thank you for bringing this point to our attention. We would like to provide further clarity about this sentence of the response letter by stating that "the differential beneficial effects of metformin on cognition (i.e., preventing dementia onset), over sulfonylureas, are not likely caused by improved glycemic control induced by metformin (specifically)."

Here, we are not arguing that improved glycemic control does not improve cognition, i.e., limit or reduce the risk of dementia onset, but rather that metformin acts pleiotropically to affect other biological activities, which further enhance cognition, i.e, reduce the risk of dementia onset, beyond glycemic control. This hypothesis will be tested in a clinical trial enrolling non-diabetic patients that our team is poised to launch. In non-diabetics, confounding due to differential glycemic control would not be an issue, since physiological mechanisms for glucose homeostasis are far more precise than oral hypoglycemic therapy. Ultimately, adequately powered clinical trials will determine whether metformin affords cognitive benefits by reducing the risk of dementia onset in non-diabetic participants.

Comment #7

In results, and Fig.5c. where is control group, need to see the gene expression difference among control, metformin and glyburide treatments.

Author Reply

Thank you for pointing out the importance of having a control group. You can refer to **Extended Data Table 8** for all details (see separate attachment as this table is too large to embed in the manuscript file). Specifically, the differences in gene expression levels between drug exposure and DMSO (control), at two time frames (after 24 and 72 hours) and under two pharmacologically relevant drug concentrations (10 and 40 μ M), are available in **Extended Data**

Figure 7 (Page 50), for neural cells exposed to metformin or glyburide. Furthermore, **Extended Data Table 9 (Page 63)** provides the detailed list of genes that encode secreted protein products whose RNA expression levels changed by more than two-fold in response to metformin exposure at 40 μ M for 72 hours, relative to glyburide exposure at the same concentration and duration, in differentiated human neural cells.

Comment #8

In Discussion last paragraph. the authors mentioned "fourth, a novel, mechanistically relevant CSF biomarker, SPP1, for recruitment into the clinical trial that can also serve as a pharmacodynamic mechanism of action.". Based on the current results (mainly from cell level), I cannot agree that SPP1 is a novel CSF pharmacodynamic biomarker. 1. it is unclear if metformin changes SPP1 in human samples (blood, CSF). 2. it is unclear how long-term treatment affect SPP1 level (currently is just up to 72hours in cells; in real patients, the usage of metformin could be months to years); 3. unclear with how metform affects SPP1 expression. miss the molecular mechanisms. 4. it is unclear how metformin affects SPP1 in specific cell types eg. microglial (currently is a mixture of cell types), and in the human brain samples. eg. hippocampus. 5. it is unclear of how different dose effect of metformin on SPP1 in human samples.

Author Reply

Per your comment, we have modified the cited sentence in the last paragraph of the discussion. It now reads as follows: "Fourth, our in-vitro systems pharmacological studies of metformin and glyburide in cultured human neural cells identified a candidate gene, SPP1/osteopontin, whose expression is reduced significantly more by metformin than by sulfonylureas. Further, the gene product is elevated in the CSF and plasma of patients diagnosed with AD or pre-MCI."

Our point-by-point responses to the reviewer's specific comments can be found below.

Point #1: Prior studies found that SPP1 was elevated in the cerebrospinal fluid of patients with mild cognitive impairment or Alzheimer's disease [8,9,10] (refs. 31-33). We agree that clinical trials in participants initiating treatment with metformin monotherapy, irrespective of their dementia diagnosis, are needed to determine the effect size on SPP1 levels in the CSF.

Point #2: We agree with the reviewer's caution about extrapolating steady-state levels of protein biomarkers in the blood or CSF from *in-vitro* systems pharmacology experiments. Similarly, emulating chronic exposure of human neural cells to the drug of interest is difficult to capture *in vitro*. Importantly, we would like to emphasize that our study uniquely considers drug concentrations – of metformin and glyburide – that are pharmacologically relevant, effectively mimicking the exposure in humans. Only clinical trials that monitor SPP1 levels by longitudinally sampling blood and CSF from patients will reveal whether monotherapy treatment initiation with metformin affects SPP1 levels. Therefore, we strongly encourage the inclusion of SPP1 in such trials, as an exploratory aim.

Point #3: The results of the complementary *in-vitro* systems pharmacology experiments presented in our manuscript suggest the potential role of SPP1 as a biomarker. However, uncovering the molecular mechanisms underpinning the effects of metformin on SPP1 gene expression levels, and therefore its increased or lowered presence in the CSF, goes beyond the scope of the present study and will be the object of future research.

Point #4: In **Figure 5d (Page 29)**, our goal was to show that SPP1 RNA levels different between patients with Alzheimer’s disease and their age-matched counterparts, across four distinct brain regions. Understanding how exactly metformin affects SPP1 protein levels in specific cell types such as microglia and in the specific regions of the human brain such as the hippocampus goes beyond the scope of this study and remains to be determined. Further *in-vitro* systems pharmacology experiments, in tandem with frequent monitoring of SPP1 levels in future clinical trials, should help elucidate the association between treatment with metformin and changes in SPP1 expression levels.

Point #5: We agree that careful consideration of the drug doses and exposure times are important in modeling the actions of metformin and glyburide in human neural cells. The strength of our study is to examine pharmacologically relevant concentrations, ranging from 10 and 40 μ M. Most in vitro studies of metformin use doses (up to 5 mM) that are not achievable in patients. Future work will involve the quantification of the dose-exposure duration-response relationship between metformin and SPP1 levels in blood and CSF in humans, but is out of scope for this study.

Comment #9

Figure 1. not clear, too many words, and some numbers are blocked and not clearly visible.

Author Reply

Thank you for bringing this to our attention. **Figure 1 (Pages 23-24)** are consort diagrams providing details about the two study populations. These diagrams are important in clinical trials and equally important for the presentation of a target trial, as described by Robins and Hernan [11] (ref. 2). We have adjusted the font size to ensure that all numbers are visible.

Comment #10

Table 1 can be moved to supplemental.

Author Reply

Table 1 (Pages 21-22) is critical to the design and execution of an emulated target trial. Thus, we would prefer to include it within the main manuscript. We will ultimately defer to the editor’s recommendation.

Comment #11

Fig.4 Two blue color lines are unclear to me to see the difference, which is the major results of the study. It looks like the effect of Metformin is not very strong in UK cohort, if looking at Fig4d, so after 12 years, Metformin increased the risk dementia onset ?

Author Reply

Both **Figures 4a and 4b (Page 28)** show four cumulative incidence curves all on the same scale to facilitate direct comparison. This choice for rigor reduces the visual impact of the dementia outcome result. **Figures 4c and 4d (Page 28)** show the risk difference and are perhaps more visually interpretable.

In the US RPDR, (the) confidence regions for both outcomes are consistently below 0, over the 12-year observational period. In the UK CPRD, the confidence region for dementia onset is below 0 until the end of year 8, indicative of metformin’s benefits towards cognitive function. For follow-up periods ranging from 9 to 16 years, the confidence region contains 0 and is wider due to a lower patient sample size—affecting statistical estimation of the risk difference. Thus, we cannot conclude about the differential benefits of metformin over sulfonylureas towards

dementia onset beyond year 8 on the sole basis of our target trial emulation in the UK CPRD patient population, thereby justifying the extension of our study framework to other patient cohorts and the deployment of a formal clinical trial to test the drug's purported role.

Comment #12

what will be the reasons of such heterogeneity.

Author Reply

Building upon our response to the above question: beyond the relatively high mortality rate registered in the UK CPRD (**Table 2, Page 23**) and the sizable number of patients lost to follow-up (**Extended Data Fig. 2, Page 45**), reasons for heterogeneous treatment effects in the long run include variation in age at treatment initiation and in drug adherence. Additionally, patients may alter their anti-diabetic treatment regimen and/or receive medications for other diseases.

Comment #13

In reply to comment 1 of reviewer 2: "more EHR-based drug repositioning studies and reviews focusing on dementia as the outcome of interest are being published [8]." I looked at reference 8, which is actually a prospective study. Rotterdam study. Not EHR

Author Reply

Thank you for looking into this related work. While **ref. 8 [12]** is indeed based on a cohort study, it is another emulated target trial, conducted using observational data to estimate the effect of statins on dementia onset. Of note, as described in the paper: "The entire cohort was continuously under surveillance for clinically diagnosed dementia through electronic linkage of the study database with medical records from general practitioners and the regional institute for outpatient mental health care". Specifically: "Incident dementia was ascertained by combining continuous surveillance through electronic linkage with medical records and MMSE and Geriatric Mental Schedule assessments." The study was thus made possible thanks to direct mapping with EHR, which is why we cited it here (see also the reply to **Comment #1 of Reviewer #2**).

Comment #14

In reply to comment 1 of reviewer 2:"In the scenario that such genetic factors were weakly associated with clinical decision-making about antidiabetic prescription, it would most likely not affect the significance of our results, based on the robustness of multiple sensitivity analyses we conducted" . I disagree with this. I understand that genetic components are not available in EHR data, but they are important for understanding the risk of dementia onset.

Author Reply

We agree with your comment that some populations have a higher risk of dementia and that understanding the genetic mechanisms conferring increased risk upon certain patients to develop the onset of dementia is a key question for the field. However, causal competing risk analyses only require adjusting for confounders, i.e., variables *affecting both the treatment the patient is prescribed and the outcomes of interest* (dementia onset and competing death). Confounders are used to re-balance the treatment and control arms of the target trial to emulate randomization. Given that a patient's genetic information has generally not been (made) available to their doctor in the past, genetic factors would not play a role in antidiabetic treatment decision-making during the time period covered by our study (2001-2017 in the UK and 2007-2017 in the US). Therefore, genetic components were not likely to influence the

prescription of metformin or sulfonylureas at the time and did not constitute a confounder that needed to be adjusted for in the propensity score model. Details about the reasoning used to determine confounders appear in Methods **Section V. titled “Covariates”, paragraph “Confounder selection” on Page 33**. However, we agree with the reviewer that general practitioners may increasingly have access to the genetic information of their patients going forward and would in part use it in their decision-making. Thus, we also believe that genetic factors will be used to predict baseline risk of disease or response to a specific treatment, thereby justifying the need to include them as confounders that need to be adjusted for in the future.

Comment #15

In reply to comment 3 of reviewer 2: "Towards the end of the abstract, we were suggesting that our results—robustly obtained in two distinct diabetic populations, in the U.S. and the U.K.—might also generalize to the case of non-diabetics". I disagree that the results can be extended to non-diabetics. pls see my previous comments regarding glycemic control

Author Reply

We agree with the reviewer that we cannot conclude about the potential benefits of metformin among non-diabetic patients, relative to sulfonylureas, solely on the basis of an observational study in type 2 diabetic patients—albeit rigorously conducted using a causal inferential approach that carefully handles selection into treatment and uniquely considers competing risks.

However, because the estimated effect of metformin on HbA1C levels—relative to sulfonylureas’ initiators—was limited and clinically insignificant (point estimate: -0.21, 95% CI: -0.26 – -0.15), the results presented in our study suggest that metformin plays a role beyond glycemic control, (thus) justifying the deployment of a clinical trial in non-diabetic patients, contrasting metformin use with placebo. In prior work solely examining differences in all-cause mortality, Bannister et al. (2014) [13] found that patients initiating with sulfonylureas had markedly reduced survival compared with both matched, non-diabetic controls and those receiving metformin monotherapy, explaining: “*this result implies that metformin may confer benefit in non-diabetic patients*”.

Besides, our findings suggest that these patients should be recruited early and initiate treatment before age 70. Such a trial could involve monitoring of SPP1 levels and other candidate CSF biomarkers. Notably, metformin is already being prescribed off-label to non-diabetic patients by providers in several countries, including the United States and Israel. These existing practices also support the implementation of rigorous clinical trials in the broader aging population.

As explained in our response to **Comment #2** above, we updated the manuscript to clarify the implications of our findings with respect to metformin’s role in the control of glycemia.

Comment #16

In reply to comment 4 of reviewer 2: "Even if we were to match patients based on age, sex, and other available sociodemographic characteristics, comparing metformin-treated diabetic with untreated diabetic patients would be confounded by the effects of diabetes on the outcome of interest, i.e., dementia onset". I do not understand this. If the authors want to analyze the effect of Metformin, comparing to untreated/not well treated cases will be a naturally logical design.

Author Reply

Thank you for this remark, and our apologies for the confusion. We would like to offer a clarification. Patients with a type 2 diabetes diagnosis coded in their EHR are likely to be treated

with a first line antidiabetic medication or more. Conversely, patients suffering from type 2 diabetes who have not yet been diagnosed will neither have a formal diagnosis nor an antidiabetic prescription documented in their medical history, making their identification impractical—if not impossible. Additionally, implementing a proper target trial requires contrasting patients with similar disease severity; untreated patients are likely to suffer from a milder form of disease. In that context, building a control arm with untreated or inappropriately treated cases would be challenging.

We would like to thank Reviewer #2 for their thoughtful comments, prompting us to clarify the hypotheses underlying our study. We hope that our responses better contextualize our research with respect to related work.

Reviewer #3 (Remarks to the Author):

I found the reviewer responses very satisfactory, and thank them for clear explanations of their methods.

We would like to thank the reviewer for highlighting the clarity of our explanations.

Reviewer #4 (Remarks to the Author):

Comment #1

The study seeks to inform trial design in health non-diabetic persons through identify the demographic characteristics and CNS biomarkers for AD of persons most likely to benefit from metformin. They do this by first, measuring the effect of metformin use compared to sulfonylureas on dementia risk among persons with diabetes and second, identifying “genes whose expression is differentially altered in neural cells with metformin treatment relative to the vehicle and to glyburide, one of the sulfonylureas.”

Author Reply

Thank you for summarizing the dual component of our study, which comprises two emulated target trials based on EHR data in tandem with a set of systems pharmacology experiments.

Comment #2

The combined approach of analysis of the two drugs in real world data and in-vitro systems pharmacology evaluation of both drugs is a strength. However, to inform the goals as stated (inform clinical trials of cognitive healthy, non-diabetic) the assumptions that are made should explicitly stated as well as the evidence supporting them. For example, is the assumption that they genes whose expression is differentially altered in neural cells with metformin will be true for persons without diabetes?

Author Reply

Thank you for this question. We would like to clarify that the human neural cells in culture were not from type 2 diabetic patients. Further, they were not grown under hyperglycemic conditions.

Per your recommendation, we added a sentence to the discussion (Page 10) to explicitly state these experimental settings, which allow us to assume that differentially expressed genes in neural cells exposed to metformin should also reflect changes in non-diabetic patients: “The neural cells were not derived from a diabetic patient and were not grown under hyperglycemic conditions, supporting the notion that the observed changes may occur in the CNS of non-diabetic patients.”

Comment #3

For example, do we need to assume that gene X environmental exposures that accompanied the onset of diabetes in one population and not another does not matter?

Author Reply

This is an excellent question that could only be addressed formally via a clinical trial of metformin vs. placebo in non-diabetic patients. Addressing gene-environment interactions goes beyond the scope of our study since **(a)** we cannot study metformin's actions in non-diabetics using EHR data and **(b)** we cannot study gene-environment interactions in cultured human neural cells in the laboratory. Therefore, we would recommend for the evaluation of gene-environment interactions that may have accompanied the onset of diabetes in each population to be the object of a dedicated study.

Per your remark, we updated a sentence in the discussion to acknowledge this limitation: "Furthermore, the strong effect of age, the changes in prescribing patterns of sulfonylureas and metformin over the observation period^{34,35}, **gene-environment interactions**, and the complex differences observed in age at baseline, length of follow-up, and calendar time across the two treatments raises the possibility of residual confounding."

Comment #4

The assumptions in the empirical analyses should also be more explicitly described and better explained. For example, in what ways do results from primary v. hospital setting do and do not provide more robust estimates?

That is, who are you likely additionally capturing and not by utilizing primary care setting at an academic health care system (e.g. not capturing minoritized populations)?

Author Reply

Thank you for highlighting this point. We agree with the reviewer that race-ethnicity should be better captured in the EHR and that minority populations should be better represented in clinical research overall.

US RPDR

The patients in a tertiary academic hospital in the US (RPDR database) are likely to have better access to healthcare and a higher education level on average; however, minority populations may be underrepresented (see **Table 2 on Page 23**; non-White patients: 21.8% and 16.6%; Hispanic patients: 4.4% and 3.3%, among metformin and sulfonylureas initiators, respectively). Despite these limitations, outcome ascertainment made by trained neurologists, psychiatrists, and gerontologists—which is also key to derive robust estimates—is expected to be more consistent and of better quality at an academic healthcare system than in primary care settings in the UK (CPRD database), with substantial variation by practice.

UK RPDR

In contrast, the population represented in primary care records should more closely match the broader UK population and be more inclusive of minority groups, although dementia outcome determination offers more challenges in this setting due to undercoding and general underuse of specific dementia codes.

In sum, each database offers advantages and disadvantages. Despite setting-based differences (primary care vs. academic hospital), our analyses yielded similar hazard ratio estimates for

dementia onset and death without dementia in two patient populations with differing sociodemographic and clinical profiles, thereby bringing additional, robust evidence supporting the moderate but statistically significant protective effect of treatment initiation with metformin on incident dementia.

Comment #5

The assumption that across settings, at medical practice differences, missing data etc. bolsters signal observed in both is again not well-justified or explained. That is, under what circumstances is this true? What is not the case that both have biases that could lead to the same conclusions?

Author Reply

Thank you for alluding to the complexity and multiplicity of biases inherent to the retrospective use of EHR data. We note that missingness patterns encountered in the US RPDR and UK CPRD differed. In the US RPDR, missingness primarily affected two baseline variables: BMI (32%) and HbA1C (38%). As reported in Table 2 (Page 23), we found higher missingness rates among sulfonylureas initiators. In the UK CPRD, missingness affected four variables, albeit to a different extent: BMI (3%), HbA1C (21%), index of multiple deprivation (7%), and smoking status (2%). We accounted for missingness in these variables by considering separate categories (i.e., a binary indicator of being). This approach allows to balance the proportions of missing values in both treatment arms, regardless of the reasons of their missingness.

Overall, despite differences in both medical practice and missingness patterns reflected in the distinct sources of data emanating from these two countries, our two parallel target trials yielded similar results. Determining the extent to which different biases might have affected the results of each study but led to the same conclusions is challenging. When heterogeneous results are obtained, differences in risks of bias among studies can help explain variation. In contrast, more rigorous studies tend to yield consistent results despite data artifacts. Given our careful handling of death as a competing risk within a causal framework, it is unlikely that one target trial is biased toward overestimating metformin's effect, while the other is biased toward underestimating it, but that residual bias associated with unmeasured confounders counterbalances these positive and negative biases, respectively, to ultimately yield the same conclusion.

Comment #6

Addressing mortality bias is always important in any observational study of this kind and it has been addressed in different ways in prior study. The potential bias resulting from sample selection into metformin v. sulf. is not solved by this study design. There are multiple differences in observables between metform and sulf. (e.g. other health conditions) and while some of these may be controlled for in the analysis, the issue are the unobserved and unmeasured factors as in all observational studies that do not use some other quasi experimental design. The use of IPTW in CPH model estimation does not solve the problem of unobservable variable bias.

Author Reply

We agree that observational studies based on EHR data can be greatly affected by unobserved confounding. Building upon the reviewer's comment, we have updated our limitations paragraph in the discussion section, where we had pointed out this issue as the first limitation in the earlier versions, to further emphasize this matter and provide additional examples (Page 11):

“First, while we addressed many sources of confounding, there were likely others that were unavailable or inadequately measured. In particular, the level of education was systematically

unavailable in either dataset, of concern since it is known to affect both the exposure and outcomes of interest in this study. Additionally, relevant lifestyle factors, like diet and physical activity³³ and a genetic risk factor, ApoE genotypes, were unavailable. Furthermore, the strong effect of age, the changes in prescribing patterns of sulfonylureas and metformin over the observation period^{38,39}, gene-environment interactions, and the complex differences observed in age at baseline, length of follow-up, and calendar time across the two treatments raises the possibility of residual confounding.”

Comment #7

The assumptions of the PH model should not only be tested results should be reported.

Author Reply

We agree with the reviewer that testing the assumptions underlying Cox PH regressions models is important, as we have reported the results of these tests in the **Supplementary Materials** (Section VI titled “Statistical Methods”, see **Pages 36-42** with particular attention to the paragraphs titled “Assuming the proportional hazards model” on **Page 39** and “Checking a proportional hazards assumption using a nonparametric framework” on **Page 41**).

In each target trial, we tested the assumption of proportional hazards (PH) underlying the use of Cox PH models. Specifically, we checked graphically that the hazard ratio did not considerably vary over time – under the assumptions of no unmeasured confounding, positivity, and stable unit treatment value assumption (SUTVA). We also ran a global test, based on Schoenfeld residuals. Results from these tests confirmed that the PH assumption was not violated in the US RPDR dataset. In the UK CPRD cohort, the deviation of the HR for death without dementia from its constant (**Extended Data Figures 4b and 5b, Pages 47-48**) prompted us to additionally implement nonparametric approaches that helped assess the sensitivity of our results to the choice of modeling structure. Our findings were deemed robust to outcome model structure.

Comment #8

What other nonparametric approach was used and what were results?

Author Reply

Thank you for paying attention to the sensitivity analyses presented in our study. We implemented a nonparametric approach to assess the sensitivity of estimated effects to the choice of the modeling structure (Cox PH model). Specifically, for the transition hazards, we used the weighted Aalen-Johansen estimator, which is a generalization of the Nelson-Aalen estimator for multi-state models **[14] (ref. 12)**. Using this nonparametric framework, which does not impose any specific structure onto the hazard functions, is equivalent to relaxing the assumption of proportional hazards, in our case. In **Extended Data Fig. 4 (Page 47)**, we compare the log of hazard ratios estimated using Cox PH models with their nonparametric counterparts, for both outcomes of interest (i.e., time-to-dementia onset and time-to-death without dementia) and in each dataset. The corresponding risk differences between metformin and sulfonylureas, derived from hazards estimated via two distinct approaches, the semiparametric Cox PH vs. the nonparametric weighted Aalen-Johansen estimator, are provided in **Extended Data Figure 5 (Page 48)**.

Comment #9

The use of the word causal is too strong given these limitations.

Author Reply

Per the reviewer's recommendation, we softened our language and used the term "causal" in conjunction with "framework", indicating that while we attempt to determine causality with application of rigorous statistical methods, we acknowledge that we will fall short due to unmeasured confounding.

For example, we changed the title on **Page 1** to include "framework": "Drug repurposing of metformin for Alzheimer's disease: Causal competing risks framework in medical records and complementary systems pharmacology for biomarker identification".

Comment #10

Is the assumption that if has a dementia diagnosis in the first year, and excluded from sample, this also excludes persons with dementia diagnosis before the study start date? Typically, study design would include a wash out window of no diagnosis of dementia over several years prior to time drug initiation is measured as well as time of drug initiation.

Best practice for identifying dementia diagnosis using ICD9/10 codes include a second diagnosis to exclude rule out diagnoses.

Author Reply

Thank you for emphasizing the importance of outcome ascertainment. We indeed excluded patients who had a diagnosis of mild cognitive impairment (MCI) or dementia prior to antidiabetic drug initiation. We also implemented a one-year washout period following metformin or sulfonylureas monotherapy, resulting in the exclusion of patients with a MCI or dementia diagnosis within 12 months of treatment initiation. Per the reviewer's suggestion, we now explicitly use the terminology "washout period" (**Table 1, Page 21; Figure 1, Page 25; Extended Data Table 14, Page 66**).

In the US RPDR cohort (Figure 1a, Page 23), these design choices resulted in the exclusion of 1,125 patients diagnosed with dementia before antidiabetic treatment initiation and of 319 patients who received a diagnosis within a year from baseline. In the UK CPRD cohort (Figure 1b, Page 25), similar considerations led to the removal of 15,776 patients who had dementia before or within a year from baseline, died during the washout period, or had less than a year of follow-up. Moreover, to ensure that the medical history of patients entering our two target trials would be sufficiently detailed, we only included those registered with a practice for at least one year (UK CPRD) or those with at least one visit in the 18 months preceding baseline, in addition to having a primary care physician within the MGB Healthcare system (US RPDR).

Outcome ascertainment in observational studies relying on EHR data is generally challenging; determination of dementia onset, a clinical diagnosis, amplifies this concern due to lack of sensitivity [15,16,17] (refs. 44-46). We agree with the reviewer that requiring at least two instances of diagnosis codes in the EHR is often a good practice [18] (ref. 64); in our propensity score model, we have precisely used such a strategy to define the covariate that adjusts for prior history of cancer. However, in the context of our target trial emulations seeking evidence of any beneficial effect of metformin on cognitive function, we favored sensitivity over specificity. Therefore, any indication of dementia onset in the EHR, in the form of a diagnosis code, was considered a positive outcome. Further, to increase the sensitivity of our outcome definition for dementia, we also considered the prescription of dementia-related drugs as an indication of disease onset. The full list of diagnoses and medications used in our study can be found in **Extended Data Tables 10-13**.

Comment #11

Limitations of EHR records for real world data analysis should be described (e.g. if leave system no longer followed) as well as other limitations (who is recording the information and heterogeneity in EHR). Strengths should also be described.

Author Reply

Sharing about the limitations of our study with transparency: Limitations of EHR records for real-world data analysis are described in great detail on **Page 11** of the manuscript, including differential duration of the follow-up period and the possibility of non-random censoring. **In brief, we commented on the following limitations:** (a) unmeasured confounding, (b) missingness among measured covariates, (c) absence of linkage to claims data, (d) transitions to other antidiabetic regimens, (e) measurement errors in the primary outcome of interest (dementia onset), (f) lack of racial-ethnic diversity among patients and underrepresentation of minority populations, and (g) (the) approximations made in systems pharmacology experiments with respect to chronic drug exposure and cell types. Overall, heterogeneity among clinicians recording patient information in the EHR might affect the robustness of findings emanating from EHR-based emulated target trials – albeit rigorously deployed.

Highlighting the strengths of our study: Strengths of our DRIAD-EHR approach to drug repurposing for dementia are extensively developed on **Page 10** of the manuscript. **In short, we emphasize four strengths:** (a) a long follow-up period (up to 12 years in the US RPDR and 16 years in the UK CPRD) allowing examination of a therapeutically relevant timeframe for dementia onset currently infeasible in randomized controlled trials, (b) the harmonization of our causal inferential framework in two large EHR databases, (c) the careful handling of death without dementia as a competing risk, and (d) the implementation of complementary mechanistic studies analyzing gene expression changes in neural cells at pharmacologically relevant drug concentrations.

Importantly, we would like to emphasize the novelty of our approach:

1. Successful implementation of parallel EHR-based emulated target trials in two different data settings, with explicit handling of death as a competing risk for the first time within a causal framework;
2. Combination of EHR-based retrospective studies with *in-vitro* systems pharmacology experiments that can inform clinical trial design going forward.

Comment #12

The large differences in mortality of sulfonylurea users in the US (7.8%) v UK (37.4% die) study and the finding of age differences in the UK and none in the US suggest that the selection may be different into drug use across the US and UK and rather than provide for evidence of robustness, calls into question who is taking sulfonylureas compared to the more common first line treatment of metformin.

Author Reply

In both populations, we found that patients initiating on metformin were younger than their sulfonylureas counterparts (average at baseline: 68.6 vs. 72.2 in the US and 64.9 vs. 70.2 in the UK; see **Table 2 on Page 23**). In both cohorts, age was one of the most important variables in the propensity score model, as suggested by covariate balance graphs provided in **Extended Data Fig. 11 (Page 54)**. After application of IPTW weights, however, we were able to

successfully re-balance the treatment and control arms of our two target trials with respect to age, as illustrated on Extended Data Figs. 10 and 11 (Pages 53-54).

Of note, the UK CPRD observational study has a longer period of follow-up (up to 16 years) than that of the US RPDR (up to 12 years). This difference may partly explain overall differences in the percentage of patients who die or who die before having dementia. As shown on Extended Data Fig. 2 (Page 45), the proportion of patients still at risk by year 5 is similar among US RPDR and UK CPRD metformin initiators, while the equivalent curves among sulfonylureas start diverging at the end of year 2. Acknowledging the presence of death as a competing risk and the differential mortality rates between the UK CPRD and the US RPDR cohorts, we thus implemented a causal inferential framework that considered two outcomes in parallel: dementia onset and death without dementia. This feature helped to account for a different evolution of patient risk sets in each cohort over time, reflecting cohort-specific hazard functions for death that were calibrated flexibly in each dataset, using both a semi-parametric approach and a nonparametric approach to assess robustness. Furthermore, our competing risk analysis demonstrated the extent to which the risk of dementia depends on the baseline mortality rate of the population, with a concrete illustration in two large databases that can inform future studies. Our unique approach also provides insight into the heterogeneity among previous studies, which did not control for death as a competing risk (see Comment #15 below). An implication of this work is that incorporating a 3-5 year mortality index as an inclusion criterion in clinical trials evaluating the efficacy of metformin towards dementia onset may be beneficial.

Comment #13

Some conclusions may be consistent with the data but are not tested.

Whether the 'discordant hazard ratio and CIF results in the UK samples were due to the protective effect on both dementia and death is not tested - it may be consistent with the findings, but other things may also have changed over time (e.g. switching across diabetes treatments, onset of other health conditions and medications).

Author Reply

The current study is an intention-to-treat analysis aiming to estimate the relative effects of the antidiabetic treatment(s) assigned at baseline, as recorded in the EHR.

We agree that the population of patients initiating on metformin differs from the population initiating on sulfonylureas. Table 2 (Page 23) contrasts confounders observed in the EHR in the two arms of the target trial, prior to the application of any weighting strategy. To address this treatment selection bias, we built a propensity score model in each database and assigned an IPTW weight to every patient to re-establish balance between the two groups of antidiabetic drug initiators with respect to the set of observed confounders identified by our team. After applying the IPTW procedure, the absolute standardized mean difference significantly reduced, below the targeted threshold of 0.1 typically used in the literature, across confounders and in both cohorts. Interestingly, this decrease was especially marked for age, BMI, HbA1C, and prior history of hypertension (Extended Data Fig. 11, Page 54).

Our seemingly discordant findings for HRs and CIFs are not contradictory and can be explained mathematically [19]. Several such examples appear in the literature, with an intervention found effective in terms of one effect measure, but not in terms of another effect measure (see discussion in "Explanation in Causal Inference. Methods for Mediation and Interaction" [20]).

Importantly, we provided illustrative figures suggesting that the discordant HR and CIF results in the UK sample were associated with a protective effect of metformin on both dementia onset and death without dementia. By year 12, the risk difference between metformin and sulfonylureas initiators was greater than 0.1 in absolute value in the UK CPRD cohort (**Figure 4d, Page 28**), while it is only half of that in the US RPDR (**Figure 4c, Page 28**). We confirmed that this result held among patients initiating before age 70.

Since the use of antidiabetics was the primary exposure of interest, we investigated the possibility of antidiabetic treatment switches – from one type of monotherapy to another as well as from monotherapy to polytherapy, including the additional prescription of insulin. Notably, in the UK CPRD, participants with antidiabetic polytherapy or monotherapy other than metformin or sulfonylureas in the first year of treatment were excluded. Although such a time-varying exposure variable could be used in the future as part of a time-varying analysis where propensity scores are re-computed at a predefined frequency (e.g., every year), the current study is an intention-to-treat analysis contrasting treatments assigned at the onset of type 2 diabetes. Similarly, (precisely) because we conducted an intention-to-treat analysis, we accounted for other health conditions documented in the EHR and medications being prescribed at baseline only (including hypertension, CVD, CKD, stroke, cancer, COPD, and smoking).

Comment #14

Similarly, the conclusion that metformin is especially beneficial for those who initiate at a younger age again provokes the question, who does not initiate metformin and why?

Author Reply

Thank you for raising this question about selection into treatment. While today it would be much less common for an incident type 2 diabetic patient to initiate monotherapy with one of the sulfonylureas, we are leveraging the temporal evolution of practice patterns to conduct this emulated target trial, as there was more equipoise between initiating metformin and sulfonylureas in the past. We handled selection into metformin vs. sulfonylureas treatment by propensity scores. For instance, since metformin is contraindicated in individuals with severe renal impairment, renal function is an important consideration affecting the clinician's prescribing decision—irrespective of the patient's age at treatment initiation. Our finding that patients who initiated their antidiabetic treatment at a younger age benefited more from metformin monotherapy than their older counterparts cannot be an artifact of metformin initiators being younger on average than sulfonylureas initiators because we conducted and reported analyses of sensitivity in the stratum of participants initiating early (i.e., before age 70, see **Extended Data Tables 3 and 4 on Pages 57-58**). In that case, both treatment arms of the target trial are thus precisely of the same younger age. Of note, within the stratum of younger new diabetics, we also accounted for additional sources of selection bias—beyond age at treatment initiation—by recomputing propensity scores for each eligible patient using the same set of variables, except age at baseline.

Comment #15

In the discussion, it is not sufficient to only state that previous observational studies were mixed given the heterogeneity of study design. How do the results here compare to the most rigorous of prior studies?

Author Reply

Thank you for this question. Due to length constraints, we cannot comment in detail about each study. However, we would like to highlight key differences between our study and prior research, focusing on the rigorous work of **Orkaby et al. (2017) [7] (ref. 30)** and **Scherrer et al. (2019)**

[21] (ref. 29). In the table below, we compare the methods employed and the results emanating from each study with ours. In the future, a meta-analysis could contrast each of the previously published studies with ours in more depth.

The main difference between our study and the two aforementioned publications is that they do not consider death as a competing outcome and do not present sensitivity analyses to the underlying model structure, e.g., using nonparametric models to ensure the robustness of their findings. Another important difference is the selection bias due to the exclusion of patients with missing data.

The study by **Orkaby et al. (2017) [7] (ref. 30)** leverages VA data and thus suffers from a strong gender imbalance, despite female patients being at higher risk of dementia onset [22,23,24] and therefore being an important subpopulation to study. Additionally, the determination of dementia onset is made solely on the basis of diagnosis codes and does not consider drug prescriptions. However, given the undercoding of dementia diagnoses in the EHR [15,16,17] (refs. 44-46), we included dementia-related medications in our study to improve the sensitivity of outcome ascertainment. During the one-year washout period, we monitored patient status and removed those with incident MCI, while the Orkaby et al. study only excluded patients with dementia onset during that period.

The study by **Scherrer et al. (2019) [21] (ref. 29)** is more similar to ours and relies on the analysis of two distinct datasets (one based on VA data, thus also suffering from gender balance, and the other using Kaiser Permanente Washington data). However, it is also a complete case analysis with respect to creatinine, HbA1C, and demographic characteristics, introducing a risk of selection bias.

Further, we tested for differential treatment effects as a function of BMI at baseline and sex. Our study is thus complementary from Orkaby's work, which investigated the role of race as well as eGFR and HbA1C at baseline.

Finally, our study captures a longer calendar time period than the two presented here, with antidiabetic treatment initiation between 2001 and 2017 in the UK CPRD and between 2007 and 2017 in the US RPDR.

Methods

	Our study	Orkaby³⁰	Scherrer²⁹
Confounding by treatment indication	Renal impairment addressed. Stabilized IPTW. No trimming.	Renal impairment addressed. Non-stabilized IPTW. Trimming at PS=20.	Renal impairment addressed. Stabilized IPTW. Trimming at PS=10.
Competing death	Yes	No	No

Sensitivity to outcome model structure (Cox PH)	Yes , using a nonparametric approach	No	No
Age-based stratification	Dichotomous splits using 65, 70, and 75 as thresholds	Dichotomous split using 75 as a threshold	Categorical split: 50-64; 65-74; 75+
Handling of data missingness	Patients with missing race-ethnicity were considered. Missingness rates were as follow: UK CPRD (35%); US RPDR (<1%). Missing as a separate category for the following variables. UK CPRD: IMD (7%), smoking status (2%), HbA1C (21%), and BMI (3%). US RPDR: HbA1C (38%), BMI (32%).	Complete case analysis wrt to the following variables: race (used to calculate eGFR), HbA1C, BMI, and renal function (estimated via eGFR or presence of ICD codes).	Complete case analysis wrt to the following variables: creatinine and HbA1C, demographic characteristics.
Outcome ascertainment	ICD-9/10 codes or dementia-related drugs ICD-9/10 codes: 290.X, 294.X, and 331.X; 780.93, G30.X, and G31.X Dementia-related drugs: Donepezil, Galantamine,	ICD-9/10 codes only 290.x, 291.2, 294.1, 294.11, 331.x (except 331.83 [MCI]), 333.0, 333.4, 797, 332.0, 294.8, 046.1, and 046.3	ICD-9/10 codes only 290.0 290.1x 290.2x 290.3 290.4x 294.1x 294.2x 331.0 331.1x 331.2 331.82 331.83, used as a qualifying definition for dementia in sensitivity analysis

	Rivastigmine, and their respective brand names Aricept, Razadyne, Exelon		
--	--	--	--

Results

Study population

	Our study	Orkaby	Scherrer
Age at treatment initiation (average)	UK CPRD Metformin: 64.9 years old Sulfonylureas: 70.2 years old US RPDR Metformin: 68.6 years old Sulfonylureas: 72.2 years old	Older at treatment initiation (73.5, +/- 5.9 years old) because an inclusion criterion imposed age \geq 65 at new type 2 diabetes diagnosis	VHA: 60.8 years old KPW: 63.1 years old
Sex	UK CPRD (% male) Metformin: 57.4% Sulfonylureas: 58.3% US RPDR (% male) Metformin: 49.2% Sulfonylureas: 52.9%	% male Metformin: 98.8% Sulfonylureas: 99.0%	% male VHA: 96.8% KPW: 50.4%
Calendar time	UK CPRD: 2001-2017 US RPDR: 2007-2017	2001-2012	VHA: 2002-2012 KPW: 1996-2012
Washout period	One year	Two years	VHA: Two years KPW: No washout, continuous enrollment
Follow-up	UK CPRD (median) Metformin: 7.0 years Sulfonylureas: 7.0 years US RPDR (median) Metformin: 5.3 years Sulfonylureas: 5.3 years	Average of 5 years (+/-3.1)	VHA (median) Metformin: 6.4 years Sulfonylureas: 6.7 years KPW (median) Metformin: 6.1 years Sulfonylureas: 7.3 years

Statistical analysis

	Our study	Orkaby	Scherrer
Overall HR	UK CPRD HR = 0.86 (0.77-0.96) US RPDR HR = 0.81 (0.69-0.94)	Not provided	VHA HR=0.93 (0.87–0.99) KPW HR=0.89 (0.74–1.07) Pooled analysis HR=0.92 (0.87–0.98)
<= 75	UK CPRD HR = 0.83 (0.70-0.98) US RPDR HR = 0.77 (0.63-0.94)	0.89 (0.79–0.99)*	VHA* 50-64 = 0.86 (0.76–0.98); 65-74 = 0.87 (0.80–0.96) KPW* 50-64 = 1.33 (0.78-2.25); 65-74 = 0.79 (0.58-1.07)
>75	UK CPRD HR = 0.88 (0.78-1.00) US RPDR HR = 0.90 (0.73-1.10)	0.96 (0.87–1.05) *used <75 and >=75 instead	VHA* 75+ = 1.03 (0.9-1.14) KPW* 75+ = 0.79 (0.60-1.03)

*In the study, the authors used <75 and >=75, while we used <=75 and >75.

To reflect this comparison between **Orkaby et al. (2017) [7] (ref. 30)**, **Scherrer et al. (2019) [21] (ref. 29)**, and our study, we updated the discussion as follows: “Our results corroborate the benefits of metformin on dementia risk in type 2 diabetics reported in previous observational studies^{10,11,18,19,29,30}. Moreover, our competing risks analysis demonstrates how the risk of dementia depends on the baseline mortality rate of the population, a potential explanation of the neutral^{10,29} or deleterious¹¹ effect of metformin on dementia onset seen in other observational studies (**Extended Data Table 14**).”

To our knowledge, this work is the first attempt to comprehensively address competing death in a study of metformin and dementia, with a rigorous causal framework harmonized across two EHR databases.” Moreover, we added the three comparative tables as **Extended Data Table 14**.

Comment #16

There were others that used a match design.

Author Reply

Other studies used a matched design, but suffered from several limitations that we describe below. Our target trial methodologic approach is distinct.

UK CPRD nested case-control study by Imfeld et al. (2012) [25] (ref. 11): Matching criteria included age, sex, general practice, calendar time, and years of history in the database.

This study did not employ a new user design, therefore not accounting for prevalent user or survivor bias. However, survival is a key consideration, especially as metformin users had

generally fewer comorbidities and lived longer than sulfonylurea users, putting them at increased risk of developing dementia. In addition, this work was based on a limited patient sample size (n=570 cases and 747 controls). Furthermore, the authors did not address confounding by indication associated with renal function, despite it being a critical clinical consideration, as metformin is contraindicated in individuals with substantial renal impairment.

Taiwanese nested case-control study by Cheng et al. (2014) [26]:

This study, nested in a large population-based Taiwanese cohort, used a new-user design to assess the risk of incident dementia in patients with new-onset diabetes initiating monotherapy. Yet the study had a small sample size (n=1033 metformin and 796 sulfonylureas initiators, respectively). Further, covariate adjustment was limited to age, sex, hypertension, hyperlipidemia, and cerebrovascular disease. Importantly, the absence of adjustment for renal function limits the reliability of the authors' conclusions.

Unlike these two studies, we used the IPTW propensity approach to account for systematic differences in renal function between groups.

As part of our references, we also listed an analysis conducted by Sluggett et al. (2020) [27] (ref. 17) about the Finnish population and another by Wium et al. (2019) [28] (ref. 18) about the Danish population.

The Finnish study¹⁸ was a rigorous case-control study, although it only investigated Alzheimer's disease – rather than the outcome of dementia defined more broadly, across disease subtypes. Cases were matched with up to two controls by age, sex, and diabetes duration. The conclusion of these authors aligns with ours: *“Long-term and high-dose metformin use was associated with a lower risk of incident AD in older people with diabetes.”*

The Danish study¹⁹ is a nationwide nested case–control study. Controls were selected among cohort members who remained dementia-free. They were matched to case patients based on follow-up time and calendar year in a 1:4 ratio. They demonstrated that patients with diabetes who used metformin, DPP4 inhibitors, GLP1 analogs, or SGLT2 inhibitors had lower odds of developing dementia, even after adjustment for potential confounders and for use of other types of antidiabetic medication. Their conclusions align with ours.

Comment #17

The conclusions are a bit strong given that causality here too cannot be established.

Author Reply

Per your recommendation, we revised the language at the end of the abstract:

“Overall, our findings suggest that metformin might prevent dementia onset in patients with type 2 diabetes via actions beyond glycemic control. Together, our results may inform the design of clinical trials of metformin for dementia onset in cognitively normal seniors without diabetes to evaluate metformin's actions beyond diabetic control and suggest a pharmacodynamic CSF biomarker, SPP1, for metformin's action in the human brain.”

Similarly, at the end of the discussion on **Page 12**, we conclude the manuscript with a call for rigorous clinical trials to evaluate our findings from these emulated clinical trials:

“Future clinical trials of metformin in cognitively intact non-diabetics will determine whether anti-aging actions of metformin beyond hyperglycemic control could be an important component of strategies to prevent dementia.”

We would like to thank Reviewer 4 for their insightful questions. We hope that our responses further clarify the novelty of our research.

References

The following references were cited in the point-by-point response letter. Here is the full list for your convenience.

1. Berry SD, Ngo L, Samelson EJ, Kiel DP. Competing risk of death: an important consideration in studies of older adults. *J Am Geriatr Soc.* 2010;58(4):783-787. doi:10.1111/j.1532-5415.2010.02767.x
2. Buzkova P. Competing risk of mortality in association studies of non-fatal events. *PLoS One.* 2021;16(8):e0255313. Published 2021 Aug 13. doi:10.1371/journal.pone.0255313
3. Austin PC, Lee DS, Fine JP. Introduction to the Analysis of Survival Data in the Presence of Competing Risks. *Circulation.* 2016;133(6):601-609. doi:10.1161/CIRCULATIONAHA.115.017719
4. Noordzij M, Leffondré K, van Stralen KJ, Zoccali C, Dekker FW, Jager KJ. When do we need competing risks methods for survival analysis in nephrology?. *Nephrol Dial Transplant.* 2013;28(11):2670-2677. doi:10.1093/ndt/gft355
5. Feakins BG, McFadden EC, Farmer AJ, Stevens RJ. Standard and competing risk analysis of the effect of albuminuria on cardiovascular and cancer mortality in patients with type 2 diabetes mellitus. *Diagn Progn Res.* 2018;2:13. Published 2018 Jul 23. doi:10.1186/s41512-018-0035-4
6. Hsu CC, Wahlqvist ML, Lee MS, Tsai HN. Incidence of dementia is increased in type 2 diabetes and reduced by the use of sulfonylureas and metformin. *J Alzheimers Dis.* 2011;24(3):485-493. doi:10.3233/JAD-2011-101524
7. Orkaby AR, Cho K, Cormack J, Gagnon DR, Driver JA. Metformin vs sulfonylurea use and risk of dementia in US veterans aged ≥ 65 years with diabetes. *Neurology.* 2017;89(18):1877-1885. doi:10.1212/WNL.0000000000004586
8. Krasemann S, Madore C, Cialic R, et al. The TREM2-APOE Pathway Drives the Transcriptional Phenotype of Dysfunctional Microglia in Neurodegenerative Diseases. *Immunity.* 2017;47(3):566-581.e9. doi:10.1016/j.immuni.2017.08.008
9. Jack CR Jr, Bennett DA, Blennow K, et al. NIA-AA Research Framework: Toward a biological definition of Alzheimer's disease. *Alzheimers Dement.* 2018;14(4):535-562. doi:10.1016/j.jalz.2018.02.018
10. Livingston G, Huntley J, Sommerlad A, et al. Dementia prevention, intervention, and care: 2020 report of the Lancet Commission. *Lancet.* 2020;396(10248):413-446. doi:10.1016/S0140-6736(20)30367-6
11. Hernán MA, Robins JM. Using Big Data to Emulate a Target Trial When a Randomized Trial Is Not Available. *Am J Epidemiol.* 2016;183(8):758-764. doi:10.1093/aje/kwv254
12. Campbell JM, Bellman SM, Stephenson MD, Lisy K. Metformin reduces all-cause mortality and diseases of ageing independent of its effect on diabetes control: A systematic review and meta-analysis. *Ageing Res Rev.* 2017;40:31-44. doi:10.1016/j.arr.2017.08.003
13. Bannister CA, Holden SE, Jenkins-Jones S, et al. Can people with type 2 diabetes live longer than those without? A comparison of mortality in people initiated with metformin or sulphonylurea monotherapy and matched, non-diabetic controls. *Diabetes Obes Metab.* 2014;16(11):1165-1173. doi:10.1111/dom.12354

14. Andersen PK, Borgan O, Gill RD, Keiding N. Statistical models based on counting processes. 1993.
15. Chen Y, Tysinger B, Crimmins E, Zissimopoulos JM. Analysis of dementia in the US population using Medicare claims: Insights from linked survey and administrative claims data. *Alzheimers Dement (N Y)*. 2019;5:197-207. Published 2019 Jun 6. doi:10.1016/j.trci.2019.04.003
16. Benchimol EI, Smeeth L, Guttman A, et al. The REporting of studies Conducted using Observational Routinely-collected health Data (RECORD) statement. *PLoS Med*. 2015;12(10):e1001885. Published 2015 Oct 6. doi:10.1371/journal.pmed.1001885
17. Imam TH. Changes in metformin use in chronic kidney disease. *Clin Kidney J*. 2017;10(3):301-304. doi:10.1093/ckj/sfx017
18. Ash PE, Bieniek KF, Gendron TF, et al. Unconventional translation of C9ORF72 GGGGCC expansion generates insoluble polypeptides specific to c9FTD/ALS. *Neuron*. 2013;77(4):639-646. doi:10.1016/j.neuron.2013.02.004
19. Allignol A, Schumacher M, Wanner C, Drechsler C, Beyersmann J. Understanding competing risks: a simulation point of view. *BMC Med Res Methodol*. 2011;11:86. Published 2011 Jun 3. doi:10.1186/1471-2288-11-86
20. Böhnke JR. Explanation in causal inference: Methods for mediation and interaction. *Q J Exp Psychol (Hove)*. 2016;69(6):1243-1244. doi:10.1080/17470218.2015.1115884
21. Scherrer JF, Salas J, Floyd JS, Farr SA, Morley JE, Dublin S. Metformin and Sulfonylurea Use and Risk of Incident Dementia. *Mayo Clin Proc*. 2019;94(8):1444-1456. doi:10.1016/j.mayocp.2019.01.004
22. Beam CR, Kaneshiro C, Jang JY, Reynolds CA, Pedersen NL, Gatz M. Differences Between Women and Men in Incidence Rates of Dementia and Alzheimer's Disease. *J Alzheimers Dis*. 2018;64(4):1077-1083. doi:10.3233/JAD-180141
23. Niu H, Álvarez-Álvarez I, Guillén-Grima F, Aguinaga-Ontoso I. Prevalence and incidence of Alzheimer's disease in Europe: A meta-analysis. *Prevalencia e incidencia de la enfermedad de Alzheimer en Europa: metaanálisis*. *Neurología*. 2017;32(8):523-532. doi:10.1016/j.nrl.2016.02.016
24. Liu CC, Li CY, Sun Y, Hu SC. Gender and Age Differences and the Trend in the Incidence and Prevalence of Dementia and Alzheimer's Disease in Taiwan: A 7-Year National Population-Based Study. *Biomed Res Int*. 2019;2019:5378540. Published 2019 Nov 11. doi:10.1155/2019/5378540
25. Imfeld P, Bodmer M, Jick SS, Meier CR. Metformin, other antidiabetic drugs, and risk of Alzheimer's disease: a population-based case-control study. *J Am Geriatr Soc*. 2012;60(5):916-921. doi:10.1111/j.1532-5415.2012.03916.x
26. Cheng C, Lin CH, Tsai YW, Tsai CJ, Chou PH, Lan TH. Type 2 diabetes and antidiabetic medications in relation to dementia diagnosis. *J Gerontol A Biol Sci Med Sci*. 2014;69(10):1299-1305. doi:10.1093/gerona/glu073
27. Slugggett JK, Koponen M, Bell JS, et al. Metformin and Risk of Alzheimer's Disease Among Community-Dwelling People With Diabetes: A National Case-Control Study. *J Clin Endocrinol Metab*. 2020;105(4):dgz234. doi:10.1210/clinem/dgz234
28. Wium-Andersen IK, Osler M, Jørgensen MB, Rungby J, Wium-Andersen MK. Antidiabetic medication and risk of dementia in patients with type 2 diabetes: a nested case-control study. *Eur J Endocrinol*. 2019;181(5):499-507. doi:10.1530/EJE-19-0259

Reviewers' Comments:

Reviewer #2:

Remarks to the Author:

I appreciate the tremendous work by the authors to improve the paper. I agree with reviewer 4 that it is still weak to have the conclusion that "our results may inform the design of clinical trials of metformin for dementia onset in cognitively normal seniors without diabetes to evaluate metformin's actions beyond diabetic control" (see abstract), based on the evidence provided in the manuscript.

There are clinical trials using metformin for non-diabetic MCIs:

<https://clinicaltrials.gov/ct2/show/NCT04098666>. But not for cognitively normal individuals. It should be further discussed

Reviewer #4:

Remarks to the Author:

Thank you for your thoughtful and detailed responses to the comments. Careful testing of hypotheses with real world data may lead to novel insights into the potential for repurposing drugs to reduce AD risk. An important aspect of this approach is transparency in assumptions and in study limitations and conclusions. My main comments were about this, and the authors were responsive and the revision addressed my main concerns.

RESPONSE TO THE REVIEWERS' COMMENTS:

Reviewer #2 (*Remarks to the Author*):

I appreciate the tremendous work by the authors to improve the paper. I agree with reviewer 4 that it is still weak to have the conclusion that "our results may inform the design of clinical trials of metformin for dementia onset in cognitively normal seniors without diabetes to evaluate metformin's actions beyond diabetic control" (see abstract), based on the evidence provided in the manuscript.

There are clinical trials using metformin for non-diabetic MCIs: <https://clinicaltrials.gov/ct2/show/NCT04098666>. But not for cognitively normal individuals. It should be further discussed.

Thank you, our efforts to address your constructive concerns raised in last review improved our manuscript. The target trial presented in this manuscript demonstrated that metformin reduces the risk of MCI/dementia onset in two very distinct cohorts (US and UK) beyond glycemic control using two orthogonal approaches: 1. by quantifying times to MCI/dementia onset in metformin initiators relative to initiators of sulfonylurea medications using causal inference approaches and 2. by controlling for hemoglobin A1C and not losing the protective effect against dementia. These reproducible results generate the hypothesis that metformin acts in a distinct, off-indication (beyond glycemic control) manner to reduce the risk of MCI/dementia onset. Our closing statement is not a conclusion, but rather a suggestion to rigorously test this hypothesis in a randomized clinical trial to determine whether the hypothesized indirect, off-indication action that may extend to non-diabetics.

The ongoing Metformin in Alzheimer' Dementia Prevention (MAP) clinical trial in non-diabetics who already have the diagnosis of MCI (NCT04098666) is distinct from our proposed clinical trial. All participants in the MAP clinical trial would not have met the eligibility criteria of our target trial, since they reached the diagnosis of MCI. Our work provides evidence that metformin can reduce the risk of onset of MCI or dementia, whereas the clinical trial is aimed to determine if metformin reduces the risk of cognitive progression (SRT) in symptomatic non-diabetic patients.

Based on these comments, we will revise the abstract to state, "Together, our findings suggest that metformin might reduce the risk of dementia in diabetes patients by a mechanism beyond glycemic control, and that SPP1 is a candidate biomarker for metformin's action in the brain."

Reviewer #4 (*Remarks to the Author*):

Thank you for your thoughtful and detailed responses to the comments. Careful testing of hypotheses with real world data may lead to novel insights into the potential for repurposing drugs to reduce AD risk. An important aspect of this approach is transparency in assumptions and in study limitations and conclusions. My main comments were about this, and the authors were responsive and the revision addressed my main concerns.

Thank you for your careful review, which improved our manuscript substantially.